# *FAM13A* affects body fat distribution and adipocyte function

Mohsen Fathzadeh [1,2,3,14], Jiehan Li[1,2,3,14], Abhiram Rao[1,4,14], Naomi Cook [5], Indumathi Chennamsetty[1,2], Marcus Seldin [6], Xiang Zhou[1,2], Panjamaporn Sangwung [1,2,3], Michael J. Gloudemans [7], Mark Keller[8], Allan Attie [8], Jing Yang [9], Martin Wabitsch[10], Ivan Carcamo-Orive [1,2,3], Yuko Tada [1,2], Aldons J. Lusis [6], Myung Kyun Shin[11], Cliona M. Molony[11], Tracey McLaughlin[3,12], Gerald Reaven[1,2,3], Stephen B. Montgomery [3,7,12,13], Dermot Reilly [11], Thomas Quertermous [1,2,3], Erik Ingelsson [1,2,3,15✉] & Joshua W. Knowles [1,2,3,15✉]

Genetic variation in the *FAM13A* (Family with Sequence Similarity 13 Member A) locus has been associated with several glycemic and metabolic traits in genome-wide association studies (GWAS). Here, we demonstrate that in humans, *FAM13A* alleles are associated with increased *FAM13A* expression in subcutaneous adipose tissue (SAT) and an insulin resistance-related phenotype (e.g. higher waist-to-hip ratio and fasting insulin levels, but lower body fat). In human adipocyte models, knockdown of *FAM13A* in preadipocytes accelerates adipocyte differentiation. In mice, *Fam13a* knockout (KO) have a lower visceral to subcutaneous fat (VAT/SAT) ratio after high-fat diet challenge, in comparison to their wild-type counterparts. Subcutaneous adipocytes in KO mice show a size distribution shift toward an increased number of smaller adipocytes, along with an improved adipogenic potential. Our results indicate that GWAS-associated variants within the *FAM13A* locus alter adipose *FAM13A* expression, which in turn, regulates adipocyte differentiation and contribute to changes in body fat distribution.

[1] Department of Medicine, Division of Cardiovascular Medicine, Stanford University School of Medicine, Stanford, CA, USA. [2] Stanford Cardiovascular Institute, Stanford University, Stanford, CA, USA. [3] Stanford Diabetes Research Center, Stanford University, Stanford, CA, USA. [4] Bioengineering Department, School of Engineering and Medicine, Stanford, CA, USA. [5] Department of Medical Sciences, Molecular Epidemiology, Uppsala University, Uppsala, Sweden. [6] Department of Human Genetics, David Geffen School of Medicine, UCLA, Los Angeles, CA, USA. [7] Department of Genetics, Stanford University, California, CA, USA. [8] Department of Biochemistry, University of Wisconsin, Madison, WI, USA. [9] Department of Comparative Biosciences, College of Veterinary Medicine, University of Illinois at Urbana–Champaign, Urbana, IL, USA. [10] Division of Paediatric Endocrinology and Diabetes, Department of Paediatrics and Adolescent Medicine, University of Ulm, Ulm, Germany. [11] Genetics and Pharmacogenomics, Merck & Co., Inc., Kenilworth, NJ, USA. [12] Department of Medicine, Division of Endocrinology, Stanford University School of Medicine, Stanford, CA, USA. [13] Department of Pathology, Stanford University, California, CA, USA. [14] These authors contributed equally: Mohsen Fathzadeh, Jiehan Li, Abhiram Rao. [15] These authors jointly supervised this work: Erik Ingelsson, Joshua W. Knowles. ✉email: eriking@stanford.edu; knowlej@stanford.edu

Excess adiposity is known to be a key contributing factor to the development of insulin resistance (IR), type 2 diabetes (T2D), dyslipidemia, and cardiovascular disease (CVD)[1]. Body mass index (BMI) is a measure of adiposity[1]. However, not all obese individuals (BMI ≥30 kg/m$^2$) suffer from metabolic complications (i.e. metabolically healthy obese). Concurrently, not all normal-weight individuals (18.5 kg/m$^2$ ≤ BMI ≤ 25 kg/m$^2$) are protected from metabolic comorbidities (i.e. metabolically obese normal weight)[2]. Recently, we and others[1,3,4] identified a cluster of common risk variants from genome-wide association studies (GWAS) that are associated with normal or lower adiposity and, at the same time, with a poorer cardiometabolic profile (e.g. increased fasting insulin (FI), increased triglycerides, and decreased HDL-cholesterol levels), resembling a normal weight but metabolically unhealthy phenotype. By using refined measures of body composition and fat distribution, follow-up analysis shows that the lower BMI-associated variants are associated with a higher waist-to-hip ratio (WHR) and visceral-to-subcutaneous adipose tissue ratio (VAT/SAT), which is driven by a fat deposition shift from subcutaneous depots around the hip to visceral depots around the waist[5]. Among these unfavorable fat distribution risk loci are variants located in or near FAM13A (Family With Sequence Similarity 13 Member A)[4,5].

Common non-coding variants of FAM13A have been identified in several GWAS of anthropometric and glycemic traits. For instance, the same lead variant (rs3822072) is associated with higher fasting insulin levels adjusted for BMI[3] and lower HDL cholesterol levels[6]. Another common non-coding FAM13A variant (rs9991328, in high linkage disequilibrium [LD] with rs3822072) is associated with increased WHR adjusted for BMI (WHRadjBMI)[1]. However, the biological function of FAM13A in this context is not fully understood. So far, FAM13A is best known to be associated with human lung diseases, including chronic obstructive pulmonary disease, asthma, and pulmonary fibrosis[7]. A recent study also revealed an association between FAM13A variants and liver cirrhosis[8]. Given that FAM13A is highly expressed in human adipose tissue[9], we hypothesized that a comprehensive study combining human genetics with in vivo and in vitro experiments focusing on adipose tissue would uncover the biological mechanism linking FAM13A with body fat distribution.

Here, we first explore the phenotypic effects of alleles within the FAM13A locus using human genomic and adipose gene expression datasets. Secondly, we perform in vitro loss-of-function studies in human adipocytes to investigate the cell-autonomous effect of FAM13A on adipocyte biology. Lastly, we assess the effects of Fam13a knockout (KO) in mice on glucose and lipid metabolism, adipocyte size and number distribution, and overall gene expression in adipose tissue. Our human genetics studies suggest that the disease risks conferred by the FAM13A variants are mediated by a change in FAM13A expression specifically in SAT. Our in vitro and in vivo experimental studies show that FAM13A plays a role in adipocyte differentiation and function.

## Results

### IR SNPs in FAM13A associate with higher FAM13A levels in SAT. 

We performed fine-mapping and colocalization analyses to explore several lines of evidence suggesting that non-coding GWAS-associated variants in the FAM13A locus have functional impact on FAM13A in adipose tissue. First, variants in the FAM13A locus, including rs3822072, rs1377290, and rs9991328, have been reported as lead variants within the locus across GWAS studies of IR-related traits such as body fat percentage, WHRadjBMI, fasting insulin levels. Next, these variants were

associated with FAM13A expression in subcutaneous adipose tissue (SAT), but not in visceral adipose tissue (VAT; Fig. 1) in data from the Genotype-Tissue Expression project (GTEx v7). Finally, variants in the locus showed some evidence of enrichment in regions annotated with the H3K27ac epigenetic mark in adipose nuclei as compared to other tissues (Fig. 1). Bayesian fine-mapping analysis of the SAT expression association signal using the FINEMAP algorithm revealed that rs9991328, which is associated with increased gene expression ($b = 0.22 \pm 0.03$, $p = $ 1e-08), had the highest posterior probability of driving the signal (fine-mapping posterior probability = 0.66) among all the variants in the region; we retained rs9991328 as the lead variant for further analyses. This variant is located within an intronic region ~3.2 Mb away from the transcription start site of FAM13A. The most commonly reported GWAS lead variant, rs3822072, was among the top 5 variants in the fine-mapping analysis (posterior probability = 0.06). In SAT, rs9991328 showed a robust association with FAM13A expression ($P = $ 1e-08) and was not a cis-eQTL (expression quantitative trait locus) for any other genes with transcription start sites within 1 Mb of the variant. In contrast, rs9991328 showed a weaker association with FAM13A expression ($P > $ 1e-05) in VAT. The variant was also not a significant trans-eQTL for any genes in adipose tissue (data from GTEx v6p). Two-sided t-tests were used in cis-eQTL mapping. To further evaluate whether FAM13A expression mediates the genotype-phenotype associations at this locus, we performed colocalization analysis of FAM13A gene expression in SAT, VAT, liver, and skeletal muscle from GTEx (v7) with seven IR-related GWAS traits using eCAVIAR[10]. We observed colocalizations (posterior probability >0.8; lead variant rs9991328) between SAT FAM13A expression and fasting insulin adjusted for BMI, WHRadjBMI, triglycerides, and HDL-cholesterol. We did not observe similar colocalizations or strong associations between variants in the locus with gene expression in the other investigated tissues, suggesting that FAM13A plays a specific role in SAT (Fig. 1 and Supplementary Fig. 1). We also performed a phenome-wide association study (PheWAS) in 337,536 individuals from the UK Biobank[11,12] using the directly genotyped variant rs1377290 as a proxy for the imputed variant rs9991328 (LD $R^2 = 1.0$). The rs1377290 T allele was associated with increased WHR, and decreased trunk/body fat percentage, mean platelet volume, and BMI (Fig. 2a). There were no other significant associations across the phenotypic spectrum.

We then evaluated correlations of FAM13A expression data in SAT with IR-related traits in the Metabolic Syndrome in Men (METSIM)[13,14] cohort. FAM13A expression levels in SAT adjusted for BMI were positively correlated with WHR and fasting insulin, and inversely correlated with fat mass (Fig. 2b). These correlations replicate the genomic associations at the FAM13A locus, where risk alleles resulting in higher SAT FAM13A expression are associated with higher WHRadjBMI and fasting insulin adjusted for BMI, and lower fat mass and BMI. Pathway analysis with ConsensusPathDB[15] using adipose expression data from the METSIM[13] and STAGE[16] cohorts showed that mitochondria TCA cycle, lipid metabolism, cell cycle and Wnt-signaling pathways were overrepresented in the genes correlated with FAM13A (Supplementary Fig. 2).

Collectively, these human data support the notion that FAM13A-increasing variants may predispose individuals to a normal weight but metabolically unhealthy phenotype, while decreased FAM13A is associated with favorable adiposity. Considered jointly with GWAS studies, our results from colocalization, eQTL and PheWAS analyses strongly indicate that FAM13A is the causal gene in this locus, which prompted further functional and metabolic assessments with a special focus on the role of FAM13A in SAT.

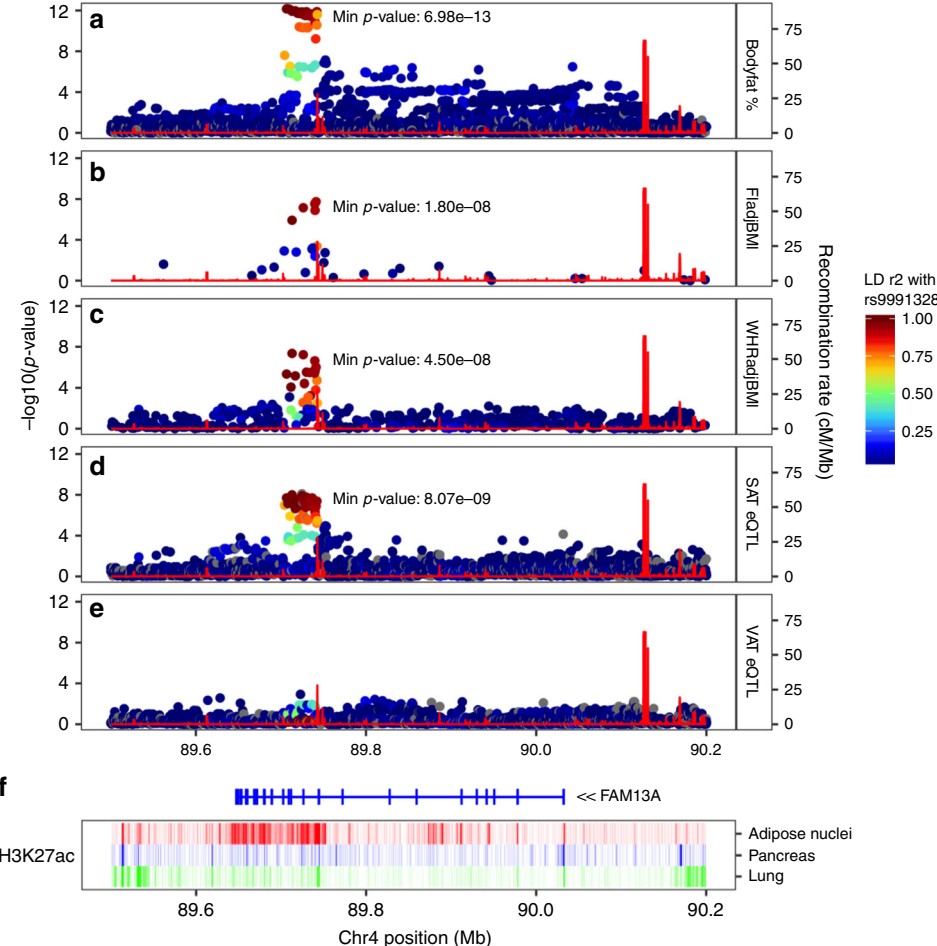

**Fig. 1 Regulatory variants within the *FAM13A* locus are associated with several IR traits.** Intronic non-coding variants in the *FAM13A* locus show associations with several IR-related and metabolic traits, including body fat percentage (**a**), and fasting insulin (**b**) and waist-hip ratio (**c**) (adjusted for BMI) across cohorts and studies. These GWAS association signals colocalize with FAM13A eQTL association signal in subcutaneous adipose tissue (from GTEx v7, **d**). However, this association is not robust in visceral adipose tissue (**e**). **f** The variants show evidence of enrichment in a regulatory active H3K27ac region in adipose nuclei. Annotations from pancreas and lung are shown as negative comparators.

***FAM13A* knockdown effects on human adipocyte differentiation.** To translate the potential role of *FAM13A* into a human cell model of adipocytes, we studied the expression and function of *FAM13A* during the differentiation of Simpson-Golabi-Behmel syndrome (SGBS) preadipocytes – a cell line isolated from subcutaneous depots of an infant with SGBS[17]. Expression of *FAM13A* was significantly increased during adipocyte differentiation (Fig. 3a). *FAM13A* knockdown by siRNA in SGBS preadipocytes (Fig. 3b) resulted in upregulation of adipogenesis markers (e.g. *CEBPA* and *PPARG*) (Fig. 3c) as well as beige adipogenic markers (e.g. *PGC1A* and *UCP1*) (Supplementary Fig. 3A) in the early differentiation stage (Day 5). Following 90–95% of *FAM13A* knockdown achieved in preadipocytes by CRISPR-interference (CRISPRi) (Fig. 3d), a similar induction of adipogenic markers (*CEBPA* and *PPARG*) in early differentiating adipocytes (Day 5) (Fig. 3e) confirmed a more active adipogenic capability of *FAM13A*-deficient preadipocytes. To investigate the impact of *FAM13A* on mature adipocyte function, we introduced siRNAs in the late stage of adipocyte differentiation (Day 8) to separate the confounding effect of *FAM13A* on adipogenesis. Under this experimental timing and condition, although *FAM13A* knockdown (Fig. 3f) did not significantly affect gene expression of adipokines (e.g. *ADIPOQ* and *LEP*) (Supplementary Fig. 3B) and lipolysis (Supplementary Fig. 3C), there was an increased trend of glucose uptake into *FAM13A*-deficient SGBS

adipocytes under both basal and insulin-stimulated conditions (Fig. 3g). Taken together, these results demonstrate that *FAM13A* knockdown in human fat cells of subcutaneous origin may favor adipocyte differentiation and function in glucose deposition.

**Effect of *Fam13a* knockout on body fat distribution of mice.** Next, we investigated the metabolic consequences of whole-body *Fam13a* loss in mice. Under normal chow-fed dietary conditions, male WT and *Fam13a* KO mice showed no significant difference in body weight (Supplementary Fig. 4A), mild decreases in plasma concentrations of fasting glucose, insulin, free fatty acids, and triglycerides (Supplementary Fig. 4B), and no change in glucose or insulin tolerance tests (Supplementary Fig. 4C, D). No differences in the above metabolic phenotypes were observed between female WT and *Fam13a* KO mice (Supplementary Fig. 5).

To explore the potential effects of *FAM13A* on the amount and distribution of body fat, we assessed gross mass and histological appearance of inguinal white adipose tissue (iWAT) and gonadal white adipose tissue (gWAT) as representatives of SAT and VAT, respectively. Neither the absolute amount of fat depots (Supplementary Fig. 4E) nor the ratio of VAT/SAT (Supplementary Fig. 4F) were different from WT to *Fam13a* KO mice on chow. Normal histological appearance was maintained in *Fam13a* KO mice

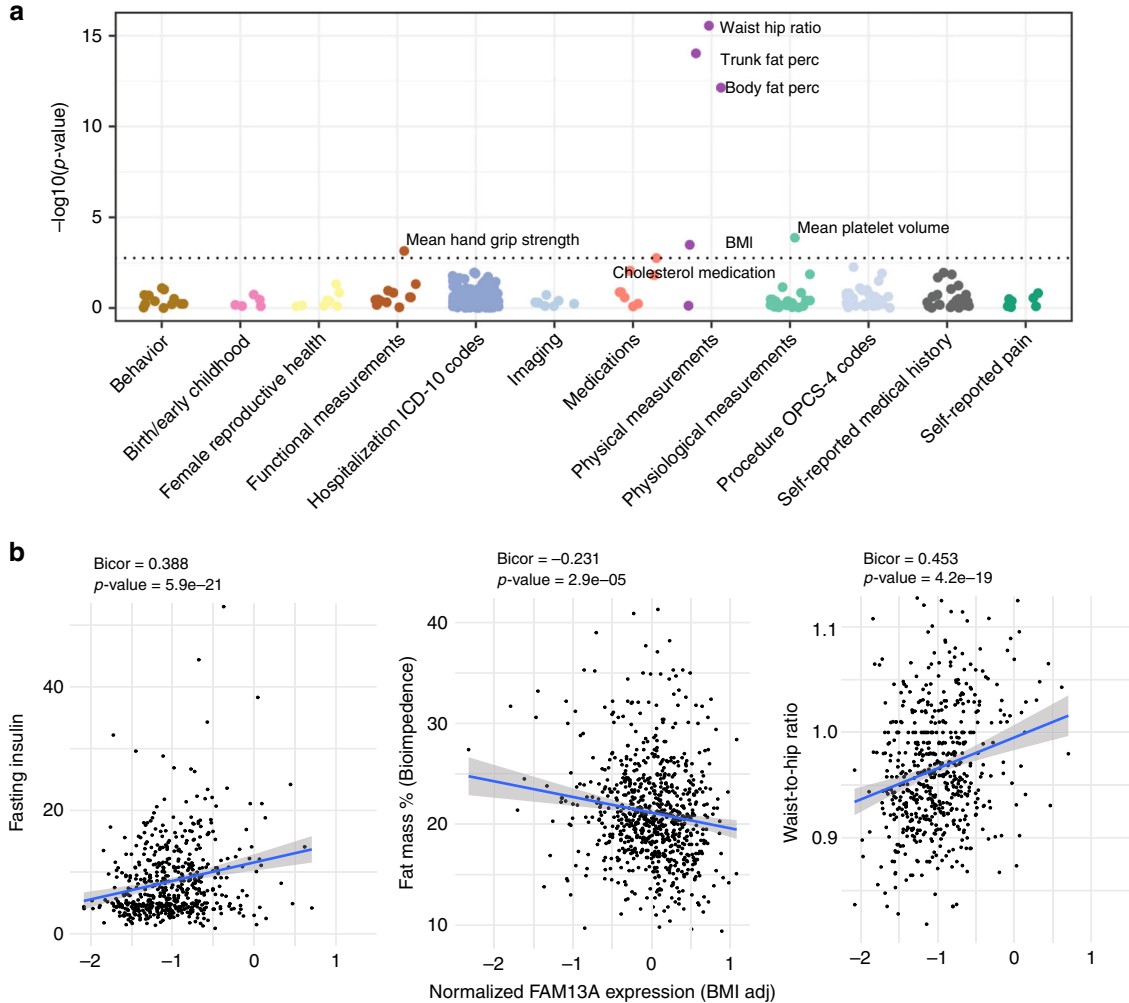

**Fig. 2 PheWAS analysis of *FAM13A* variants. a** Phenome-wide association analysis of the variant rs1377290 (used as a proxy for finemapping lead variant rs9991328, LD R2 1.0) performed in ~337 K individuals in the U.K. Biobank shows associations with metabolically related phenotypes, including body fat and trunk fat percentage. Associations significant at 10% FDR are labeled. **b** In male subjects in the METSIM cohort, normalized *FAM13A* expression in subcutaneous adipose tissue (adjusted for BMI) shows a positive correlation with fasting insulin and waist-hip-ratio, but a negative correlation with bioimpedance measured fat percentage. Two-sided *t*-tests were used in cis-eQTL mapping.

for both SAT and VAT (Fig. 4a). There was no significant difference in total number of adipocytes per depot (Fig. 4b) or in the average size of adipocytes between WT (SAT diameter = 40.4 ± 17.6 μm, VAT diameter = 51.7 ± 24.7 μm, n = 7) and KO (SAT diameter = 36.1 ± 12.1 μm, VAT diameter = 56.5 ± 22.8 μm, n = 7) (Fig. 4c). However, when comparing the size distribution curve of adipocytes in SAT and VAT, there was a slight increase in the number of small adipocytes in SAT (Fig. 4a, d, Supplementary Fig. 4G, H) with a corresponding modest increase in the number of relatively large adipocytes in VAT (Fig. 4a, e, Supplementary Fig. 4G, I) of *Fam13a* KO mice. These results indicate that depot-specific size distribution of adipocytes may be associated with the presence of *Fam13a* in adipose tissue.

To explore if the metabolic influence of *Fam13a* deletion is more profound under diet-induced obesity, mice were challenged with a high-fat diet (HFD) for 12–14 weeks. Male *Fam13a* KO mice gained slightly more weight compared to WT (Fig. 4f), while maintaining a comparable profile of plasma glucose, insulin, free fatty acids and triglycerides (Supplementary Fig. 6A) as well as a similar level of response to both glucose and insulin challenge (Supplementary Fig. 6B, C). Of note, despite higher overall weight compared to WT mice, male *Fam13a KO* mice maintained a lower weight of VAT (Fig. 4g) and a reduced VAT/SAT ratio

(Fig. 4h), indicating a potential role of *Fam13a* perturbation in driving a shift of fat deposition away from visceral depots. Metabolic profiling of Female *Fam13a KO* mice after HFD challenge did not show significant changes as observed in male mice (Supplementary Fig. 7).

**Effect of *Fam13a* KO on subcutaneous adipogenesis of mice.** To gain insight into the cellular mechanisms underlying *Fam13a*'s role in fat distribution, we examined the expression and function of *Fam13a* in adipocyte development. In WT mice, *Fam13a* expression was enriched in primary mature adipocytes as compared to freshly prepared stromal vascular fraction of cells (SVFs), which comprise adipocyte progenitor cells (Fig. 5a). After growing SVFs into confluence (i.e. Day 0) and inducing them to adipogenic differentiation, *Fam13a* expression was found to be elevated during in vitro adipogenesis (Fig. 5a). The identity of non-fat vs. fat cell fractions and the successful adipogenic differentiation of SVFs were confirmed by the distinct expression pattern of adiponectin-a mature adipocyte-specific gene (Fig. 5b).

Next, the adipogenic potential of SVFs, isolated from SAT of WT and *Fam13a* KO mice, were assessed by comparing gene expression of adipogenic markers (*Pparg, Cebpa, Fabp4*), ORO

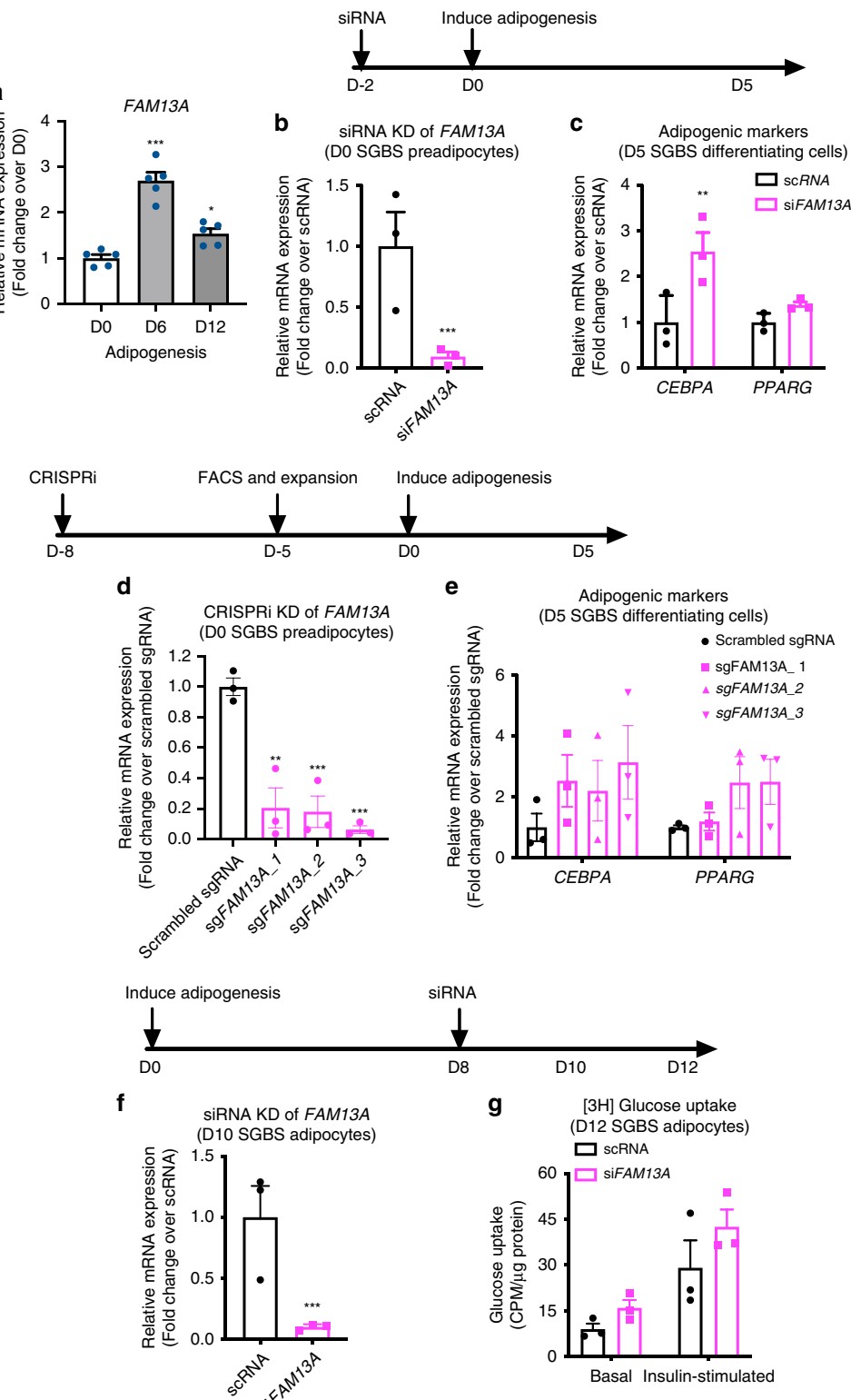

**Fig. 3 Effects of *FAM13A* knockdown in human adipocyte differentiation. a** *FAM13A* mRNA expression during adipogenesis of human SGBS preadipocytes. **b** mRNA expression of *FAM13A*, measured 2 days after siRNA transfection and before initiation of adipogenesis. **c** mRNA expression of adipogenic markers (*CEBPA, PPARG*), measured 5 days after adipogenic induction in cells transfected with scrambled siRNA or si*FAM13A*. **d** mRNA expression of *FAM13A*, measured 8 days after lentiviral infection and before initiation of adipogenesis. **e** mRNA expression of adipogenic markers (*CEBPA, PPARG*), measured 5 days after adipogenic induction in cells transduced with scrambled sgRNA or three independent sgRNAs against *FAM13A*. **f** mRNA expression of *FAM13A*, measured on D10 of adipogenesis and 2 days after siRNA transfection. **g** Basal and insulin-stimulated [3H] glucose uptake, measured on D12 differentiated adipocytes and 4 days after siRNA transfection. Data are presented as mean ± SEM. $n = 3$ independent experiments. *$p < 0.05$, **$p < 0.01$, ***$p < 0.001$, unpaired $t$ test (for **b** and **f**), one-way ANOVA followed by Turkey's multiple comparison test (for **a** and **d**), 2-way ANOVA followed by Sidak's multiple comparison test (for **c**). Source Data are provided as a Source Data file.

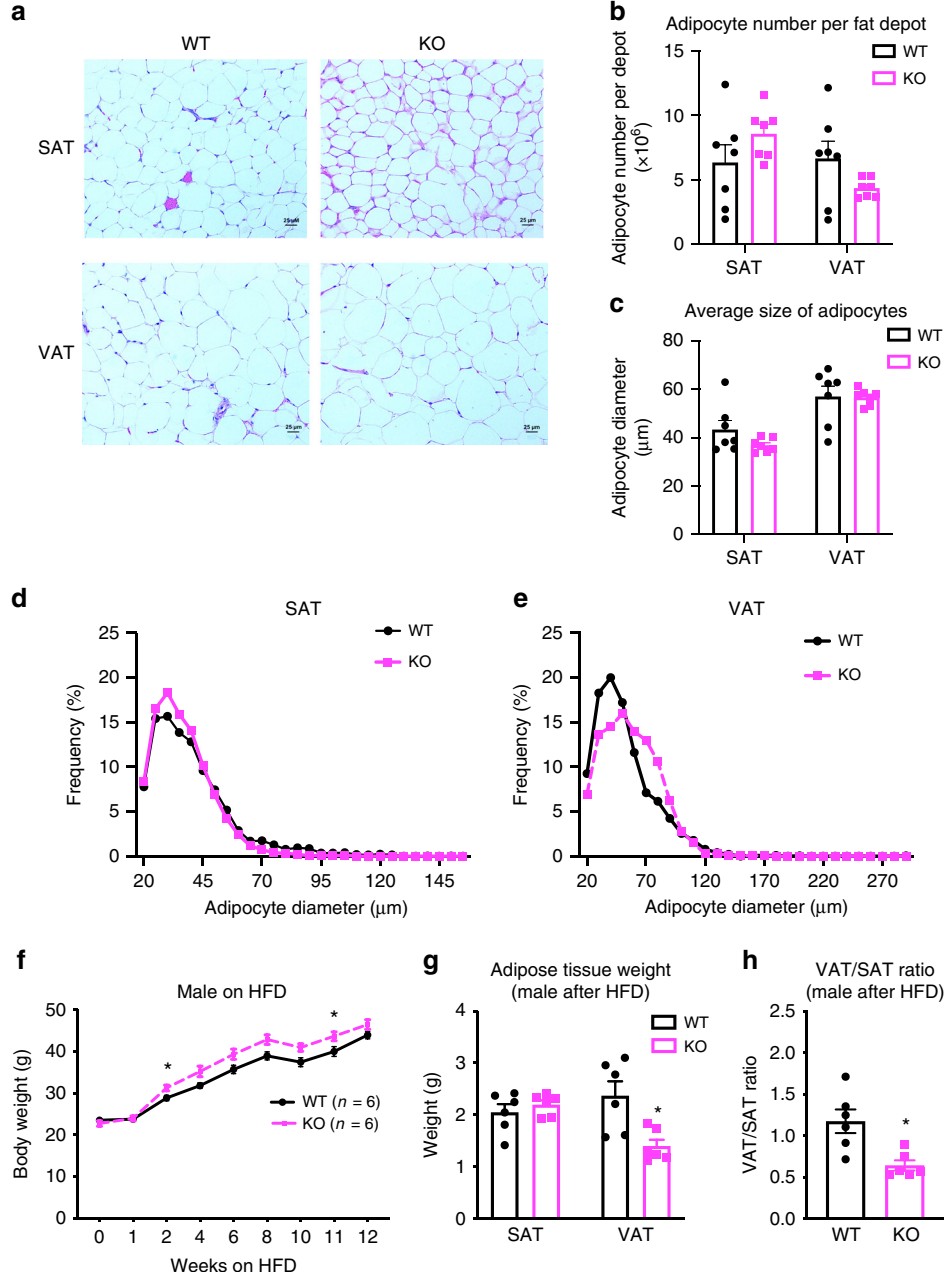

**Fig. 4 Metabolic profiling of male *Fam13a* KO mice. a** Representative H&E images (×20 magnification) of VAT and SAT in 14 weeks old of male WT and *Fam13a* KO mice fed on chow. (n = 7 per group, ×20 magnification, scale bar = 25 μm). **b, c** Adipocyte number per SAT or VAT depot (**b**) and the average diameter of adipocytes in SAT or VAT (**c**), in 14-week-old male WT and *Fam13a* KO mice fed on chow. (n = 7 per group). **d, e** Adipocytes size distribution in SAT (**d**) or VAT (**e**) of 14 weeks old of male WT and *Fam13a* KO mice fed on chow. (n = 7 per group; 2000–2200 cells per animal were used for adipocyte diameter determination; statistical differences between WT and KO mice in adipocyte diameter distribution curve were estimated by Kolmogorov-Smirnov test, ns for both SAT and VAT). **f** Body weight of male WT and *Fam13a* KO mice fed on HFD from 8 weeks to 20 weeks old. (n = 6 per group). **g** Tissue mass of male WT and *Fam13a* KO mice after 14 weeks HFD. (n = 6 per group). **h** Ratio of VAT/SAT, based on tissue mass, in male WT and *Fam13a* KO mice after 14 weeks HFD. (n = 6 per group). All values are presented as mean ± SEM. *$p < 0.05$, unpaired $t$-test. Source Data are provided as a Source Data file.

staining of lipid droplets and intracellular triglyceride levels. Results from all these methods showed a similar trend toward induced adipogenesis in *Fam13a* KO mice, although none of them reached statistical significance (Fig. 5c–f). The differentiated subcutaneous adipocytes from *Fam13a* KO mice also expressed a marginally but not significantly higher level of beige adipocyte markers (e.g. *Pgc1a* and *Ucp1*; Supplementary Fig. 8). These subtle but noticeable increases in subcutaneous adipocyte

differentiation may be relevant to the distribution shift toward an increased number of smaller adipocytes in SAT of male *Fam13a* KO mice observed earlier (Fig. 4a, d, Supplementary Fig. 4G, H). There was no difference in subcutaneous adipogenesis between female WT and KO mice (Supplementary Fig. 5D, E).

To explore the functional consequences of the modestly improved subcutaneous adipogenesis, we quantified the basal and insulin-stimulated glucose uptake in newly differentiated cells

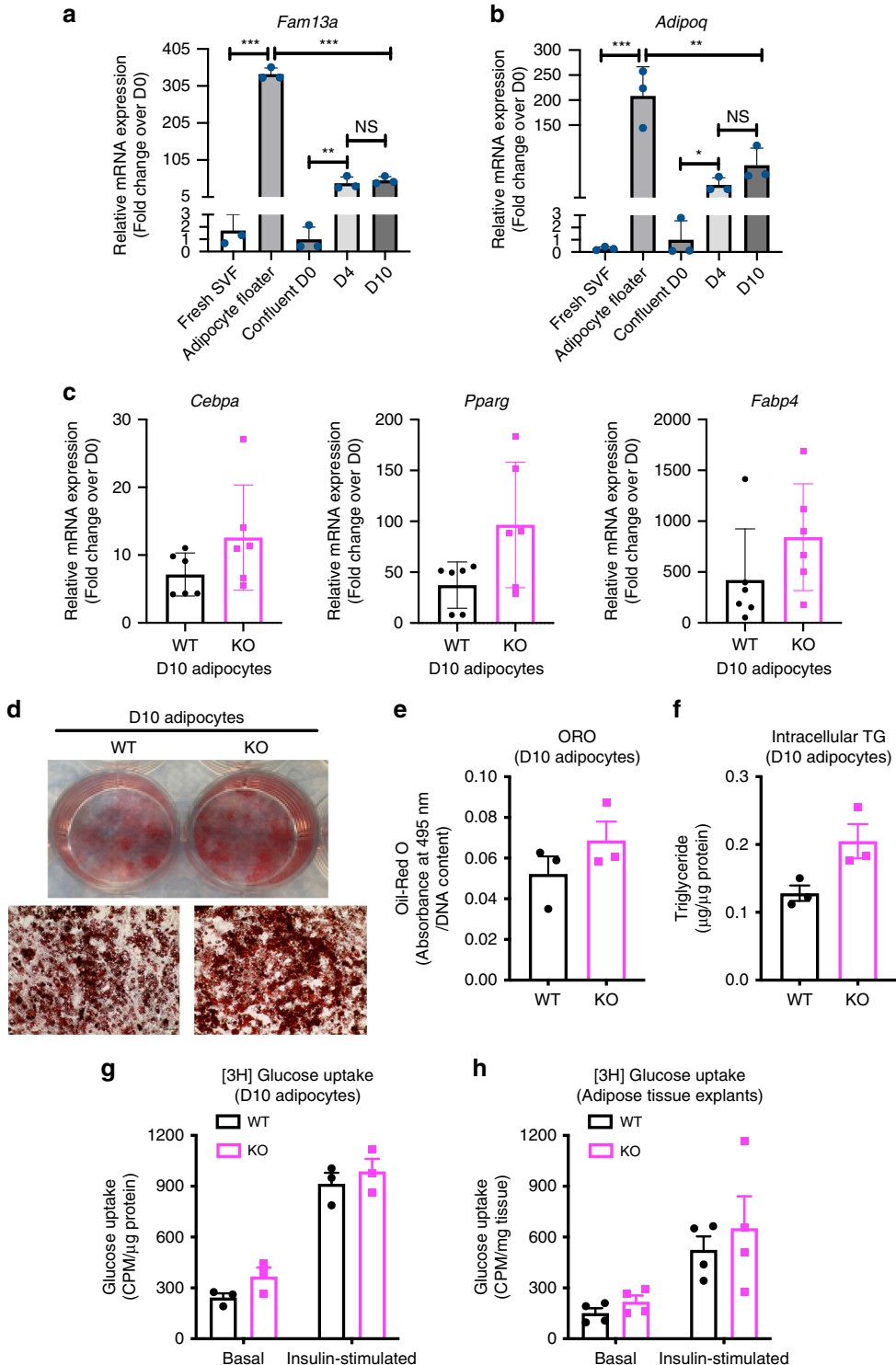

**Fig. 5 Effect of *Fam13a* on adipocyte differentiation of mouse SVFs isolated from SAT. a, b** *Fam13a* (**a**) and *Adipoq* (**b**) mRNA expression, quantified by qRT-PCR, in freshly isolated SVF, primary mature adipocytes (floater), or during in vitro adipogenesis of cultured SVFs (D0, D4, and D10). (8-week-old male WT mice, $n = 3$ per group, $n = 3$ culture wells per animal) **c** mRNA expression of adipogenic markers (*Pparg, Cebpa, Fabp4*), measured by qRT-PCR during in vitro adipogenesis of cultured SVFs (D0, D4, D10). (8-week-old male WT or *Fam13a* KO mice when isolating SVFs, $n = 6$ per group, $n = 3$ culture wells per animal). **d**, **e** Oil-Red O staining of lipid droplets (**d**) and semi-quantification (**e**) in D10 differentiated adipocytes from SVFs. ($n = 6$ per group, pictures represent $n = 6$ independent experiments, $n = 3$ cultures/animal, ×10 magnification, scale bar=100 μm). **f** Intracellular triglyceride content in D10 differentiated adipocytes from SVFs. ($n = 6$ per group, $n = 3$ culture wells per animal). **g** Basal and insulin-stimulated [3H] glucose uptake in D10 differentiated adipocytes from SVFs. ($n = 3$ per group, $n = 3$ culture wells per animal). **h** Basal and insulin-stimulated [3H] glucose uptake in SAT explants. (8-week-old male mice, $n = 3$ per group, $n = 3$ ex vivo cultures per animal). Data are presented as mean ± SEM. *$p < 0.05$, **$p < 0.01$, ***$p < 0.001$, one-way ANOVA followed by Turkey's multiple comparison test. Source Data are provided as a Source Data file.

and observed a non-significant trend toward induction of both basal and insulin-mediated glucose uptake in *Fam13a*-deficient adipocytes (Fig. 5g). To elucidate whether this slight increase in glucose uptake is due to the presence of more differentiated adipocytes in the culture of *Fam13a* KO observed above (Fig. 5c–f), we also performed ex vivo glucose uptake directly on SAT explants isolated from WT and *Fam13a* KO mice, and observed a similar directional trend (Fig. 5h). These data suggest that adipocytes from *Fam13a* KO mice, either differentiated in vitro or natively present in adipose tissue, are functionally active in depositing glucose and responding to insulin.

**Effect of *Fam13a* KO on gene profiles of mouse adipose tissue.** To understand the effect of *Fam13a* KO on gene expression in mouse tissues, we performed RNA sequencing of the VAT, SAT, and liver transcriptomes in *Fam13a* KO and WT male and female mice on chow and HFD. In WT mice, *Fam13a* was down-regulated in all three tissues in response to HFD (Fig. 6a).

Next, we performed a differential expression (DE) analysis across SAT and VAT samples from male mice ($N = 32$), adjusted for known (diet, tissue depot, RNA quality parameters) and hidden covariates (6 uncorrelated PEER factors). We observed 122 DE genes (FDR < 10%) between KO and WT samples (combined VAT and SAT; Fig. 6b). The DE genes included several genes previously associated with fat cell biology, including *Klf14*[18,19] (Fold change 2.3), *Agpat2*[20,21] (FC 0.51), *Slc7a10*[22] (FC 0.65), *Vegfa*[23], *Celsr2*[24] (FC 2.14), and *Fgfr2*[25] (FC 0.49). Further, we observed an overrepresentation of genes in adipogenesis pathways among genes overexpressed in KO, corroborating evidence from our in vitro and in vivo data suggesting that *Fam13a* deficiency induces subcutaneous adipogenesis. We also observed overrepresentation of genes in a $NAD^+$ salvage pathway among underexpressed genes; this pathway has been discussed previously in SAT in the context of obesity and weight loss, and intracellular levels of $NAD^+$ have a role in maintaining the metabolic status of the cell[26]. In addition to adipogenesis and $NAD^+$ salvage pathways, we found an overrepresentation of

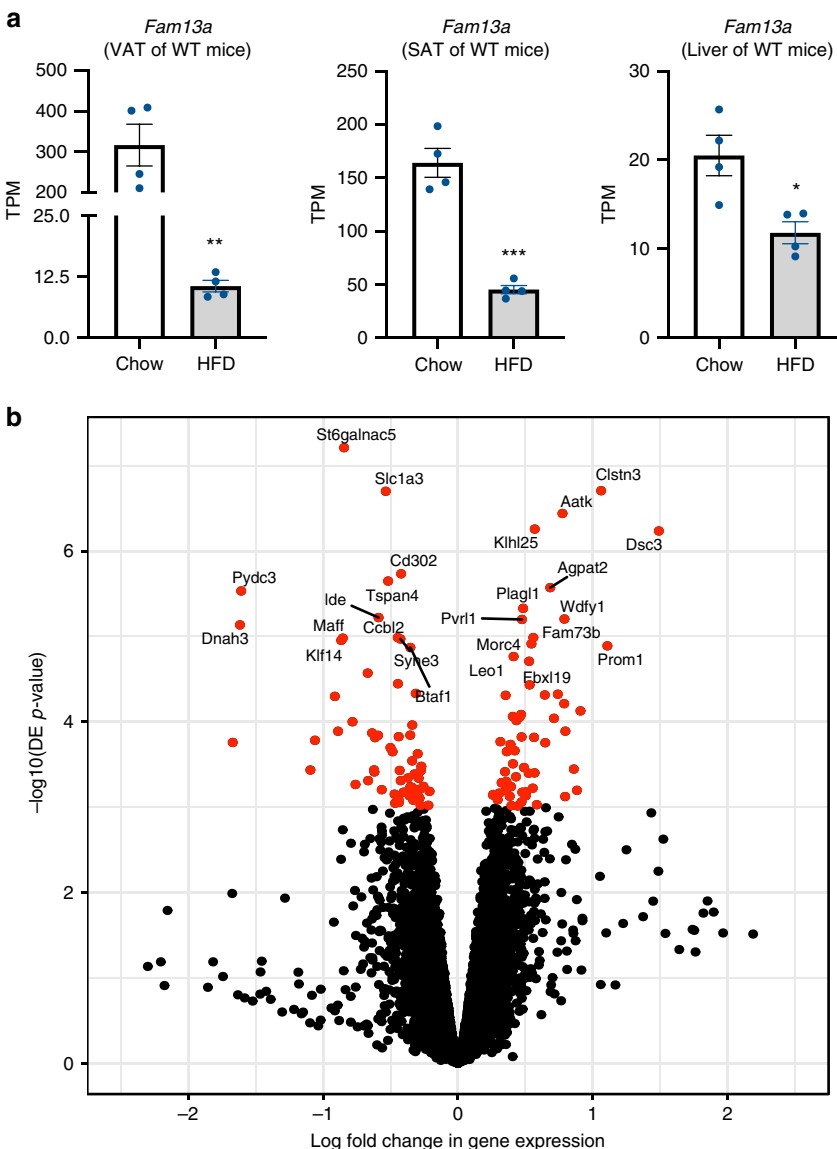

**Fig. 6 Transcriptome profiling of VAT and SAT in *Fam13a* KO and WT mice. a** *Fam13a* transcript per million (TPM) in VAT, SAT, and liver of male WT mice on chow and HFD. **b** The volcano plot showing the differentially expressed (DE) genes in VAT and SAT male *Fam13a* KO on chow and HFD. $n = 4$ per group. Data are presented as mean ± SEM. *$p < 0.05$, **$p < 0.01$, ***$p < 0.001$, unpaired $t$ test. Source Data are provided as a Source Data file.

overexpressed genes in fatty acid beta-oxidation and TCA cycle pathways in SAT from KO animals. There were no significant pathway enrichments in VAT. In a secondary DE analysis stratified by fat depot (SAT and VAT; including male and female samples [$N = 23$] and adjusting for sex in addition to the covariates listed above), we found a 13-fold difference in the number of DE genes in SAT compared to VAT ($n = 288$ in SAT, $n = 22$ in VAT, FDR < 10%), suggesting that *Fam13a* KO has a dramatically larger impact on the transcriptome in SAT.

The negative correlation of adipose *Fam13a* expression with obesity in a genetically obese and diabetic mouse dataset of BTBR[ob/ob] was demonstrated through a collaboration with Dr. Alan Attie's lab (Supplementary Fig. 9A). Additionally, in these mouse models, adipose *Fam13a* expression was negatively associated with proliferation of cells residing in adipose tissue, measured by the incorporation of deuterium into newly synthesized DNA of adipose tissue (Supplementary Fig. 9B). This negative correlation between *Fam13a* expression and adipose cell proliferation may be reflected by the quantity and phenotypic composition of adipose stromal cells. Within the phenotypic spectrum of SVFs, the percentage of CD45- non-immune cell subset, including adipose progenitor cells (APC, CD45- Sca1+ CD31-) and endothelial cells (EC, CD45- Sca1+ CD31+), increased in subcutaneous fat of *Fam13a* KO mice (Supplementary Fig. 9C–E). In addition, purified APCs from KO mice tended to differentiate better into adipocytes, as compared to WT (Supplementary Fig. 9F, G). These results suggest that *Fam13a* expression in SAT is negatively related to the number of adipose precursors as well as the adipogenic ability of these cells, which is consistent with our previous result that the crude SVFs of *Fam13a*-deficient mice exhibit higher adipogenic potential (Fig. 5).

## Discussion

Our human genetic data coupled with in vivo and in vitro experiments provide evidence that *FAM13A* regulates fat distribution and metabolic traits through its action on adipose tissue. This selective role of *FAM13A* is supported by the SAT-specific *FAM13A* eQTLs that are associated with a metabolically obese normal weight phenotype and by the correlation of SAT *FAM13A* expression with a range of insulin resistance-related phenotypes. The inverse relationship between SAT *FAM13A* gene expression levels and metabolic health is further supported by both in vivo and in vitro experimental evidence. In WT mice, endogenous *Fam13a* expression in adipose tissue decreased upon HFD treatment. Male *Fam13a* KO mice showed a reduction in VAT/SAT ratio despite increasing more in weight during HFD treatment. Finally, *Fam13a*-deficient SVFs derived from SAT demonstrated a tendency towards increased ability to generate new adipocytes de novo, which may help meet the excess lipid-storage needs. Together, our experimental data are consistent with the interpretation suggested by GWAS data: higher *FAM13A* expression in adipose tissue may have a negative impact on body fat distribution.

The anatomic location of excess body fat has an important impact on metabolic health as higher VAT/SAT ratio- a measurement of body fat distribution- is associated with higher cardiometabolic risks independent of absolute fat volumes[27]. A high VAT/SAT ratio can also result from low SAT adiposity, and an inability to appropriately expand SAT is one of the key factors that links excess caloric intake to insulin resistance[28]. Both genetically modified mouse models (e.g. adipose-specific overexpression of GLUT4[29] or mitoNEET[30]) and pharmacological models (e.g. treatment of the PPARg agonist, thiazolidinedione[31]) that exhibit a healthy obese phenotype tend

to display preferential expansion of SAT with improved insulin sensitivity. In our study, *Fam13a* KO mice showed a reduction in VAT/SAT ratio after HFD challenge as compared to WT mice. The reduced VAT mass and increased adipogenesis program in SAT may simultaneously downregulate the ratio of VAT/SAT in *Fam13a* KO mice.

The biological pathways underlying the differential regulation of the two fat depots were revealed by analyses of differentially expressed genes and pathways. SAT of *Fam13a* KO mice showed a coordinated overexpression of genes regulating adipogenesis and genes involved in mitochondrial function, such as beta oxidation, the TCA cycle, electron transport and oxidative respiration. SAT is known to be able to generate beige adipocytes, a mitochondria-rich and fat-burning adipocyte, while VAT is a classic depot for white adipocytes with a low number of mitochondria and limited ability for adipogenesis[32]. SGBS isolated from human SAT, as well as SVFs isolated from mouse SAT, showed an increased tendency to differentiate into beige adipocytes upon *FAM13A* knockdown (Supplementary Figs. 3A and 8). *Fam13a* depletion in SAT may trigger a more dramatic change than in VAT by generating more beige adipocytes with improved mitochondrial efficiency. The increased mitochondrial metabolism, combined with induced generation of healthy adipocytes, preserves insulin sensitivity, despite obesity. In-depth metabolic and thermogenic functional studies at various stages of adipocyte development are warranted to fully reveal the involvement of *FAM13A* on white/beige adipocyte fate determination and metabolism.

The biological function of *FAM13A* protein and the mechanisms through which *FAM13A* mediates adipogenesis remain poorly understood. Based on the amino acid sequence homology, *FAM13A* contains RhoGAP domain, an evolutionary conserved protein domain of GTPase activating proteins for small GTPases (Rho/Rac/Cdc42-like)[33]. Data from others suggests that Rho GTPase signaling regulates adipogenesis[34]. Additionally, WNT/β-catenin signaling is known to inhibit adipogenesis[35]. Tang et al. suggests a potential role of *FAM13A* in regulating adipose precursor number, potentially via WNT/β-catenin signaling[36]. Future studies are necessary to uncover the mechanistic link between *FAM13A* and adipogenesis.

Consistent with our data, additional studies have indicated that adipose tissue is the primary site for *FAM13A* action mediating downstream metabolic functions. Wardhana et al.[37] and Tang et al.[36] showed a negative correlation of adipose *Fam13a* expression with diet-induced obesity in mice, in agreement with our data. Lundbäck et al.[38] and Tang et al.[36] noted *FAM13A* as a negative regulator of adipogenesis by using different immortalized cell models and genetic manipulation approaches: Lundbäck et al.[38] showed that siRNA knockdown of *FAM13A* during a specific differentiation window of human mesenchymal stem cells (from day 4 to day 7) induced genes indicative of adipogenesis (*CEBPA*, *PPARG*)[38], while Tang et al.[36] showed that overexpression of *Fam13a* in mouse 3T3-L1 cells led to apoptosis of preadipocyte and diminished its conversion to adipocytes. Regarding whole-body metabolism, Wardhana et al.[37]. showed no difference in food intake, oxygen consumption, respiratory exchange ratio, physical activity and body temperature (under room temperature) between WT and *Fam13a*-overpression mice after 14-week HFD challenge. Wardhana et al.[37] observed a modest but significant insulin resistance and glucose intolerance in *Fam13a* KO mice, while we and Tang et al.[36] detected no difference in either insulin sensitivity or glucose tolerance between WT and *Fam13a* KO mice under both chow and HFD, despite a slightly increased adiposity in *Fam13a* KO mice after HFD. These discrepancies might be explained by variations in experimental conditions such as the sex of the mice used for

specific experiments, the content of fat and sugar in diets, and the fat percentage achieved after different lengths of HFD challenges started at various ages of animals. More inclusive studies based on consistent experimental conditions are required to fully evaluate the metabolic consequence of *Fam13a* deficiency. While our findings and other studies give support to the notion that adipose tissue mediates *FAM13A* action, the metabolic phenotypes of whole-body *Fam13a* KO mice are subtle. Adipose-specific knockouts of *Fam13a* would be useful to study the role of *Fam13a* specifically in adipose tissue and its causal impact on systemic metabolism.

Sex dimorphism in the association of *FAM13A* with body fat distribution is observed in both humans and mice. In human GWAS, the WHRadjBMI locus harboring *FAM13A* variants showed a stronger effect in women than in men. Sex dimorphism in the genetic regulation of fat distribution may be related to the difference in body fat shape between the sexes[39]. In mouse models, while in male mice there was a difference in VAT/SAT between WT and *Fam13a* KO mice upon HFD, no significant phenotypic effects of *Fam13a* deficiency were observed in female mice. Although current translational investigation on fat distribution and its metabolic consequences relies on rodent models, they may not faithfully mimic all aspects of human fat biology due to differences in anatomical location and function of fat depots between species[39].

Our current model in humans is that *FAM13A* negatively regulates adipocyte development. Carriers of the *FAM13A* susceptibility alleles associated with higher WHR and fasting insulin levels express higher levels of *FAM13A* in SAT, despite having a lower overall body fat compared to noncarriers. In noncarriers, in response to caloric excess (such as HFD) there is systemic demand for fat storage and downregulation of *FAM13A* in SAT may stimulate preadipocytes of subcutaneous origin to differentiate into new and healthy adipocytes for improved fat storage. In carriers of *FAM13A* risk alleles, *FAM13A* expression in SAT remains high after calorie challenge which may limit the healthy expansion of SAT compartment through de novo adipogenesis, thus diverting lipid deposition into VAT. Therefore, we propose that *FAM13A* alleles associated with increased SAT *FAM13A* expression may predispose an individual to a damaging body fat distribution pattern. This pattern has been termed a "lipodystrophy-like" pattern in the literature[4,5].

In conclusion, our work suggests a novel role of *FAM13A* in adipocyte differentiation and function. We have provided evidence that the presence of *FAM13A* risk alleles is correlated with higher *FAM13A* expression in SAT that, in turn, is associated with unfavorable body fat distribution. Future studies to understand the molecular mechanism of action of *FAM13A* will be important to gain insights on how fat is distributed in health and cardiometabolic diseases.

## Methods

**UK Biobank and Gene ATLAS**. The UK Biobank is a prospective cohort study of approximately 502,000 individuals conducted between 2006 and 2010. The data includes genotype and detailed phenotype information; the study was approved by the North West Multi-Center Research Ethic Committee, and all participants provided written consent.

**Fine-mapping and colocalization**. We characterized the role of *FAM13A* across tissues in humans by integrating IR-related trait GWAS data with tissue-specific gene expression from the Genotype Tissue Expression project (v7). We first performed Bayesian fine-mapping using the FINEMAP algorithm to determine posterior probabilities of each variant in the locus driving the association signal. We then used the top-variant ranked by posterior probability for further analyses. We performed colocalization analysis of *FAM13A* gene expression in subcutaneous adipose tissue (SAT), visceral adipose tissue (VAT), liver and skeletal muscle with seven traits using eCAVIAR. The seven traits included were WHRadjBMI, trigyceride levels, HDL, FI, FIadjBMI, fasting glucose, and BMI. We modified the

colocalization posterior probability (CLPP) computed in eCAVIAR to consider a locus as colocalized if the GWAS and eQTL predicted causal SNPs are in strong LD. The modified CLPP (mCLPP) was computed as

$$mCLPP = \sum_{i,j < N} g_i e_j LD_{ij}$$

where $g_i$ and $e_j$ are the probabilities that the ith and jth SNPs are causal GWAS and eQTL SNPs respectively, and $LD_{ij}$ is the LD $R^2$ between the two SNPs.

**Enrichment for epigenomic annotations**. We performed a statistical enrichment analysis for epigenomic marks in which we defined a 99% credible set consisting of 16 linked variants at the *FAM13A* locus that are most likely to be responsible for driving the association at this locus. We used FINEMAP posterior probabilities to define this set – variants were included as long as their cumulative fine-mapping posterior probability for the SAT eQTL signal was ≤0.99. We used permutations ($n = 1000$) to estimate enrichment using the mean finemapping posterior probability of variants that overlapped an annotation as a test statistic. We shifted the annotations randomly by a maximum of 500 kb for each permutation and estimated an enrichment p-value using the empirical distribution.

**PheWAS in UK Biobank**. We performed a phenome-wide association study (PheWAS) to study the phenotypic effects of the FAM13A eQTLs in 337,536 individuals of the UK Biobank. Instead of using the leading finemapped variant rs9991328, we used rs1377290 (LD $R^2 = 1.0$) as a proxy. We used a linear model adjusted for age, sex, genotyping array and 10 principal components to study the association of rs1377290 with 278 predefined phenotypes including hospitalization and procedure codes, self-reported variables, and physical and functional measurements[40]. We performed additional lookups of phenotypic associations using the Gene ATLAS (http://geneatlas.roslin.ed.ac.uk/).

**METSIM and STAGE**. THe METabolic Syndrome In Men (METSIM) Study is a resource for studies of metabolic and cardiovascular diseases[14]. For the METSIM expression arrays, there were 770 individuals in total. There were 1400 biopsies taken, although only 770 subjected to the expression arrays. These were all men collected from Kupio Finland, age 45–73 (http://www.nationalbiobanks.fi/index.php/studies2/10-metsim)[14]. The Stockholm Atherosclerosis Gene Expression (STAGE) Study is a multi-organ expression profiling (gene network across seven vascular and metabolic tissues) to uncover a gene module in coronary artery disease gene[16,41]. STAGE consisted of 105 individuals (both genders) collected from Sweden and Estonia, age 58–74. In this study, 72 individuals were subjected to expression arrays using the HuRSTA-2a520709 Affy platform[16]. Human gene-by-gene and gene-by trait correlations were performed on adipose tissue expression arrays data from the METSIM study[13,19], as well as adipose and liver expression arrays within the STAGE study[16,41]. These individual X gene expression matrices were used for correlations, which were performed using the WGCNA package[42] in R and corrected using a 5% FDR.

**T2D Knowledge Portal**. This portal enabled us to browse and analysis of human genetic information linked to *FAM13A* locus. (Accelerating Medicines Partnership, AMP). http://www.type2diabetesgenetics.org/.

**Animal care and experiments**. All animal studies were reviewed and approved prior to commencement of the activity by the Administrative Panel on Laboratory Animal Care (APLAC) at Stanford University. We obtained *Fam13a* mice, originally created from ES cells of KOMP Repository in which exon 5 of the *Fam13a* gene was flanked by LoxP sites, as a contribution from Jin's lab[43] (Department of Comparative Biosciences, University of Illinois at Urbana–Champaign, Urbana, IL). Control (WT) and KO mice were developed on *C57BL6* background.

*Fam13a* KO mice were housed at Stanford Animal Facility, with a 12-h light/dark cycle, in a pathogen-free facility. Mice (control and KO) were fed with either standard rodent chow diet (Teklad Calories from Protein 24%, from Fat 18%, from Carbohydrate 58%) or high-fat diet (HFD, Research Diets, Inc., D12451, Protein 20% kcal, Fat 45% kcal, carbohydrate 35% kcal), plus water ad libitum. They were fed on HFD beginning at 8-weeks-old for 14 weeks until they were killed at 22-weeks-old. Adipose tissue from two depots, gonadal fat (equivalent to VAT) and inguinal fat (equivalent to SAT), was harvested and weighed. Blood was also collected. Harvested tissue was either snap frozen and stored in −80 °C for RNA and protein extraction or fixed in 4% PFA for staining and immunohistochemistry evaluation.

**Hematoxylin and Eosin staining**. Mice were overnight fasted, bled and weighed before sacrificing. VAT, SAT, and liver were harvested, weighed, and fixed in 10% Neutral Buffered Formalin (Thermo Scientific). Tissues were left in fixative for 24 h and then dehydrated in 70% ethanol. Dehydrated tissue underwent standard H&E statning[44] in the Stanford Animal Histology facility, as follows: Xylene x3, 2 min each; 100% Ethanol 2 min; 100% Ethanol 1 min; 95% Ethanol x2, 1 min each; 80% Ethanol, 1 min; Running tap water, 1 min; Harris Hematoxylin, pH 2.5, 10 min; Running tap water, 1 min; Clarifier (1% HCl in 70% EtOH) 20 s; Running tap

water, 5 min; Bluing reagent (~0.5% Ammonium Hydroxide in diH2O); Running tap water, 3 min; 95% Ethanol 1 min; Eosin, 95% Ethanol solution, pH 4.6, 2 min; 95% Ethanol x3, 30 sec each; 100% Ethanol x3, 2 min each; Xylene x3, 2 min each; Cover slipped with Cytoseal XYL.

**Quantification of adipocyte size and number.** Briefly, microphotographs of H&E stained adipose tissue blocks were acquired from an optical microscope (Zeiss Axioplan2) at 20x magnification and images were captured with a Leica DC500 camera and a NIS Elements software. ImageJ software with the H&E color deconvolution plugin was used to determine the size of the adipocytes and the number of adipocytes within a certain diameter range by two individuals in a double blinded manner. Four fields of view for SAT slides and six fields of view for VAT slides were quantified. SAT and VAT from seven animals per group were analyzed. A total of 2000–2200 cell diameters were measured by Adiposoft, an automated software for analysis of adipose tissue cellularity[45]. The quantification unit of pixels were converted to microns by Photoshop CS6 (1 pixel = 0.3953 μm at ×20 objective). Adipocyte number per fat depot was calculated by dividing the total tissue weight by the mean adipocyte weight[46]. Total tissue weight was measured directly after tissue dissection, and the mean adipocyte weight was calculated by multiplying the mean adipocyte volume by the triglyceride density (0.915 g L$^{-1}$). The mean volume of adipocyte was calculated from the diameters measured above by applying the following formula: $V = 4/3\pi (D/2)^3$.

**Glucose and insulin tolerance tests.** Glucose and insulin tolerance tests were performed on 3 to 4-month-old mice[47]. For GTTs, mice were fasted overnight, and 2 g glucose kg$^{-1}$ body weight were injected through i.p. For ITTs, mice were fasted for 6-h and insulin at concentration of 1.0 U kg$^{-1}$ body weight were i.p. injected. From the tail vein of conscious animals, blood was collected before i.p. injection, termed as 0 min, and at 15, 30, 60, and 120 min after injection of either glucose (GTT) or insulin (ITT). Blood glucose was measured using a glucometer (TRUEbalance, Nipro Diagnostics, Inc.).

**Lipid panel tests.** Mice were fasted overnight. Blood was collected by retro-orbital bleeding from anesthetized mice, using Microhematocrit Capillary Tube (Fisherbrand™, Heparinized). After expelled immediately into collection tubes, blood samples were centrifuged at 6000 rpm for 10 min at 4 °C to collect plasma. Lipids (includes TG, LDL, HDL, and cholesterol) were measured in plasma samples at the Stanford Animal Diagnostic Laboratory in the Veterinary Service Center using standard protocols and instruments. Insulin was measured by using Ultra-Sensitive Mouse Insulin ELISA Kit (CrystalChem), and Free Fatty Acid was measured by using Colorimetric Non-esterified Fatty Acid Kit (Wako).

**RNA-Seq.** RNA was extracted from ~50 mg of liver, visceral and subcutaneous adipose tissues using RNeasy Lipid Tissue Mini Kit (Qiagen). Library preparation and RNA sequencing were carried out by using the Illumina TruSeq Stranded protocol, with an average number of reads per sample as 40 million. Reads were aligned to the Grcm38 mouse genome using STAR 2.6 software[48], and Gencode M11 mouse gene annotations and counts were obtained using the quantMode option in STAR. Differential expression analysis was performed using edgeR 3.20.9[49] with a false discovery rate (FDR) of 10%. Analysis of the transcriptome-wide effects of *Fam13a* KO was performed in adipose tissue samples from both SAT and VAT fat depots of male mice, adjusted for known (diet, fat depot, RIN, number of days from tissue harvest to RNA extraction) and hidden covariates (6 PEER factors).

**F2 ob/ob mice.** F2 cohort database derives from B6:BTBR F2 mice that were all killed at the same age (10 weeks) and were all obese, thereby focusing the study on genetic differences between the parental strains[50]. Proliferation of cells in adipose tissue was measured by the incorporation of deuterium into newly synthesized DNA, as described previously (http://diabetes.wisc.edu/index.php).

**Isolation and culture of SVFs from adipose tissue.** Isolation of SVF and adipocyte fraction from adipose tissue, as well as adipogenic differentiation of SVFs were performed as described below[51]. In brief, adipose tissue was minced into small pieces and digested with 0.2% type I collagenase (Life Technologies #17100) containing 4% BSA for 45 min at 37 °C under continuous shaking at 115 rpm. Cell suspension was then passed through a gauze, span at 200 × g for 5 min. Floating adipocyte fraction was collected for RNA extraction, whereas cell pellet was resuspended in ACK lysing buffer (Thermo Scientific #A1049201) for 10 min at room temperature before adding complete growth medium (DMEM/F12 with 10% FBS and 1% penicillin-streptomycin) to stop the reaction. The resulting suspension was filtered through 100μm strainers. After centrifuging at 200 × g for 5 min, the pellets were resuspended, with one small aliquot saved for RNA extraction and the rest of the cells cultured for adipogenic differentiation. In total, 5–7 days post-seeding, confluent SVF cells were subjected to adipogenesis by using 0.5 mM IBMX, 1 μM dexamethasone, 10 μg ml$^{-1}$ insulin and 2.5 μM rosiglitazone for 2 days, followed by supplementation of 10 μg ml$^{-1}$ insulin alone for an additional 8 days.

**Culture and differentiation of human SGBS preadipocytes.** Human Simpson-Golabi-Behmel syndrome (SGBS) preadipocytes[17] (a gift from Dr. Martine Wabitsch, University of Ulm) were cultured in DMEM/F12 supplemented with 10% fetal bovine serum, 8 μg ml$^{-1}$ biotin, 4 μg ml$^{-1}$ pantothenic acid and antibiotics (100 U mL$^{-1}$ penicillin and 0.1 mg mL$^{-1}$ streptomycin). To initiate adipogenic differentiation of SGBS preadipocytes, serum-free DMEM/F12 was supplemented with 8 μg ml$^{-1}$ biotin, 4 μg ml$^{-1}$ pantothenic acid, 10 μg mL$^{-1}$ transferrin, 20 nM insulin, 100 nM cortisol, 0.2 nM triiodothyronine (T3), 25 nM dexamethasone, 250 μM IBMX and 2 μM rosiglitazone for 4 days. Cells were thereafter incubated in serum-free medium containing all the components above except dexamethasone, IBMX, and rosiglitazone[52]. Adipocyte functional assays, including glucose uptake and lipolysis, were tested in differentiated SGBS adipocyte on Day 12.

**CRISPRi knockdown of *FAM13A* in SGBS preadipocytes.** For the dCas9-KRAB expressing construct, we used pHR-SFFV-KRAB-dCas9-P2A-mCherry (Addgene #60954) which expressed an N-terminal KRAB-dCas9 fusion protein and mCherry. For the sgRNA expression vector, we used a lentiviral U6 based system (pU6-sgRNA EF1alpha-puro-T2A-GFP, a gift from Jonathan Weissman) which coexpressed GFP and a puromycin-resistance cassette separated by a T2A sequence from the EF1alpha promotor. Three sgRNA sequences targeting *FAM13A* and one scramble sgRNA control sequence were selected from the human CRISPRi v2 (hCRISPRi-v2) library[53] and cloned into the lentiviral U6 based expression vector backbone[54]. Final individually cloned gRNA constructs were transformed into chemically competent cells (NEB #C2987) and extracted by a QIAprep Spin Miniprep Kit (QIAGEN #27106) for downstream lentivirus production. The sgRNA sequences of all plasmids were confirmed by Sanger sequencing: Scrambled sgRNA: GCTGCATGGGGGCGCGAATCA; sg*FAM13A*_1: GACGCTTTCTGAGAGAATGG; sg*FAM13A*_2: GTCCCAATGCAAAGGCCCCA; sg*FAM13A*_3: GTCCGCTGAACCCACATGGC.

Lentivirus was produced in HEK293T cells (ATCC: CRL-3216™) maintained in DMEM with 10% FBS, 1% Pen-Strep and 2 mM GlutaMAX™ (Invitrogen). Cells were seeded at 4 million per 10 cm dish, and transfected the following day with either the dCas9-KRAB vector (9 μg) or the sgRNA vector (9 μg) along with the envelope vector pMD2.G (1 μg, Addgene #12258) and the packaging vector pCMV-dR8.91 (8 μg, a gift from Jonathan Weissman) using Lipofectamine 3000 (Invitrogen). In all, 6–14 h after transfection, culture medium was refreshed. For the lentivirus generation of dCas9-KRAB, 1:500 volume of ViralBoost™ Reagent (ALSTEM #VB100) was supplemented into one volume of the fresh culture medium. In all, 48 h later, supernatant was collected and filtered through 0.45 μm filters. sgRNA virus was frozen immediately in −80 °C, while dCas9-KRAB virus was concentrated by 10-fold using Lentivirus Precipitation Solution (ALSTEM #VC100) before frozen.

To construct CRISPRi knockdown of *FAM13A*, SGBS preadipocytes at 70% confluence were co-transduced by concentrated dCas9-KRAB virus (1 volume) and unconcentrated gRNA virus (2 volume), along with SGBS culture medium (3 volume) supplemented with polybrene (8 μg ml$^{-1}$ in total mixture). In all, 72 h later, cells were harvested and sorted by flow cytometry using a BD Influx™ Sorter. Briefly, cells were first selected by size, on the basis of forward scatter (FSC) and side scatter (SSC). Cells were then gated on both FSC and SSC singlets to ensure that individual cells were analyzed. Wild-type SGBS preadipocytes were used as the negative control to determine background fluorescence levels. Collection gate was set on double-positive population of mCherry (dCas9-KRAB expression) and GFP (gRNA expression) cells. Sorted cells were expended in culture, grown into confluence before induction of adipogenic differentiation.

**Flow cytometry and fluorescence-activated cell sorting.** Antibodies used include CD45-FITC (eBioscience 11-0451-81), CD31-PE-Cy7 (eBioscience 25-0311-82) and Sca1-APC (eBioscience 10811). Freshly isolated SVFs were resuspended in FluoroBrite™ DMEM media (Thermo Fisher A1896701) supplemented with 2% FBS at 10 million cells per mL for antibody staining, followed by FACS. Dilution factors for the antibodies were: 1:200 for CD45-FITC, 1:100 for CD31-PE-Cy7, and 1:100 for Sca1-APC. Samples were analyzed and sorted on a BD Influx™ Sorter. Cell gating was based on the unstained controls. Live and single cells were separated by forward scatter (FSC) and side scatter analysis. Cells were sorted into Trizol for gene expression analysis or into complete media for cell culture. FACS files were analyzed using FlowJo v.9.

**Glucose uptake.** Mouse SVFs and human SGBS preadipocytes were grown and differentiated in 24-well plates, and starved in serum-free and glucose-free medium for 3 h prior to the assay. After washing 2 times with pre-warmed KRH buffer [50 mM HEPES, 137 mM NaCl, 4.7 mM kCl, 1.85 mM CaCl$_2$, 1.3 mM MgSO$_4$, pH = 7.4], cells were treated with or without insulin (100 nM) in KRH buffer for 30 min, followed by the addition of 1 μCi ml$^{-1}$ [$^3$H] 2-Deoxy-D-Glucose (PerkinElmer) and 100 μM 2-Deoxy-D-Glucose (Cayman) for 15 min. Reactions were terminated by washing the cells with ice-cold KRH buffer for three times. Cells were then lysed by M-PER™ Mammalian Protein Extraction Reagent (Thermo Scientific), and the incorporated radioactivity was determined by liquid scintillation counting. A small aliquot of each cell lysate was used for protein quantification and normalization of radioactivity counting. For ex vivo glucose uptake on adipose

tissue explants, ~20 mg tissues were weighed and cultured in one of the 24-well. After following the same procedure as described above, tissues were digested with 0.5 M NaOH overnight. Radioactivity measurements on the tissue lysates were normalized against the tissue weight per well.

**Lipolysis**. Glycerol release into the culture medium was used as an index of lipolysis. Differentiated SGBS adipocytes were starved in DMEM/F12 containing 2% fatty acid-free BSA for 3 h prior to the assay. Then, the same medium was refreshed with the addition of either vehicle (DMSO) or isoproterenol (ISO, 100 nM). After 3 h of incubation, 100 μl culture medium was transferred to a 96-well, and 100 μl of Free Glycerol Reagent (Sigma-Aldrich #F6428) was added to each well. Plates were incubated at room temperature for 15 min and the absorbance was read at 540 nm. Glycerol concentration was corrected by the protein content of the cells in the same well.

**Triglyceride quantification assay**. Intracellular triglycerides in differentiated adipocytes were extracted and homogenized in 5% NP-40. Cell suspension was centrifuged at $10,000 \times g$ for 10 min at 4 °C. Supernatant was then diluted by 1:3 and triglyceride concentration at 550 nm absorbance was measured by using the Triglyceride Colorimetric Assay Kit (Cayman #10010303). Triglyceride levels were presented by normalizing against the protein content.

**Oil Red O staining and semi-quantification**. Differentiated adipocytes were fixed with 10% formalin for at least 1 h. The cells were then rinsed with 60% isopropanol for 5 min followed by the 15-min incubation with freshly prepared ORO working solution (composed of 4 parts water and 6 parts 0.35% ORO dye in isopropanol). After four times washing with water, images were acquired under the light microscope. ORO was then extracted from fixed cells with 100% isopropanol, followed by OD measurement at the absorbance of 500 nm. DNA content of the fixed cells was extracted by QIAamp DNA FFPE Kit (Qiagen #56404). ORO content was presented by normalizing against the DNA content.

**Reverse transcription and qPCR analysis**. Total RNA was extracted using TRIzol reagent (Invitrogen). RNA was converted to cDNA using High-Capacity cDNA Reverse Transcription Kit (Applied Biosystems). Quantitative PCR reactions were prepared with TaqMan™ Fast Advanced Master Mix (Thermo Fisher Scientific) and performed on ViiA 7 Real-Time PCR System (Thermo Fisher Scientific). Data were normalized to the content of Cyclophilin A (PPIA), as the endogenous control. TaqMan primers obtained from Thermo Fisher Scientific were: *LEP* (Hs00174877_m1), *PPARG* (Hs01115513_m1), *CEBPA* (Hs00269972), *UCP1* (Hs01084772), *PPIA* (Hs04194521_s1), *Adipoq* (Mm00456425_m1), *Pparg* (Mm00440940_m1), *Cebpa* (Mm00514283_s1), *Fabp4* (Mm00445878_m1), *Ppia* (Mm02342430_g1); TaqMan primers obtained from IDT were: *FAM13A* (HS.PT.58.3237208), *ADIPOQ* (Hs.PT.58.39189358), and *PGC1A* (Hs.PT.58.14965839), *Fam13a* (Mm.PT.56a.7855516).

**Gene enrichment and pathway analysis**. The GO and pathway analysis of human and mice expression data were assessed by ConsensusPathDB-human and ConsensusPathDB-mouse, which are fully accessible online (http://cpdb.molgen.mpg.de/CPDB.[15]).

**Statistical analysis**. For comparison of plasma biochemicals, body and tissue weights of WT and KO mice, two-tailed parametric T-test were applied using GraphPad Prism version 7.00 (GraphPad Software, La Jolla California USA). A p value of <0.05 (*), <0.01 (**) and <0.001 (***) were considered significant and asterisked in the relevant plots. Between-group differences in distribution curves for adipocyte diameter were analyzed using the Kolmogorov–Smirnov test, which takes into account the ordering (i.e., cell diameter) of the categories and is therefore a suitable test for analysis of curve shifts.

The analysis of expression data in METSIM follows previously published results[14,19]. Statistical methods for analysis of data from the STAGE cohort have already been published[16,41]. Expression data were considered significant with 5% FDR. To show adjusted relationships between *FAM13A* expression adjusted for BMI and clinical traits, we implemented the following: A linear model was constructed between normalized probe levels across the population and BMI (using the base R function lm), and extracted residuals were correlated against indicated clinical traits. Biweight midcorrelation and corresponding student p values for this relationship were calculated using the R package WGCNA.

**Reporting summary**. Further information on research design is available in the Nature Research Reporting Summary linked to this article.

## Data availability

RNA-Seq data have been deposited to the National Center for Biotechnology Information Gene Expression Omnibus and are available at GSE137022. All the data related to METSIM and STAGE cohorts are available at GSE70353, PMID 28119442, PMID 28573393, PMID 28257690, PMID 19997623 and URL http://www.

nationalbiobanks.fi/index.php/studies2/10-metsim. The GWAS data for bodyfat %, FIadjBMI and WHRadjBMI are publicly available at https://www.ebi.ac.uk/gwas/. For the eQTL tracks, the variant level associations are available at https://gtexportal.org/home/datasets. PheWAS data were obtained from UK Biobank dataset and is available through an application process at https://www.ukbiobank.ac.uk/register-apply/. Data for the epigenetic annotations is publicly available and can be downloaded at GSM916066, GSM906397, and GSM906395. Source Data underlying Figs. 3–6 and Supplementary Figs. 2–9 are provided as a Source Data file. All other data are included in the article file and its supplementary information or available upon request.

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

## Acknowledgements

This study was supported by grants from the National Institute of Diabetes and Digestive and Kidney Diseases (1R01DK107437 and 1R01DK106236), the Stanford Diabetes Research Center (NIDDK award P30DK116074), and an unrestricted grant from Merck. We would like to thank Casimiro Castillejo-Lopez and Ewa Bielczyk-Maczyńska for constructing plasmids and sharing experimental protocols, respectively.

## Author contributions

T.Q., E.I., and J.W.K. conceived of and designed the study. M.F., J.L., N.C., and I.C. planned and performed experiments. M.F., J.L., A.R., M.S., and M.G. analyzed data. X.Z. and P.S. contributed with experimental and technical expertize. M.K., A.A., J.Y., M.W., A.J.L., M.K.S., I.C., C.M.M., T.M., and G.R. provided data or material support. S.B.M. and D.R. participated to supervision and interpretation of data. M.F., J.L., A.R., E.I., and J.W.K. drafted and revised the manuscript. All authors read and approved the manuscript.

## Competing interests

At the time the work was performed, Myung Kyun Shin, Cliona M Molony and Dermot Reilly were Merck employees.
