## [Peer Review File · Nature Communications]

Reviewers' Comments:

Reviewer #1:

Remarks to the Author:

The human lipodystrophy-like gene FAM13A affects fat distribution and function

General comments:

In this paper Fathzadeh et al. provide evidence that FAM13A plays a role in fat distribution in humans. Specifically, the paper addresses two important issues:

1- they provide evidence that FAM13A is indeed the gene responsible for the GWAS signal for the insulin resistant trait related variants reported by Scott et al. and Willer et al. This is important as there is a deplorable lack of GWAS hits for which the causal gene has been identified. The expression analyses that have been performed by the authors are state-of-the art and provide coherent evidence that points to FAM13A as the most likely gene responsible for the GWAS signal; 2- they show that the functional consequences of FAM13A are most likely associated with a difference in the distribution of sub-cutaneous to visceral fat depts. This is an important finding. It has long been hypothesized that individuals that have a more flexible subcutaneous fat tissue (i.e. more expandable under over-feeding conditions) are somewhat protected from the detrimental metabolic effects and vice versa that the accumulation of visceral fat, even in the absence of excess BMI (which is mainly accounted for by subcutaneous fat accumulation) is associated with insulin resistance and risk of CAD. However, the mechanisms for this difference in distribution are unknown. Thus, this paper proposes for the first time a possible biological mechanism for this hypothesis.

The methods used to provide evidence for the two topics are adequate and the analyses and experiments are well designed overall. The authors have gone to great lengths to provide evidence based on in silico methods, cell based assays, human cohort studies and mouse ko experiments. This is commendable, especially the inclusion of human cohorts to show an association between the FAM13A SAT expression and the primary hypothesis i.e. that this expression is associated with differences in SAT/VAT fat distribution or ratio. Thus this paper provides data that may be very important in our understanding of differential fat deposition in humans.

Major comments:

1- Human cohort studies: To really show that FAM13A impacts fat distribution in humans these analyses should ideally be done in a cohort that would allow to assess to directly study the association between FM13A expression and body fat distribution using gold standard methods for body composition i.e. by DEXA. The two cohorts studied in the article only provide crude proxies for body fat distribution like waist/hip ratio. There are some well-studied cohorts that have access to both SAT expression data as well as DEXA data (e.g. DiOGenes) that could be used. The authors do show that the fat distribution is altered in mice and similar data in humans would strengthen their conclusions. This may be beyond the scope of this article but maybe the authors should consider contacting these cohorts that would provide a means to test the hypothesis that FAM13A expression is indeed associated with body fat distribution.

2- Induction of subcutaneous adipogenesis: The authors propose that FAM13A (or rather the lack of it) is involved in inducing adipogenesis. The main observation for this conclusion is the higher expression of FAM13A in progenitor cells compared to mature adipocytes and the upregulation of the differentiation marker Fabp4 in FAM13A ko. They then explore the effect of FAM13A on adipogenesis and adipocyte function in the human SGBS pre-adipocyte cell line. They show that FAM13A is upregulated during early differentiation and lower in fully mature adipocytes. They then test the functional consequences of FAM13A ko and show an increase in basal and insulin

stimulated glucose uptake in the ko cells and conclude that FAM13A deficiency leads to generation of more and "healthier" adipocytes. However the SGBS results are not totally convincing. The authors should test if FAM13A ko influences the kinetics or amplitude of adipocyte differentiation markers, such as PPAR γ or CEBP α . If these markers appear earlier/increase that would indicate a more active differentiation/adipogenesis process.

If the authors could overexpress FAM13A in the SGBS and show that overexpression results in diminished/delayed differentiation, this would be a direct proof for an involvement in the adipogenic process.

It would also strengthen the paper if the authors could show the functional consequences in the FAM13A ko in the SGBS results in an increase of adipocyte specific function beyond glucose uptake. For example they could measure leptin and/or adiponectin secretion and look at basic functions of the mature adipocyte like lipolysis (e.g. both basal and isoproterenol induced) or lipogenesis in both the wt and ko SGBS.

Specific minor comments:

1- FAM13A expression in SAT (eQTL study)

The authors use the GTEx database to evaluate if the reported GWAS alleles are eQTLs for the genes in the region. They show that rs9991328 is a eQTL for FAM13A suggesting this gene as a reasonable regional candidate for the GWAS signal.

- It would be helpful if the authors provided the GTEx plots for each genotype of the associated SNPs (indicating the risk allele for each SNP), this would help interpreting the coherence of the data between the risk allele and the observed expression change.

- The authors state that rs9991328 is an eQTL only for FAM13A and "no other gene"? This should be more specific (i.e. no cis-eQTL for any other gene in the region? If yes then how is the region defined? From the GTEx database it seems that this SNP is a trans-qQTL for a number of other genes in subcutaneous fat. This needs to be clarified).

- Please provide the N and basic characteristics for the subgroups from the METSIM and STAGE cohorts used in the association study in the supplementary material (to my knowledge adipose tissue biopsies were taken from app. 1400 subjects from the METSIM study).

2- Mouse ko experiments

- Fig. 3A Body weight. The difference in body weight in male ko mice is very small. Is there a specific reason why there are so few WT mice included while in all other experiments the numbers are more balanced?

- Fig. 3E please switch the SAT adipocyte number/area graphs to match the VAT figures below. Otherwise the figure is somewhat confusing. More importantly, the authors state in the text the significant decrease of cell size in SAT but do not mention the equally significant increase seen in VAT adipocyte size (although numbers seem unchanged). This should be reported in the text.

3- Cell experiments

- Figures 5G/5H could be moved to supplementary material.

4- There is a recent paper by Tang et al. (IntJObes, October 2018:

<https://doi.org/10.1038/s41366-018-0222-y>) that explores the consequences of FAM13A knock out, albeit not to the same depths as the current paper. However, the authors should cite this publication and discuss their findings in the context of this paper.

Reviewer #2:

Remarks to the Author:

Genome-wide association studies have found an association of the FAM13A (Family With Sequence Similarity 13 Member A) locus with several glycemic and metabolic traits. In this paper, Fathzadeh, et al have attempted to define the link between FAM13A and these physiological

findings by further analysis of human data and creation of a Fam13a knockout mouse. They demonstrate that human FAM13A alleles associated with increased FAM13A expression in subcutaneous adipose tissue are associated with higher waist-to-hip ratio and fasting insulin levels, but lower body fat. They also report that Fam13a knockout (KO) mice have lower fasting circulating insulin, glucose, triglyceride and free fatty acid levels and an increased number of smaller adipocytes in subcutaneous adipose tissue (SAT), and that the lower VAT/SAT ratio is exacerbated by high-fat diet. They report a tendency towards improved differentiation potential in Fam13a KO preadipocytes, and link this to changes in gene expression. The then altered adipose FAM13A expression contributes to a "lipodystrophy-like" phenotype in humans and mice.

Comments:

The authors have studied an interesting question, namely what is the function of the Fam13a gene that might relate to its association of insulin-resistance traits in GWAS studies which are independent of BMI. They have used both human and murine model approaches, which should strengthen their conclusions.

However, this study suffers from many flaws, both in the presentation and interpretation of the data. These severely limit the value of any of the conclusions reported. In addition, many of the experiments are poorly described and over-interpreted.

1. To begin, the authors have chosen to call Fam13A a lipodystrophy-like gene, and base much of the introduction, discussion and abstract on this point. However, there is no study of this gene in models (human or rodent) of lipodystrophy, no evidence for association to lipodystrophy in humans, and the lipodystrophy genes which they compare Fam13A result in severe partial or complete lipodystrophy with very severe metabolic syndrome and insulin resistance. The changes associated with Fam13a are very modest and can only be observed on a population basis. They are not associated with any real clinical evidence of lipodystrophy. Thus, the title and many aspects of the discussion and conclusion are very misleading.

2. The SNPs which define Fam13a association with metabolic phenotypes are not well described nor is their relationship to gene structure. The authors refer to one variant rs9991328, which they conclude lies an active regulatory region in adipose tissue, but the structure of this region and that what regulators are acting at that region are not clear. Also, how many other SNPs have been identified? What and where are they? The panels in Figure 1 are poorly described both in the text and in the figure legend.

3. The analysis shown in Figure 2 is not clear, especially what has been adjusted for BMI and how this has been done. The text suggests that the metabolic traits are adjusted for BMI, but the horizontal axes in Figure 2b suggest that Fam13a expression has been adjusted for BMI (in at least 2 panels, but not the third).

4. In Figure 3, the number of mice studied is small and the interpretation is not supported by the data. All measures of fat cell size and number are arbitrary, when there are established methods to quantitate both cell size (diameter in microns or volume in microliters) and actual cell number (not the number of cells per image, but per fat pad). Furthermore, the histological pictures do not agree with the quantitation shown.

5. The experiments with HFD which are central to the conclusions appear to represent a single cohort of 6 vs 6 male mice and are incomplete in terms of exploring the phenotype. For example, the KO mice gain more weight than WT, but we have no data on food intake or energy expenditure. Also, although this single cohort of mice appear to show differences in SAT vs VAT, this phenotype is not observed in female mice (which only has 3 WT mice for comparison) – see point 6. No data on brown fat are presented.

6. The experiments in Supplemental Figure 4 are also hard to interpret with n's of 3 vs 7. It is also

not clear why in panels E and F of this figure there are 3 vs 7 mice, whereas in panel F, which uses the same data plotted a different way there appear to be 7 vs. 6.

7. The whole section on the ob/ob mice is confusing. The authors appear to be confused about what is the F2 cross in these data, but it is not in the B6 ob/ob mice. This adds little to the story. Also, the data in Sup. Figure 4G appear to come from the website of the Alan Attie lab and are not original to these authors. The authors refer to the Attie lab website in the methods, but do not indicate these are not their data in the Figure legend. In addition, there are two panels labelled Sup. Figure 4G.

8. The most important data, which are not followed up on and which change the interpretation of all other figures, are in Figure 5A. This shows that in adipose tissue virtually all of the Fam16a is in the stromovascular fraction, not in the adipocyte. While the authors like to conclude on page 7 that this indicates that Fam13a "may affect adipocyte generation", there is no direct data on this. The authors have not shown that the expression in the SVF is in preadipocytes vs. other cells in this fraction. Even if the Fam16a is in preadipocytes, this also means that all differences in expression observed in the previous human and murine studies could simply reflect differences in ratio of preads to adipocytes. The other needed follow-up to this observation would be to study the expression of Fam16a through preadipocyte to adipocyte differentiation.

9. The data on glucose uptake in Figure 5F show no response to insulin even in the control cells. Since differentiated adipocytes usually show a 3-10 fold increase in glucose uptake with insulin stimulation, it appears that the experiment did not work.

10. All of the figure legends are sparse and lacking details.

Reviewer #3:

Remarks to the Author:

Fathzadeh and colleagues provide some evidence from genetic studies and model KO model that FAM13A expression is associated with lower body fat and higher central fat (waist to hip ratio), higher insulin in humans. A similar phenotype of altered fat distribution and a trend toward impaired adipogenesis was observed in F13A KO mice. The manuscript addresses an important gap in knowledge about mechanisms that regulate fat distribution and consequently metabolic risk. While this phenotype is of interest, the data as presented are incomplete /preliminary and likely experiments likely underpowered to draw sufficiently strong conclusions.

Specific concerns:

1. In the mouse model, there is an effect of F13a KO only in males. Is the low fat phenotype only seen in human males, not females? If not, the translational relevance/limitation of the mouse model should be discussed.
2. Figure 3E and F: Adipocyte number and size per image are calculated (E). To enable the reader to evaluate the data and most importantly to assess whether the lower adipocyte size in SAT and the greater size in VAT affect the number of adipocyte per depot (i.e. group difference in hyperplasia vs hypertrophy). Pixels should not be presented – they should be converted to microns and a scale bar should be shown on the micrographs shown in Figure F. Furthermore and importantly, the number of adipocytes per depot in each animal should be calculated by dividing depot weight by mean adipocyte weight (ug lipid/cell calculated from mean volume and the specific gravity of triglyceride (0.915)) and is easily calculated by standard formulas in the literature (and assuming a % lipid in the depot (~85% in VAT, ~75% in SAT) if a direct measure is not available).
3. It is stated that FAM13a expression was negatively correlated with 'adipose cellular proliferation' but no direct measurements of the proliferation of adipose progenitors was presented. Changes in the cell cycle pathway may suggest this, but no conclusion can be drawn without direct measurements.

4. It would be important to list the transcripts associated with pathways listed as differentially regulated (Fig S4 I and J) - what is magnitude of differences in transcripts at the leading edge of the differentially regulated pathways.

5. Figure S5B shows a low magnification image of differentiated adipose progenitors from iWAT. It appears that fewer cells differentiated into adipocytes in the KO fat. However, no quantitation of lipid/DNA or the reproducibility of this apparent finding is provided. Also, only one late marker of differentiation (Fabp4) was measured and this is inadequate. The figure legends do not provide adequate detail to evaluate the rigor or reproducibility of the findings (e.g. number of independent observations/number of preparations from different mice and their age).

6. The header on page 6, last paragraph states 'Fam13A tends to induce subcutaneous adipogenesis' - additional and more rigorous experiments are needed to even support this weak statement. It is interesting that Fam13a is higher in SVF than floating adipocytes, but freshly isolated SVF is a mixed population of cells (Fig FA). A time course of changes in F13a gene expression from confluent cultures (mainly adipose progenitors/fibroblasts) until D10 is needed to address this question more directly. Figure 5E provides oil red O staining in WT and KO cultures. However, this is poorly quantitative measure and no denominator (i.e. number of cells per plate by DNA or some other measure) is provided. Further, the differences in Fabp4 expression (a marker of differentiation) were not statistically significant.

Figure 5F also shows non-significant trends for basal and glucose uptake in iWAT from the control and KO mice. Unfortunately, the data are presented as "fold over basal of WT" so can't be properly evaluated with respect to actual rates, and are confounded by the apparently higher number or size of differentiated adipocytes in the KO. A denominator like triglyceride content/well or protein is needed. Further, the insulin effect on glucose uptake is far smaller than expected, questions the preparation or the assay. The number of observations/mice/cultures is not provided in the figure legends (and should be) - but it appears the study was underpowered.

The final sentence of the prior paragraph also states 'data from F2ob/ob mice support a role of Fam13a in adipose turnover and fat plasticity'. No direct experimental data are provided to support this statement either.

Reviewer #4:

Remarks to the Author:

In "The human lipodystrophy-like gene FAM13A affects fat distribution and function," Dr. Fathzadeh and colleagues first focus on genetic variation in the FAM13A locus -- identified via genome-wide associations of fat distribution, insulin resistance, and other cardiometabolic phenotypes -- and then proceed to study the potential biological mechanism of changes in FAM13A on fat distribution.

First, I would like to commend the authors on a very clearly written manuscript. The writing is crisp, succinct, avoids the passive voice, and is overall very easy to follow. The figures are equally clear and can stand alone as helpful pieces of narrative.

I do have comments on some of the analytic pieces, which I will further outline below:

Major comments

[1] The biggest piecing piece for me is how, precisely, the authors get from non-coding variants in FAM13A to focusing all of their analyses on strictly the FAM13A gene. GWAS loci are notoriously complex due to LD, gene density in the region, and the fact that associated SNPs themselves are non-coding. While I understand that the SNPs the authors have focused on lie in FAM13A, this is no guarantee that FAM13A is the gene to focus on (the FTO locus in BMI is perhaps the best argument for this; intronic FTO variants seem to influence regulation of IRX3 and IRX5 -- genes that are quite some distance away -- which in turn affects adiposity).

Did the authors do any fine-mapping to figure out if FAM13A was the gene to focus on? Or look to see if this locus falls in a topologically associating domain? What other evidence did the authors consider before ultimately deciding that FAM13A was the single gene they wanted to focus on?

[2] The analyses presented in the first paragraph of the results, in my opinion, need a bit more fleshing out to feel truly robust. First, it's not clear how many SNPs you are actually looking up. Is it only the two SNPs mentioned in the paragraph before this? Or SNPs that span the full locus?

Next, to say that a SNP is an eQTL is, on its own, rather unsurprising given the ubiquity of eQTLs in the genome. Would there be a way to assess whether this particular eQTL finding is non-random? How likely is it to pull a SNP from the genome and fall upon an eQTL in adipose tissue?

Lastly, when you say that the SNP and some LD partners fall into an H3K27ac epigenetic mark tagging enhancers in adipose nuclei, could you assess whether this enrichment is non-random? For example, this paper (<https://www.nature.com/articles/ng.3437>) gives a nice method for deciding if SNPs in a locus are enriched in a particular annotation. It seems helpful to show this, particularly if the footprint of the annotation is large and it is 'easy' for a SNP to fall into the annotation? Do you show lung and pancreas in Figure 1 to demonstrate a negative comparator? The reason that data also appears isn't clear.

[3] The colocalisation analysis presented in the next paragraph is a far more compelling piece of data than the eQTL lookup. This analysis made me wonder if the authors had thought of running Bayesian fine-mapping in this region, before embarking on the rest of their analysis, so as to better understand which SNPs might be the likely causal ones?

[4] I am not a mouse expert, so I will leave comment on the mouse work to other reviewers. However, I did realise that most of the mice studied were male. Fat distribution -- as shown from GWAS -- has a strong sex dimorphic signature; some biology appears to be truly sex specific. Did the authors check to see if FAM13A has a dimorphic signal? Did they consider this when selecting mice for experimentation? Could the sex dimorphism in fat distribution influence how generalisable these results are to either men or women?

Minor comments

[1] On the whole, I wanted more data to be directly reported in the text. For example, when associations or correlations are mentioned, stating the association effect and p-value or the value of the correlation would be tremendously helpful. The writing feels a bit too qualitative at the moment (with much of the numerical bits show in the figures). Textual bits that focus on the quantitative details would be very helpful.

[2] Similar to my point above, including details such as sample sizes in the mouse analyses would be helpful.

[3] I have, as a reviewer, become a real stickler for authors placing data and code online in an accessible place. Will the data and code for this project be released? This is a crucial detail to make scientific work more open, transparent, and reproducible.

Reviewer #1 (Remarks to the Author):

General comments:

In this paper Fathzadeh et al. provide evidence that *FAM13A* plays a role in fat distribution in humans. Specifically, the paper addresses two important issues:

1- they provide evidence that *FAM13A* is indeed the gene responsible for the GWAS signal for the insulin resistant trait related variants reported by Scott et al. and Willer et al. This is important as there is a deplorable lack of GWAS hits for which the causal gene has been identified. The expression analyses that have been performed by the authors are state-of-the art and provide coherent evidence that points to *FAM13A* as the most likely gene responsible for the GWAS signal.

- Thank you for this thoughtful comment highlighting the importance of functional genomics studies of insulin resistance-related variants.

2- they show that the functional consequences of *FAM13A* are most likely associated with a difference in the distribution of subcutaneous to visceral fat depts. This is an important finding. It has long been hypothesized that individuals that have a more flexible subcutaneous fat tissue (i.e. more expandable under over-feeding conditions) are somewhat protected from the detrimental metabolic effects and vice versa that the accumulation of visceral fat, even in the absence of excess BMI (which is mainly accounted for by subcutaneous fat accumulation) is associated with insulin resistance and risk of CAD. However, the mechanisms for this difference in distribution are unknown. Thus, the paper proposes for the first time a possible biological mechanism for this hypothesis.

- We appreciate the Reviewer's opinion that emphasizes a main goal of this study: exploring *FAM13A* regulation of fat distribution between subcutaneous and visceral depots. The "Adipose tissue expandability" hypothesis suggests that a limited capacity of (subcutaneous) fat in storing excess lipids may lead to cardiometabolic comorbidities. We believe that our work focusing on adipocyte development and fat distribution contributes to the understanding of the biological mechanism(s) underlying the association of *FAM13A* risk variants with cardiometabolic complications.

The methods used to provide evidence for the two topics are adequate and the analysis and experiments are well designed overall. The authors have gone to great lengths to provide evidence based on *in silico* methods, cell based assays, human cohort studies and mouse ko experiments. This is commendable, especially the inclusion of human cohorts to show an association between the *FAM13A* SAT expression and the primary hypothesis i.e. that this expression is associated with differences in SAT/VAT fat distribution or ratio. Thus this paper provides data that may be very important in our understanding of differential fat deposition in humans.

- We thank the Reviewer for the support of our approach of combining human genetics data with *in vitro* and *in vivo* experimental studies. We appreciate the Reviewer's opinion that follow-up studies on differential fat deposition will help gain insight into genetic associations with cardiometabolic diseases.

Major comments:

1- Human cohort studies: To really show that *FAM13A* impacts fat distribution in humans these analyses should ideally be done in a cohort that would allow to assess to directly study the association between *FAM13A* expression and body fat distribution using gold standard methods for body composition i.e. by DEXA. The two cohorts studied in the article only provide crude proxies for body fat distribution like waist/hip ratio. There are

some well-studied cohorts that have access to both SAT expression data as well as DEXA data (e.g. DiOGenes) that could be used.

The authors do show that the fat distribution is altered in mice and similar data in humans would strengthen their conclusions. This may be beyond the scope of this article but maybe the authors should consider contacting these cohorts that would provide a means to test the hypothesis that *FAM13A* expression is indeed associated with body fat distribution.

- We thank the Reviewer for this insightful comment and suggestion. The Reviewer raises an important point that it would be preferable to have additional information from cohorts using gold standard methods for body composition such as DEXA. As the Reviewer notes, incorporating additional, unpublished primary data from cohorts such as DiOGenes is beyond the scope of this paper; but we appreciate this methodology for our future studies. However, we explored publicly available GWAS data for evidence of association between variants in the *FAM13A* locus and VAT/SAT ratio, indicative of their association with body fat distribution. A 2017 study from Chu et al. ¹ explored genetic associations with radiologically-derived ectopic fat measures including SAT and VAT volume, and VAT/SAT ratio in a cohort of men and women. Although variants at the *FAM13A* locus were not associated with these traits at genome-wide significance, an inflation in locus-specific p-values is observed in the associations with VAT/SAT ratio as compared with SAT or VAT volume associations as shown in the figure below. Considering that association statistics from same variants with similar linkage disequilibrium patterns are presented across the panels of this figure, this pattern suggests the presence of a weak association signal between variants at the *FAM13A* locus and fat distribution. Further, as demonstrated by Bayesian fine-mapping and colocalization results with adiposity-related traits discussed below and in the manuscript, the same variants also show robust association with *FAM13A* expression in SAT. In the absence of direct measurement of *FAM13A* expression in large human cohorts with relevant phenotypes, this analysis suggests that *FAM13A* expression is associated with body fat distribution.

- While some of our analyses rely on proxies for overall body composition (such as BMI and WHRadjBMI), we also include data from some cohorts using less crude measures such as bioimpedance. For instance, in the METSIM cohort, body fat percentage was assessed by bioimpedance. There are studies ² that have demonstrated that bioimpedance is a good alternative for estimating body fat percentage when subjects are within a normal body fat range. Moreover, a recent combined study of abdominal MRI and GWAS of favorable adiposity have used bioimpedance measures of body fat % as a measure of adiposity in the discovery step ³.
- Our PheWAS analysis of rs1377290 in the UKBB demonstrates robust association with bioimpedance measures including body fat percentage ($p = 7.2e-13$) and trunk fat percentage ($p = 9.33e-15$). The variant rs1377290 was selected for the following reasons: (1) It has an LD R² of 1.0 with rs9991328, which has the highest Bayesian fine-mapping posterior probability of being causal for the association signals in SAT eQTL analysis and several GWAS traits we considered; and (2) it is directly genotyped in the UKBB.

2. Induction of subcutaneous adipogenesis: The authors propose that *FAM13A* (or rather the lack of it) is involved in inducing adipogenesis. The main observation for this conclusion is the higher expression of *FAM13A* in progenitor cells compared to mature adipocytes and the upregulation of the differentiation marker *Fabp4* in

FAM13A ko. They then explore the effect of FAM13A on adipogenesis and adipocyte function in the human SGBS pre-adipocyte cell line. They show that FAM13A is upregulated during early differentiation and lower in fully mature adipocytes. They then test the functional consequences of FAM13A ko and show an increase in basal and insulin stimulated glucose uptake in the ko cells and conclude that FAM13A deficiency leads to generation of more and “healthier” adipocytes. However the SGBS results are not totally convincing. The authors should test if FAM13A ko influences the kinetics or amplitude of adipocyte differentiation markers, such as PPARg or CEBPa. If these markers appear earlier/increase that would indicate a more active differentiation/adipogenesis process.

- We would like to thank the Reviewer for suggesting that we provide more experimental support in human SGBS cells. We have now added loss-of-function experiments on FAM13A in human cells. We knocked down FAM13A in human SGBS preadipocytes by siRNA and CRISPR interference (CRISPRi), separately; and obtained similar results with both methods: the expression of the adipogenic marker PPARg and CEBPa were measured to show an upregulated differentiation of FAM13A-deficient SGBS adipocytes following early adipogenic induction (Fig. 6B-E). These data suggest that FAM13A knockdown in the preadipocyte stage may initiate a more active adipogenic process. We added in Fig. 6B-E (below) and the following text to describe the results:

Fig. 6B and 6C: Impact of siRNA knockdown of FAM13A on adipogenesis of SGBS preadipocytes

Fig. 6D and 6E: Impact of CRISPRi knockdown of FAM13A on adipogenesis of SGBS preadipocytes

p.8, line 9: FAM13A knockdown by siRNA in SGBS preadipocytes (Fig. 6B) resulted in upregulation of adipogenesis markers e.g. CEBPA and PPARG; Fig. 6C) as well as beige adipogenic markers (e.g. PGC1A and UCP1; Fig. S6A), in the early differentiation stage (Day 5). In addition, 90-95% of FAM13A knockdown was also achieved in preadipocytes by CRISPR-interference (CRISPRi), using three sgRNAs targeting FAM13A independently (Fig. 6D). A similar induction of adipogenic markers (CEBPA and PPARG) in early differentiating adipocytes (Day 5; Fig. 6E) also indicated a more active adipogenic capability of FAM13A-deficient preadipocytes.

3. If the authors could overexpress FAM13A in the SGBS and show that overexpression results in diminished/delayed differentiation, this would be a direct proof for an involvement in the adipogenic process.

- We agree with the Reviewer that gain-of-function studies can further explore the role of FAM13A in adipogenesis. Indeed, after our original submission, another group of investigators performed such experiments showing that overexpression of Fam13a (Fam13a-OE) in mouse 3T3-L1 preadipocytes suppresses preadipocyte survival and inhibits early-stage adipocyte differentiation⁴. This was evidenced by reduced expression of adipogenic markers (Pparg, Cebpa, Plin1), decreased ORO staining and diminished intracellular TG content in Fam13a-OE adipocytes 8 days after adipogenic induction. These results agree with our findings from loss-of-function studies in human SGBS preadipocytes that FAM13A negatively regulate adipogenesis. Given these observations, we did not carry out Fam13a-OE experiments in SGBS cells, but have discussed this in the manuscript and added the appropriate citation as follows:

p.11, line 25.: Tang et al³⁵ showed that overexpression of Fam13a in mouse 3T3-L1 cells led to apoptosis of preadipocyte and diminished its conversion to adipocytes.

4. It would also strengthen the paper if the authors could show the functional consequences in the *FAM13A* ko in the SGBS results in an increase of adipocyte specific function beyond glucose uptake. For example they could measure leptin and/or adiponectin secretion and look at basic functions of the mature adipocyte like lipolysis (e.g. both basal and isoproterenol induced) or lipogenesis in both the wt and ko SGBS.

- We thank the Reviewer for the suggestion. We have now studied the potential impact of *FAM13A* knockdown on adipocyte-specific functions beyond glucose uptake. In human SGBS cells, knockdown of *FAM13A* by siRNA in differentiating adipocytes did not significantly influence the expression level of adiponectin or leptin (Fig. S6B). Thus, we did not pursue studies to measure the secretion of these adipokines. Regarding lipolysis, another basic metabolic function of adipocytes, we did not observe a significant difference in either basal or isoproterenol-stimulated WT and *FAM13A*-KD SGBS adipocytes (Fig. S6C). We agree with the Reviewer that more rigorous studies are needed to fully uncover the roles of *FAM13A* in mature adipocyte-specific functions.

Fig. S6B: Impact of *FAM13A* KD on SGBS adipokine expression

Fig. S6C: Impact of *FAM13A* KD on SGBS lipolysis

(a) Specific minor comments:

1- FAM13A expression in SAT (eQTL study)

The authors use the GTEx database to evaluate if the reported GWAS alleles are eQTLs for the genes in the region. They show that rs9991328 is an eQTL for *FAM13A* suggesting this gene as a reasonable regional candidate for the GWAS signal. It would be helpful if the authors provided the GTEx plots for each genotype of the associated SNPs (indicating the risk allele for each SNP), this would help interpreting the coherence of the data between the risk allele and the observed expression change.

- We would like to thank the Reviewer for the suggestion of providing a clear visual representation of the relationship between *FAM13A* expression and rs9991328 genotype in GTEx (v7). We have included the following plot showing this relationship in subcutaneous and visceral adipose tissues in **Supplementary Figure 1**.

Supplemental Fig 1.

- The authors state that rs9991328 is an eQTL only for *FAM13A* and “no other gene”? This should be more specific (i.e. no cis-eQTL for any other gene in the region? If yes then how is the region defined? From the GTEx database it seems that this SNP is a trans-qQTL for a number of other genes in subcutaneous fat. This needs to be clarified).

- We thank the reviewer for pointing the lack of clarity in this statement. We intended to convey that rs9991328 is a cis-eQTL only for *FAM13A* and not for any other gene in the region. The cis-eQTL analysis was performed using previously used parameters⁵. Variants within 1Mb of the transcription start site of each gene, with a minor allele frequency > 0.01 and minor allele observed in at least 10 samples in the tissue of interest were

tested for association with the expression of each gene. In this analysis, rs9991328 was a cis-eQTL solely for *FAM13A* in SAT but not in VAT as depicted in **Figure 1**.

- We did not observe rs9991328 to be a significant trans-eQTL for any genes in either SAT or VAT. Significant trans-eQTL association data is publicly available on the GTEx portal datasets page (<https://gtexportal.org/home/datasets>) under the GTEx Analysis V6p section. To address this lack of clarity, we have now amended the text in the Results section on page to read as follows:

p. 4, line 26: In SAT, rs9991328 showed a robust association with FAM13A expression (P=1e-08) and was not a cis-eQTL (expression quantitative trait locus) for any other genes with transcription start sites within 1Mb of the variant. In contrast, rs9991328 showed a weaker association with FAM13A expression (P=1.5e-05) in VAT. The variant was also not a significant trans-eQTL for any genes in adipose tissue (data from GTEx v6p).

- Please provide the N and basic characteristics for the subgroups from the METSIM and STAGE cohorts used in the association study in the supplementary material (to my knowledge adipose tissue biopsies were taken from app. 1400 subjects from the METSIM study).

- For the METSIM expression arrays, there were 770 individuals in total. There were 1400 biopsies taken, although only 770 subjected to the expression arrays. These were all men collected from Kupio Finland, age 45-73 (<http://www.nationalbiobanks.fi/index.php/studies2/10-metsim>)⁶. STAGE consisted of 105 individuals (both genders) collected from Sweden and Estonia, age 58-74. In this study, 72 individuals were subjected to expression arrays using the HuRSTA-2a520709 Affy platform⁷. We have added these data to the manuscript in the Methods section (subsection “Human cohort studies”; “METSIM and STAGE” and the figure legends where appropriate.

2. Mouse ko experiments

- Fig. 3A Body weight. The difference in body weight in male ko mice is very small. Is there a specific reason why there are so few WT mice included while in all other experiments the numbers are more balanced?

We agree with the point that the body weight changes seen are small and that including more WT mice would have been ideal. For consistency, these studies were performed on the entire cohorts of available WT (n=7) and KO (n=14) mice (age- and sex-matched) in our colony at the time of starting the experiment for body weight monitoring. These results are consistent with another study from *Tang et al.*⁴ that was published after our initial submission and that showed no significant difference between WT and KO male mice in weight gain under normal chow condition. To reflect these minor changes, we have revised the manuscript as follows:

p.6, line 2): Under normal chow-fed dietary conditions, male WT and Fam13a KO mice showed no significant difference in body weight (Fig. 3A), mild decreases in plasma concentrations of fasting glucose, insulin, free fatty acids, and triglycerides (Fig. 3B), and no change in glucose or insulin tolerance tests (Fig. 3C-D). No differences in the above metabolic phenotypes were observed between female WT and Fam13a KO mice (Fig. 3J).

- Fig. 3E please switch the SAT adipocyte number/area graphs to match the VAT figures below. Otherwise the figure is somewhat confusing. More importantly, the authors state in the text the significant decrease of cell size in SAT but do not mention the equally significant increase seen in VAT adipocyte size (although numbers seem unchanged). This should be reported in the text.

- We appreciate this comment and have revised the figures accordingly. To provide more detailed and accurate quantitative analysis of adipocyte size and number, we present total number of adipocytes per fat pad in **Fig. 3H** and distribution of adipocyte size in **Fig. 3I-J**, as shown below, to replace the column comparison of the mean size or number in the original manuscript. We also added the following text:

p.6, line 13 : However, there was a significant increase in the number of small adipocytes in SAT (Fig. 3G and 3I) with a corresponding increase in the diameter of large adipocytes in VAT (Fig. 3G and 3J) of Fam13a KO mice. These results indicate that depot-specific size distribution of adipocytes may be associated with the presence of Fam13a in adipose tissue.

Fig. 3H: Total number of adipocytes per fat pad

Fig. 3I and 3J: Adipocyte size distribution in SAT and VAT of WT vs. *Fam13a* KO

3. Cell experiments

- Figures 5G/5H could be moved to supplementary material.

- We agree and have done this as suggested. In addition, we performed an extended set of experiments in human SGBS cells and have now included new and stronger evidence in support of the role of *FAM13A* in human adipocyte differentiation and metabolic function (**Figure 6**) in place of the previous panels G and H in Figure 5.

4. There is a recent paper by Tang et al. (IntJObes, October 2018: <https://doi.org/10.1038/s41366-018-0222-y>) that explores the consequences of *FAM13A* knock out, albeit not to the same depths as the current paper. However, the authors should cite this publication and discuss their findings in the context of this paper.

- We appreciate the Reviewer for bringing up this paper, which was published shortly after our initial submission. Our data agrees with three major observations presented by *Tang et al.*⁴: 1) *Fam13a* expression was downregulated in adipose tissue upon HFD; 2) *Fam13a* KO mice exhibited a tendency of higher adiposity, without developing insulin resistance; 3) Overexpression of *Fam13a* in 3T3-L1 led to apoptosis of preadipocyte and diminished its conversion to adipocytes. Thus, taken together, *Fam13a* seems to act as a negative regulator of adipogenesis. We have now cited this paper and discussed their findings in the context of ours. We added in the following text in the manuscript "section "DISCUSSION",:

*p.11, line 19: Tang et al.*³⁵ showed a negative correlation of adipose *Fam13a* expression with diet-induced obesity in mice, in agreement with our data.

*p.11, line 25: Tang et al.*³⁵ showed that overexpression of *Fam13a* in mouse 3T3-L1 cells led to apoptosis of preadipocyte and diminished its conversion to adipocytes.

*p.11, line 30: we and Tang et al.*³⁵ detected no difference in either insulin sensitivity or glucose tolerance between WT and *Fam13a* KO mice under both chow and HFD, despite a slightly increased adiposity in *Fam13a* KO mice after HFD.

Reviewer #2 (Remarks to the Author):

Genome-wide association studies have found an association of the **FAM13A (Family With Sequence Similarity 13 Member A)** locus with several glycemic and metabolic traits. In this paper, Fathzadeh, et al have attempted to define the link between **FAM13A** and these physiological findings by further analysis of human data and creation of a **Fam13a** knockout mouse. They demonstrate that human **FAM13A** alleles associated with increased **FAM13A** expression in subcutaneous adipose tissue are associated with higher waist-to-hip ratio and fasting insulin levels, but lower body fat. They also report that **Fam13a** knockout (KO) mice have lower fasting circulating insulin, glucose, triglyceride and free fatty acid levels and an increased number of smaller adipocytes in subcutaneous adipose tissue (SAT), and that the lower VAT/SAT ratio is exacerbated by high-fat diet. They report a tendency towards improved differentiation potential in **Fam13a** KO preadipocytes, and link this to changes in gene expression. The then altered adipose **FAM13A** expression contributes to a “lipodystrophy-like” phenotype in humans and mice.

Comments:

The authors have studied an interesting question, namely what is the function of the **Fam13a** gene that might relate to its association of insulin-resistance traits in GWAS studies, which are independent of BMI. They have used both human and murine model approaches, which should strengthen their conclusions.

However, this study suffers from many flaws, both in the presentation and interpretation of the data. These severely limit the value of any of the conclusions reported. In addition, many of the experiments are poorly described and over-interpreted.

- We thank the Reviewer for appreciating that the overall purpose of the work is to begin characterization of a biological function for *FAM13A* in the context of insulin resistance, based on findings from GWAS of related traits. We have made substantial revisions of the paper both in terms of presentation and interpretation of the data. We have also performed extensive additional relevant experiments to clarify the concerns over the causal effects of *Fam13a* on adipocyte differentiation, inadequate analyses of the animal model and the lack of interpretation of sex-specific effects. We have described all the experiments in details and interpreted the results more precisely. We hope the Reviewer will find following experiments and detailed responses useful.

1. To begin, the authors have chosen to call **Fam13A a lipodystrophy-like gene, and base much of the introduction, discussion and abstract on this point. However, there is no study of this gene in models (human or rodent) of lipodystrophy, no evidence for association to lipodystrophy in humans, and the lipodystrophy genes which they compare **Fam13A** result in severe partial or complete lipodystrophy with very severe metabolic syndrome and insulin resistance. The changes associated with **Fam13a** are very modest and can only be observed on a population basis. They are not associated with any real clinical evidence of lipodystrophy. Thus, the title and many aspects of the discussion and conclusion are very misleading.**

- We apologize for the confusion that the use of the word "lipodystrophy" has created. We agree with the Reviewer that this gene is not a classic lipodystrophy gene, and we do not have direct evidence supporting this gene's association with lipodystrophy in either human or rodent models. Therefore, in the revised manuscript, we removed "lipodystrophy" from the title, abstract, results and the conclusion. The revised title is: "*FAM13A* affects body fat distribution and adipocyte function".
- We chose the original title because this term “subtle form of lipodystrophy” (or “lipodystrophy-like”) has already been used by other researchers^{8,9} to describe a common but milder form of the phenotype in the general population. Epidemiological studies have specified that the location and distribution of excess fat, rather than

overall adiposity, are more revealing for predicting population-level polygenic risk score of obesity and cardiometabolic disease and cancer¹⁰. Body fat distribution is a complex trait with varying degrees of phenotypic diversity³. As the list of involved loci is expanding we have learned that the majority of loci associated with body fat distribution and with limited peripheral adipose storage capacity, such as *FAM13A*, are not classic lipodystrophy genes^{9,10}.

2. The SNPs which define *Fam13a* association with metabolic phenotypes are not well described nor is their relationship to gene structure. The authors refer to one variant rs9991328, which they conclude lies an active regulatory region in adipose tissue, but the structure of this region and that what regulators are acting at that region are not clear. Also, how many other SNPs have been identified? What and where are they? The panels in Figure 1 are poorly described both in the text and in the figure legend.

- We thank the Reviewer for suggesting that we include additional details clarifying our variant-level fine-mapping analyses at the *FAM13A* locus. As with many GWAS loci, identifying the causal variant presents a significant challenge due to linkage disequilibrium. In our study, we performed Bayesian fine-mapping to obtain variant-level posterior probabilities of being causal for observed association signals, including for the eQTL signal in SAT and phenotypic traits such as fasting insulin (both adjusted and unadjusted for BMI), WHR (adjusted for BMI), HDL and TG. Across all of these traits, rs9991328 ranked either the highest or in the top 5 variants ordered by posterior probability of driving the association signal. Further, rs9991328 was the top-ranked variant in fine-mapping analysis of the subcutaneous adipose eQTL signal. Therefore, while other SNPs may also be associated with the phenotype due to linkage disequilibrium, we decided to perform colocalization analyses using rs9991328 as the top-variant.
- For the UKBB phenome-wide association study depicted in **Figure 2**, we used rs1377290 instead of rs9991328 for two reasons: (1) rs1377290 has an LD R² of 1.0 with rs9991328, which has the highest Bayesian fine-mapping posterior probability of being causal for the association signals in SAT eQTL analysis and several GWAS traits considered, (2) It is directly genotyped in the UK Biobank while rs9991328 is imputed. To add details about the finemapping analysis and clarify how we chose to highlight these two variants, we have included the following text in the Results section of the manuscript:

*p.4, line 20: FINEMAP algorithm revealed that rs9991328, which is associated with increased gene expression ($b = 0.22 \pm 0.03$, $p = 1e-08$), had the highest posterior probability of driving the signal (finemapping posterior probability = 0.66) among all the variants in the region; we retained this variant as the lead variant for further analyses. This variant is located within an intronic region ~3.2Mb away from the transcription start site of *FAM13A*. The most commonly reported GWAS lead variant, rs3822072, was among the top 5 variants in the finemapping analysis (posterior probability = 0.06).*

*p.5, line 7: We also performed a phenome-wide association study (PheWAS) in 337,536 individuals from the UK Biobank^{11,12} using the directly genotyped variant rs1377290 as a proxy for the imputed variant rs9991328 (LD R² = 1.0). The rs1377290 T allele was associated with increased WHR, and decreased trunk/body fat percentage, mean platelet volume, and BMI (**Fig. 2A**). There were no other significant associations across the phenotypic spectrum.*

- The variants we refer to lie in an epigenetic mark that tags active promoters (H3K27ac). We also performed a statistical enrichment analysis in which we defined a 99% credible set consisting of 16 linked variants at the *FAM13A* locus that are most likely to be responsible for driving the association at this locus. We used FINEMAP posterior probabilities to define this set – variants were included as long as their cumulative fine-mapping posterior probability for the SAT eQTL signal was ≤ 0.99 . To perform the enrichment analysis, we shifted the annotations randomly by a maximum of 500kb and estimated the proportion of randomizations in which the number of credible set variants that overlap with the annotations was greater than or equal to the test statistic. While insignificant, adipose nuclei showed evidence of enrichment (adjusted $p = 0.07$) compared with no evidence in other tissues, which are highlighted as negative comparators (p values in these other tissues). Further, visual inspection of H3K27ac annotation as shown in the bottom panel of **Figure 1** suggests stronger concentration of this epigenomic annotation in the locus compared with the negative comparators. Due to the enrichment analysis potentially being influenced by the type of fine-mapping used and the number of variants

included in the credible set, we chose to report this overlap due to its suggestive low p-value. We highlight this analysis in the Results and the Methods with the following text:

p.14, line 24: We performed a statistical enrichment analysis for epigenomic marks in which we defined a 99% credible set consisting of 16 linked variants at the FAM13A locus that are most likely to be responsible for driving the association at this locus. We used FINEMAP posterior probabilities to define this set – variants were included as long as their cumulative fine-mapping posterior probability for the SAT eQTL signal was ≤ 0.99 . To perform the enrichment analysis, shifted the annotations randomly by a maximum of 500kb and estimated the proportion of samples in which the number of credible set variants that overlap with the annotations was greater than or equal to the test statistic.

- We also thank the Reviewer for suggesting that we highlight the location of the SNPs mentioned in the analysis. We have explained the location of the lead finemapping SNP rs9991328 in relation to gene structure using the following text in the Results section:

p. 4, line 19 : Bayesian finemapping analysis of the adipose subcutaneous expression association signal using the FINEMAP algorithm revealed that rs9991328, which is associated with increased gene expression ($b = 0.22 \pm 0.03$, $p = 1e-08$), had the highest posterior probability of driving the signal (finemapping posterior probability = 0.66) among all the variants in the region; we retained rs9991328 as the lead variant for further analyses. This variant is located within an intronic region ~ 3.2 Mb away from the transcription start site of FAM13A. The most commonly reported GWAS lead variant, rs3822072, was among the top 5 variants in the finemapping analysis (posterior probability = 0.06).

3. The analysis shown in Figure 2 is not clear, especially what has been adjusted for BMI and how this has been done. The text suggests that the metabolic traits are adjusted for BMI, but the horizontal axes in Figure 2b suggest that Fam13a expression has been adjusted for BMI (in at least 2 panels, but not the third).

- We thank the Reviewer for bringing this to our attention. As described in the text, the metabolic traits were adjusted for BMI. We have altered the x-and y-axes in **Figure 2b** to reflect the appropriate adjustment for BMI.

4. In Figure 3, the number of mice studied is small and the interpretation is not supported by the data. All measures of fat cell size and number are arbitrary, when there are established methods to quantitate both cell size (diameter in microns or volume in microliters) and actual cell number (not the number of cells per image, but per fat pad). Furthermore, the histological pictures do not agree with the quantitation shown.

- We appreciate the comment regarding cell size and number quantification. We have reexamined and extended the adipose histomorphometry analyses in WT and *Fam13a* KO mice, now using more established methods to quantify both cell size and number. We have described these changes in the methods in the text (p. 15, under subsection “Quantification of adipocyte size and number”). Results are now presented in microns and the scale bar in micrometers is shown on each image. The actual number of adipocytes was quantified per fat depot (**Fig. 3H**), rather than counting number of cells per image. In addition, the distribution of SAT and VAT adipocyte diameter, instead of the mean cell size, in *Fam13a* KO and WT mice is now presented in **Fig. 3I** and **3J**, to reflect our quantification analysis more accurately and with greater detail. The representative histological picture is shown in **Fig. 3G**. The interpretation matching the histological and quantification data is presented in the revised manuscript as follows:

p.17, line 6: *Methods: Briefly, microphotographs of H&E stained adipose tissue blocks were acquired from an optical microscope (Zeiss Axioplan2) at 20x magnification and images were captured with a Leica DC500 camera and a NIS Elements software. ImageJ software with the H&E color deconvolution plugin was used to determine the size of the adipocytes and the number of adipocytes within a certain diameter range by two individuals in a double blinded manner. Four fields of view for SAT slides and six fields of view for VAT slides were quantified. SAT and VAT from seven animals per group were analyzed. A total of 2000-2200 cell diameters were measured by Adiposoft, an automated software for analysis of adipose tissue cellularity as established¹³. The quantification unit of pixels were converted to microns by Photoshop CS6 (1 pixel=0.3953 μm at 20x objective). Adipocyte number per fat depot was calculated by dividing the total tissue weight by the mean adipocyte weight¹⁴. Total tissue weight was measured directly after tissue dissection, and the mean adipocyte weight was calculated by multiplying the mean adipocyte volume by the triglyceride density (0.915 g/L). The mean volume of adipocyte was calculated from the diameters measured above by applying the following formula: $V = 4/3 \pi (D/2)^3$.*

Interpretation of the data (p.6, line 12): Normal histological appearance was maintained in Fam13a KO mice for both SAT and VAT (Fig. 3G). Although the total number of adipocytes per fat pad was not significantly different between WT and KO mice (Fig. 3H), there was a significant increase in the number of small adipocytes in SAT (Fig. 3G and 3I) with a corresponding increase in the diameter of large adipocytes in VAT (Fig. 3G and 3J) of Fam13a KO mice. These results indicate that depot-specific size distribution of adipocytes may be associated with the presence of Fam13a in adipose tissue.

5. The experiments with HFD which are central to the conclusions appear to represent a single cohort of 6 vs 6 male mice and are incomplete in terms of exploring the phenotype. For example, the KO mice gain more weight than WT, but we have no data on food intake or energy expenditure. Also, although this single cohort of mice appear to show differences in SAT vs VAT, this phenotype is not observed in female mice (which only has 3 WT mice for comparison) – see point 6. No data on brown fat is presented.

- In response to this comment from the Reviewer, we have: 1) studied additional male mice on HFD; 2) discussed the data on food intake or energy expenditure from another study; 3) explained the lack of phenotypes in female mice; and 4) explored the potential role of *FAM13A* in brown/beige fat.
- First, we studied an additional cohort of 5 WT and 5 KO male mice on HFD for 6-wks. Even with this shorter duration of HFD treatment, we observed a trend to a reduction in the VAT/SAT ratio of KO mice (shown on the right), supporting the initial observation and indicating a role of *Fam13a* in regulating fat distribution during the onset of obesity.
- Second, the assessment of food intake and whole-body energy expenditure was conducted by another group *Wardhana et al.*³⁶ using metabolic chambers. No difference was observed between WT and *Fam13a* KO mice, as now mentioned and discussed in the text as below. Therefore, we did not repeat this set of experiments, but focused on exploring the role of *Fam13a* in more fat-specific phenotypes, such as lipid profiling, fat depot mass or distribution, and adipocyte development.

p. 11, line 26: *Regarding whole-body metabolism, Wardhana et al.³⁶ showed no difference in food intake, oxygen consumption, respiratory exchange ratio, physical activity and body temperature (under room temperature) between WT and Fam13a-overexpression mice after 14-week HFD challenge.*

- Third, in female mice, we have not seen any significant difference between WT and KO regarding any of the phenotypes assessed. Although, we kept all data about the female mice in the supplementary figures and did not repeat the experiments during the revision process, we have now more carefully discussed these observations in the revised manuscript (also see response to comment #6).

- Last but not least, we performed some studies on brown fat in response to the Reviewer's comment. We did not observe significant difference in the appearance (i.e. color and size), nor the expression level of brown fat-specific markers (e.g. *Ucp1*, *Pgc1a* or *Cidea*) in brown adipose tissue (BAT) of WT and KO mice 6-weeks after HFD (Fig. S5C). However, we have noticed that the differentiated adipocytes from primary SVFs of SAT - a beige fat depot - in *Fam13a* KO mice expressed a higher level of *Ucp1* and *Pgc1a* (Fig. S5A-B). A similar result was also observed in *FAM13A*-deficient human adipocytes differentiated from SGBS preadipocytes (Fig. S6A), which was derived from the subcutaneous fat of an infant with SGBS¹⁵. Together, these results suggest a potential role of *FAM13A* in beige adipocyte differentiation. To fully assess the involvement of *Fam13a* in adipose thermogenic program, future studies on rectal temperature and brown/beige fat markers following the cold challenge, and mitochondrial mass and oxygen consumption following administration of norepinephrine or CL-316,243, could be carried out. We included these results and the related discussion in the revised manuscript as below:

Fig. S5C: Expression of brown fat-specific markers in BAT of mice

Fig. S5A-B: Expression of beige adipocyte markers during adipogenesis of SVFs from SAT of mice

Fig. S6A: Expression of beige markers in differentiating adipocytes of human SGBS

p.7, line 16 : The differentiated subcutaneous adipocytes from *Fam13a* KO mice did not express significantly higher levels of beige adipocyte markers (e.g. *Pgc1a* and *Ucp1*) (Fig. S5).

p.8, line 10: *FAM13A* knockdown by siRNA in SGBS preadipocytes (Fig. 6B) resulted in upregulation of adipogenesis markers e.g. *CEBPA* and *PPARG*) (Fig. 6C) as well as beige adipogenic markers (e.g. *PGC1A* and *UCP1*) (Fig. S6A), in early differentiation stage (Day 5).

p.10, line 29: SAT is known to be able to generate beige adipocytes, a mitochondria-rich and fat-burning adipocyte, while VAT is a classic depot for white adipocytes with a low number of mitochondria and limited ability for adipogenesis¹⁶. SVFs isolated from SAT of *Fam13a* KO mice showed an increased tendency to differentiate into beige adipocytes (Fig. S6).

6. The experiments in Supplemental Figure 4 are also hard to interpret with n's of 3 vs 7. It is also not clear why in panels E and F of this figure there are 3 vs 7 mice, whereas in panel F, which uses the same data plotted a different way there appear to be 7 vs. 6.

- We thank the Reviewer for the opportunity to clarify **Supplementary Figure S4**, which mainly presents the metabolic profiling of female *Fam13a* KO mice on high fat diet (HFD). In this experiment, initially there were 4 vs 7 female mice put on HFD. However, one female WT mouse died during HFD feeding and thus at the time of sacrificing (after 15 weeks of HFD, at 24 weeks of age), the number of animals was different from the baseline (3 WT, 7 KO).
- We apologize for the confusing number of animals in panel E and F, which represent male mice rather than the female mice. This figure has been revised in the manuscript and data on female mice on chow and HFD are presented in Fig. S3 and S4. Data about males on chow and HFD are shown in Fig. 3 and Fig. 4, respectively.

7. The whole section on the ob/ob mice is confusing. The authors appear to be confused about what is the F2 cross in these data, but it is not in the B6 ob/ob mice. This adds little to the story. Also, the data in Sup. Figure 4G appear to come from the website of the Alan Attie lab and are not original to these authors. The authors refer to

the Attie lab website in the methods, but do not indicate these are not their data in the Figure legend. In addition, there are two panels labelled Sup. Figure 4G.

- We agree that the presentation about the data on BTBR^{ob/ob} mice was not clear and its support to our main argument - the role of *FAM13A* in adipogenesis - was not convincing. Therefore, we decreased the emphasis on these data, but kept it as **Fig. S7** to support the negative association of *FAM13A* expression in adipose tissue with genetic-induced obesity, which is consistent with our diet-induced obesity models (**Fig. 7**). As the Reviewer indicated, these mice are not C57BL/6 *ob/ob* mice. The F2 *ob/ob* mice data used in our study are from BTBR^{ob/ob} mouse model.¹⁷ The advantage of using BTBR is that this mouse strain harbors alleles promoting insulin resistance and when crossed with *ob/ob* mice, these mice develop severe type 2 diabetes whereas C57BL/6 *ob/ob* mice compensate by increasing pancreatic insulin secretion and are diabetes-resistant.¹⁷
- We have been collaborating with Alan Attie's lab and the data of F2 mice have been included per their permission. Both Alan Attie and Mark Keller are co-authors on this manuscript. With their assistance, we have

Fig. S7A: Correlation of adipose *Fam13a* expression with obesity in F2 cohort database

Fig. S7B: Association of adipose *Fam13a* expression with proliferation of cells in adipose tissue of BTBR^{ob/ob}

improved and simplified the data presentation. In the revised manuscript we have included new graphs based on data from one parental strain of F2 mice. These data show the expression of *Fam13a* in white adipose tissue is negatively correlated with cell proliferation in adipose tissue, measured by the incorporation of deuterium into newly synthesized DNA (**Fig. S7B**). In addition, in **Fig. S7A**, we present obesity-dependent expression of *Fam13a* in white adipose tissue compared with other tissues. We carefully discussed the relevance of this study in the text () as below:

p.9, line 22: *The negative correlation of adipose *Fam13a* expression with obesity in a genetically obese and diabetic mouse dataset of BTBR^{ob/ob} was demonstrated, through a collaboration with Dr. Alan Attie's lab. However, no correlation was observed between obesity and *Fam13a* expression in other tissues such as hypothalamus, skeletal muscle, liver and islets (Fig. S7A). Additionally, in these mouse models, adipose *Fam13a* expression was negatively associated with proliferation of cells residing in adipose tissue (adipose cellular turnover), measured by the incorporation of deuterium into newly synthesized DNA of adipose tissue (Fig. S7B). These data support our observations in the diet-induced obese mouse models that *Fam13a* expression, specifically in adipose tissue, is negatively associated with obesity.*

8. The most important data, which are not followed up on and which change the interpretation of all other figures, are in Figure 5A. This shows that in adipose tissue virtually all of the *Fam13a* is in the stromovascular fraction, not in the adipocyte. While the authors like to conclude on page 7 that this indicates that *Fam13a* “may affect adipocyte generation”, there is no direct data on this. The authors have not shown that the expression in the SVF is in preadipocytes vs. other cells in this fraction. Even if the *Fam13a* is in preadipocytes, this also means that all differences in expression observed in the previous human and murine studies could simply reflect differences in ratio of preads to adipocytes. The other needed follow-up to this observation would be to study the expression of *Fam13a* through preadipocyte to adipocyte differentiation.

Single-cell RNA-Seq analysis of *Fam13a* expression in mesenchymal stem cells of crude SVFs isolated from adipose tissue of 3-month-old mice (male and female combined, SAT and VAT combined) (<https://tabula-muris.ds.czbiohub.org/>)

Fig. 5A: *Fam13a* expression during adipogenic differentiation of mouse primary SVFs

Fig. 5C: Expression of adipogenic markers during differentiation of mouse primary SVFs isolated from WT vs. *Fam13a* KO mice

Fig. 6B and 6C: Impact of siRNA knockdown of *FAM13A* on adipogenesis of SGBS preadipocytes

Fig. 6D and 6E: Impact of CRISPRi knockdown of *FAM13A* on adipogenesis of SGBS preadipocytes

- We agree with the Reviewer that additional data are needed to support the statement that *FAM13A* may play a role in adipocyte generation. In accordance with the Reviewer's request, we now present the following evidence to demonstrate the expression and functionality of *FAM13A* in adipogenesis: 1) *FAM13A* is expressed in mesenchymal stem cells in SVF, which possess the lineage differentiation ability to adipocytes, indicated by the online resource of single-cell RNA-Seq analysis on SVF of adipose tissue, shown as above; 2) *FAM13A* expression is elevated during the adipogenic differentiation of primary adipose precursor cells (**Fig. 5A**); 3) *FAM13A* may inhibit adipogenesis, suggested by a slight increase in adipogenic capacity of primary precursor cells isolated from SAT of *FAM13A* knockout mice (**Fig. 5C**); 4) *FAM13A* may inhibit adipogenesis in a cell-autonomous manner, shown by the loss-of-function studies in human SGBS preadipocytes that cells with reduced *FAM13A* exhibited enhanced adipogenic potential (**Fig. 6B-E**). We have reflected changes in the text and as below.

p. 7, line 6: After growing SVFs into confluence (i.e. Day 0) and inducing them to adipogenic differentiation, Fam13a expression was found to be elevated during in vitro adipogenesis (Fig. 5A).

p. 7, line 12: Next, the adipogenic potential of SVFs, isolated from SAT of WT and Fam13a KO mice, were assessed by comparing gene expression of adipogenic markers (Pparg, Cebpa, Fabp4), ORO staining of lipid droplets and intracellular triglyceride levels. Results from all these methods showed a similar trend toward induced adipogenesis in Fam13a KO mice, although none of them reached statistical significance (Fig. 5C-E).

p. 8, line 9: FAM13A knockdown by siRNA in SGBS preadipocytes (Fig. 6B) resulted in upregulation of adipogenesis markers e.g. CEBPA and PPARG; Fig. 6C) as well as beige adipogenic markers (e.g. PGC1A and UCP1) (Fig. S6A), in early differentiation stage (Day 5). In addition, 90-95% of FAM13A knockdown was also achieved in preadipocytes by CRISPR-interference (CRISPRi), using 3 sgRNAs targeting FAM13A independently (Fig. 6D). A similar induction of adipogenic markers (CEBPA and PPARG) in early differentiating adipocytes (Day 5; Fig. 6E) also indicated a more active adipogenic capability of FAM13A-deficient preadipocytes.

9. The data on glucose uptake in Figure 5F show no response to insulin even in the control cells. Since differentiated adipocytes usually show a 3-10 fold increase in glucose uptake with insulin stimulation, it appears that the experiment did not work.

- We agree with the Reviewer's assessment that our original glucose uptake assay using 2-NBDG as a fluorescent glucose analog may not be a quantitatively robust method. We have now repeated these experiments using established radiometric assay to quantify differences in glucose uptake. Adipocytes derived from primary adipogenic precursors of WT and KO mice both responded to insulin normally (~3-fold increase compared to basal control), and there was no difference in glucose uptake between *FAM13A* positive and negative adipocytes under both basal and insulin-stimulated conditions. We have updated **Fig 5F**.

10. All of the figure legends are sparse and lacking details.

- Thank you for bringing this up. We agree and have included more details in the legends in the revised version.

Fig. 5F: Glucose uptake in *in vitro*-differentiated adipocytes from SVFs

Reviewer #3 (Remarks to the Author):

Fathzadeh and colleagues provide some evidence from genetic studies and model KO model that FAM13A expression is associated with lower body fat and higher central fat (waist to hip ratio), higher insulin in humans. A similar phenotype of altered fat distribution and a trend toward impaired adipogenesis was observed in F13A KO mice. The manuscript addresses an important gap in knowledge about mechanisms that regulate fat distribution and consequently metabolic risk. While this phenotype is of interest, the data as presented are incomplete /preliminary and likely experiments likely underpowered to draw sufficiently strong conclusions.

- We appreciate the positive comments from the Reviewer about our efforts in addressing a functional role of *FAM13A* in body fat distribution and metabolic risk by connecting the evidence in human genetics to experimental data in animal models. To address the concerns over insufficient and underpowered data, we carried out additional *in vivo*, *ex vivo* and *in vitro* experiments, improved the presentation and the quality of the analyses, and modified the interpretation of the data to more precisely reflect our findings.

Specific concerns:

1. In the mouse model, there is an effect of *Fam13a* KO only in males. Is the low fat phenotype only seen in human males, not females? If not, the translational relevance/limitation of the mouse model should be discussed.

- We thank the Reviewer for asking about this important translational aspect of our research. In humans, the associations of variation in the *FAM13A* locus with WHRadjBMI were more prominent in women than in men – a pattern observed for almost all loci associated with WHRadjBMI.^{3,18} In contrast, in our study of mouse models, the difference between WT and *Fam13a* KO mice in terms of VAT/SAT ratio was stronger in males than in female. The directionality of the association, though, was the same between human and rodent, with low expression of *FAM13A* associated with metabolically favorable adiposity. The limitations of using rodent models for the study of human adiposity may be due to the disparities in terms of anatomical distribution of fat and sexual dimorphism between rodents and humans (Chusyd et al. 2016. *Frontiers in Nutrition*. 3:10). We agree with the Reviewer's suggestion and the translational relevance as well as limitations of the mouse models are further discussed in the revised manuscript and as outlined below:

Sexual dimorphism in fat distribution in humans (p. 12, line 11): *In human GWAS, the WHRadjBMI locus harboring FAM13A variants showed a stronger effect in women than in men. Sex dimorphism in the genetic regulation of fat distribution may be related to the difference in body fat shape between the sexes. Women tend to deposit more subcutaneous fat around hips and legs, whereas men tend to accumulate more visceral fat around the waist. During menopause, body shape of women can shift toward a more android type, including increased abdominal fat deposition. The sexual dimorphism in fat distribution could be, in part, due to the influence of sex hormones.*

Sexual dimorphism in fat distribution in mice (p.12, line 18): *In our study of mouse models, equal metabolic phenotyping experiments were performed on male and female mice. While in male mice there was a difference in VAT/SAT between WT and *Fam13a* KO mice upon HFD, no significant phenotypic effects of *Fam13a* deficiency were observed in female mice. Fat accumulation in rodents is also influenced by sex hormones. Unlike their male counterparts, C57BL/6 female mice are relatively resistant to HFD-induced obesity, but this protection has been shown to be removed by ovariectomizing the female mice. In our study, we did not perform experiments in ovariectomized female mice upon HFD.*

Translational relevance/limitation of mouse models for the study of human fat distribution (p. 12, line 26): *Although current translational investigation on fat distribution and its metabolic consequences relies on rodent models, they may not faithfully mimic all aspects of human fat biology due to differences in fat location and function between species. The notable difference between human and mice regarding the sex dimorphism observed in association of*

FAM13A with fat distribution (e.g. stronger negative association in women and male mice) may be related to the species-specific anatomical location of fat depots¹⁹. For SAT, humans deposit peripheral fat into upper body (abdominal region) and lower body (around hips and thighs), while rodents accumulate subcutaneous fat anteriorly and posteriorly. This difference between human and mouse in distribution of SAT affects local proximity to reproductive hormones, which govern fat pad development and function²⁰. For VAT, humans harbor visceral fat in the intra-abdominal region, while the largest visceral fat in rodents is the perigonadal pad, which does not have a human visceral analog. In our study, we did not rigorously evaluate mesenteric fat in mice – the most analogous pad to human visceral fat because of its direct access to the portal vein – due to difficulties in its surgical removal and separation from surrounding vessels¹⁹.

2. Figure 3E and F: Adipocyte number and size per image are calculated (E). To enable the reader to evaluate the data and most importantly to assess whether the lower adipocyte size in SAT and the greater size in VAT affect the number of adipocyte per depot (i.e. group difference in hyperplasia vs hypertrophy). Pixels should be not be presented – they should be converted to microns and a scale bar should be shown on the micrographs shown in Figure F. Furthermore and importantly, the number of adipocytes per depot in each animal should be calculated by dividing depot weight by mean adipocyte weight (ug lipid/cell calculated from mean volume and the specific gravity of triglyceride (0.915)) and is easily calculated by standard formulas in the literature (and assuming a % lipid in the depot (~85% in VAT, ~75% in SAT) if a direct measure is not available).

- We appreciate this comment that was also mentioned by Reviewer #2 (comment 4). In the revised manuscript, we have reanalyzed and extended adipocyte histomorphometry data in WT and *Fam13a* KO mice. First, the conversion was done from pixel to microns and the scale bar in micrometers is shown on each image (Fig. 3G). Second, the total number of adipocytes per fat pad was calculated by dividing the weight of whole tissue by the mean weight of an adipocyte, as suggested by the Reviewer (Fig. 3H). We described the method into details in the text (p. 15, line 520-533), presented the histological pictures with proper units in Figure 3G, calculated the total number of adipocytes per depot (Fig. 3H), and displayed the difference in adipocyte size distribution between WT and KO mice in Figure 3I-J.

Methods for quantification of adipocyte size and number:

p. 17, line 6: Briefly, microphotographs of H&E stained adipose tissue blocks were acquired from an optical microscope (Zeiss Axioplan2) at 20x magnification and images were captured with a Leica DC500 camera and a NIS Elements software. ImageJ software with the H&E color deconvolution plugin was used to determine the size of the adipocytes and the number of adipocytes within a certain diameter range by two individuals in a double blinded manner. 4 fields of view for SAT slides and 6 fields of view for VAT slides were quantified. SAT and VAT from 7 animals per group were analyzed. A total of 2000-2200 cell diameters were measured by Adiposoft, an automated software for analysis of adipose tissue cellularity as established¹³. The quantification unit of pixels were converted to microns by Photoshop CS6 (1 pixel=0.3953 μm at 20x objective). Adipocyte number per fat depot was calculated by dividing the total tissue weight by the mean adipocyte weight¹⁴. Total tissue weight was measured directly after tissue dissection, and the mean adipocyte weight was calculated by multiplying the mean adipocyte volume by the triglyceride density (0.915 g/L). The mean volume of adipocyte was calculated from the diameters measured above by applying the following formula: $V = 4/3 \pi (D/2)^3$.

3. It is stated that FAM13a expression was negatively correlated with ‘adipose cellular proliferation’ but no direct measurements of the proliferation of adipose progenitors was presented. Changes in the cell cycle pathway may suggest this, but no conclusion can be drawn without direct measurements.

- We agree with the Reviewer that we did not provide direct measurements to test *Fam13a*'s role in proliferation of adipose progenitors. Thus, we have deleted or modified all misleading statements suggesting a causal role of *FAM13A* in the proliferation of adipocyte precursors.
- The experiments leading to the original statement that “*FAM13A* expression was negatively correlated with adipose cellular proliferation” was conducted in the lab of co-author Alan Attie, who used the deuterium incorporation into DNA to measure cell proliferation in adipose tissue *in vivo*. In the genetically modified obese and diabetic model of *BTBR^{ob/ob}*, *Fam13a* expression specifically in adipose tissue was negatively associated with obesity, as well as the cell proliferation in adipose tissue. Nevertheless, we agree that this is a correlation rather than causation study, and the measurement of cellular proliferation in adipose tissue is not limited to adipogenic progenitors. As also suggested by Reviewer#2, we removed the overinterpretation of this set of data from the Results section and carefully discussed its relevance to our study in the text as follows:

p.9, line 22: The negative correlation of adipose Fam13a expression with obesity in a genetically obese and diabetic mouse dataset of BTBR^{ob/ob} was demonstrated, through a collaboration with Dr. Alan Attie’s lab. However, no correlation was observed between obesity and Fam13a expression in other tissues such as hypothalamus, skeletal muscle, liver and islets (Fig. S7A). Additionally, in these mouse models, adipose Fam13a expression was negatively associated with proliferation of cells residing in adipose tissue (adipose cellular turnover), measured by the incorporation of deuterium into newly synthesized DNA of adipose tissue (Fig. S7B). These data support our observations in the diet-induced obese mouse models that Fam13a expression, specifically in adipose tissue, is negatively associated with obesity.

4. It would be important to at list the transcripts associated with pathways listed as differentially regulated (Fig S4 I and J) - what is magnitude of differences in transcripts at the leading edge of the differentially regulated pathways.

- We thank the reviewer for the suggestion that we focus on specific transcripts. We chose to streamline the presented information in the supplementary figures, and following edits, we have not included the panels I and J of **Supplementary Figure 4**. However, we now highlight important genes, the magnitude of their differential expression, and the pathways they are associated with in the Results section with the following text:

p. 9, line 7: The DE genes included several genes previously associated with fat cell biology, including Klf14^{21,22} (Fold change 2.3), Agpat2^{23,24} (FC 0.51), Slc7a10²⁵ (FC 0.65), Vegfa²⁶, Celsr2²⁷ (FC 2.14) and Fgfr2²⁸ (FC 0.49). Further, we observed an overrepresentation of genes in adipogenesis pathways among genes overexpressed in KO, corroborating evidence from our in vitro and in vivo data suggesting that Fam13a deficiency induces subcutaneous adipogenesis. We also observed overrepresentation of genes in a NAD⁺ salvage pathway among underexpressed genes; this pathway has been discussed previously in SAT in the context of obesity and weight loss, and intracellular levels of NAD⁺ have a role in maintaining the metabolic status of the cell²⁹. In addition to adipogenesis and NAD⁺ salvage pathways, we found an overrepresentation of overexpressed genes in fatty acid beta oxidation and TCA cycle pathways in SAT from KO animals.

5. Figure S5B shows a low magnification image of differentiated adipose progenitors from iWAT. It appears that fewer cell differentiated into adipocytes in the KO fat. However, no quantitation of lipid/DNA or the reproducibility of this apparent finding is provided. Also, only one late marker of differentiation (Fabp4) was measured and this is inadequate. The figure legends do not provide adequate detail to evaluate the rigor or reproducibility of the findings (e.g. number of independent observations/number of preparations from different mice and their age).

- We thank the Reviewer for the comments. We have now modified the figure legend to provide adequate experimental details. Data presented here was obtained from female mice. Since we did not observe significant *in vivo* or *ex vivo* phenotypes in female *Fam13a* KO mice, we concentrated experiments on male mice during the revision process and repeated the adipogenesis assay of primary progenitors to meet the Reviewer’s suggestion. We assessed the adipogenic potential of progenitor cells by comparing the results of several complementary methods (**Fig. 5C-E**): 1) quantitatively measuring the expression level of adipogenic markers (e.g. PPAR γ , CEBP α , FABP4); 2) morphologically evaluating ORO staining followed by normalization of lipid staining against DNA

content; 3) quantitatively measuring TG content in differentiated adipocytes. All these data are shown below addressing this reviewer's comments regarding **Figure 5E** in the original manuscript.

6. The header on page 6, last paragraph states 'Fam13A 'tends to induce subcutaneous adipogenesis' – additional and more rigorous experiments are needed to even support this weak statement. It is interesting that Fam13a is higher in SVF than floating adipocytes, but freshly isolated SVF is a mixed population of cells (Fig 5A). A time course of changes in *Fam13a* gene expression from confluent cultures (mainly adipose progenitors/fibroblasts) until D10 is needed to address this question more directly.

- We thank the Reviewer for the suggestion. We have now included the data to show the expression pattern of *Fam13a* from confluent cultures (i.e. Day 0) until Day 10 (**Figure 5A**).

7. **Figure 5E provides oil red O staining in WT and KO cultures. However, this is poorly quantitative measure and no denominator (i.e. number of cells per plate by DNA or some other measure) is provided. Further, the differences in *Fabp4* expression (a marker of differentiation) were not statistically significant.**

- We agree with the Reviewer that ORO staining of lipid droplets is only a semi-quantitative method to evaluate adipogenesis. Hence, upon the Reviewer's suggestion, not only have we repeated the ORO staining by normalizing the lipid staining against the DNA content of the same culture well (**Figure 5E**); but we have also presented the measurement of adipogenic markers and intracellular TG content as a more accurate method to quantify the degree of adipogenesis (**Figure 5C and 5H**). The Reviewer is correct to point out that we did not observe a statistically significant difference in adipogenic capacity of WT and *Fam13a* KO precursors, based on the results from a comprehensive set of experiments mentioned above. However, there was a trend towards an increase in adipogenesis of *Fam13*-deficient cells consistently observed across a spectrum of different experiments: adipogenic marker measurement, TG quantification and ORO staining. We think these data still have a value in offering insightful information about the regulatory role of *Fam13a* in subcutaneous adipocyte differentiation, even if not statistically significant, but we have modified the text to reflect this as follows.

*p. 7, line 12: Next, the adipogenic potential of SVFs, isolated from SAT of WT and *Fam13a* KO mice, were assessed by comparing gene expression of adipogenic markers (*Pparg*, *Cebpa*, *Fabp4*), ORO staining of lipid droplets and intracellular triglyceride levels. Results from all these methods showed a similar trend toward induced adipogenesis in *Fam13a* KO mice, although none of them reached statistical significance (**Fig. 5C-E**). The differentiated subcutaneous adipocytes from *Fam13a* KO mice also expressed a marginally but not significantly higher level of beige adipocyte markers (e.g. *Pgc1a* and *Ucp1*; **Fig. S5**). These subtle but noticeable increases in subcutaneous adipocyte differentiation may be relevant to the distribution shift toward an increased number of smaller adipocytes in SAT of male *Fam13a* KO mice observed earlier (**Fig. 3G-H**).*

8. **Figure 5F also shows non-significant trends for basal and glucose uptake in iWAT from the control and KO mice. Unfortunately, the data are presented as "fold over basal of WT" so can't be properly evaluated with respect at actual rates, and are confounded by the apparently higher number or size of differentiated adipocytes in the KO. A denominator like triglyceride content/well or protein is needed. Further, the insulin effect on glucose uptake is far smaller than expected, questions the preparation or the assay. The number of observations/mice/cultures is not provided in the figure legends (and should be) – but it appears the study was underpowered.**

Fig. 5A: *Fam13a* expression during adipogenic differentiation of mouse primary SVFs

Fig. 5E: Semi-quantification of ORO in adipocytes differentiated from SVFs

Fig. 5C: Expression of adipogenic markers during differentiation of mouse primary SVFs isolated from WT vs. *Fam13a* KO mice

Fig. 5H: Quantification of triglyceride in adipocytes differentiated from SVFs

- We agree with the Reviewer's comments and have now conducted a new set of experiments to address these concern. We performed radiometric glucose uptake assay, a highly sensitive quantification method, to re-evaluate the difference in glucose deposit into WT and *Fam13a* KO adipocytes differentiated from primary progenitors. No significant difference was observed, but there was a trend toward an increase in glucose uptake in both basal and insulin-treated *Fam13a* KO adipocytes, albeit with a normal insulin response in both WT and KO cells (Fig. 5G). Also, to separate the confounding effects of *Fam13a* on glucose uptake from adipocyte generation, we conducted the radiometric glucose uptake assay on explants of SAT isolated from WT and *Fam13a* KO mice, and observed a similar directional trend (Fig. 5H). As suggested by the Reviewer, we now present the results by normalizing the radioactivity counts against the protein content or the tissue weight of the same sample (Fig. 5G-H), and modified the figure legend by providing adequate experimental details (n=3 mice per WT or KO; n=3 culture wells per mouse).

Fig. 5G: Glucose uptake in *in vitro*-differentiated adipocytes from SVFs

Fig. 5H: Glucose uptake in *ex vivo* adipose tissue explants

9. The final sentence of the prior paragraph also states 'data from F2ob/ob mice support a role of *Fam13a* in adipose turnover and fat plasticity'. No direct experimental data are provided support this statement either.

- We apologize for the misleading statement. As mentioned in previous answers above, we agree that the whole section of F2 ob/ob mice was misleading and the interpretation was not precise. In this study, we have focused on studying the impact of *Fam13a* in adipogenesis, which is only one process - among many others - in contributing to "adipose turnover". We agree that we have no direct experimental data to support a comprehensive role of *Fam13a* in all processes of "adipose turnover and fat plasticity". Thus, we corrected or deleted all statements regarding this issue throughout the paper.

Reviewer #4 (Remarks to the Author):

In “The human lipodystrophy-like gene FAM13A affects fat distribution and function,” Dr. Fathzadeh and colleagues first focus on genetic variation in the FAM13A locus -- identified via genome-wide associations of fat distribution, insulin resistance, and other cardiometabolic phenotypes -- and then proceed to study the potential biological mechanism of changes in FAM13A on fat distribution.

First, I would like to commend the authors on a very clearly written manuscript. The writing is crisp, succinct, avoids the passive voice, and is overall very easy to follow. The figures are equally clear and can stand alone as helpful pieces of narrative.

- We thank the Reviewer for these kind words and interest in our study.

I do have comments on some of the analytic pieces, which I will further outline below:

Major comments

[1] The biggest piecing piece for me is how, precisely, the authors get from non-coding variants in FAM13A to focusing all of their analyses on strictly the FAM13A gene. GWAS loci are notoriously complex due to LD, gene density in the region, and the fact that associated SNPs themselves are non-coding. While I understand that the SNPs the authors have focused on lie in FAM13A, this is no guarantee that FAM13A is the gene to focus on (the FTO locus in BMI is perhaps the best argument for this; intronic FTO variants seem to influence regulation of IRX3 and IRX5 -- genes that are quite some distance away -- which in turn affects adiposity). Did the authors do any fine-mapping to figure out if FAM13A was the gene to focus on? Or look to see if this locus falls in a topologically associating domain? What other evidence did the authors consider before ultimately deciding that FAM13A was the single gene they wanted to focus on?

- This is an insightful comment that refers to the challenge of identifying disease-relevant genes based on GWAS associations. In this study, we aligned multiple observations to conclude that FAM13A was most likely to be the putative causal gene underlying the GWAS associations with intronic FAM13A variants. Broadly, these include Bayesian fine-mapping, tissue-specific eQTL mapping and colocalization with GWAS associations, and analysis of FAM13A expression in human adipose tissue and its correlation with traits characteristic of body fat distribution. We also use data from F2ob/ob mice to show that FAM13A expression is correlated with adipose cellular proliferation. We broadly explain how we focused our analyses that eventually identified FAM13A as a putative causal gene in the Results section as follows:

p.4, line 11: We performed fine-mapping and colocalization analyses to explore several lines of evidence suggesting that non-coding GWAS-associated variants in the FAM13A locus have functional impact on FAM13A in adipose tissue. First, variants in the FAM13A locus, including rs3822072, rs1377290 and rs9991328, have been reported as lead variants within the locus across GWAS studies of IR-related traits such as body fat percentage, WHRadjBMI, fasting insulin levels. Next, these variants were associated with FAM13A expression in subcutaneous, but not in visceral adipose tissue (VAT; Fig. 1) in data from the Genotype-Tissue Expression project (GTEx v7). Finally, variants in the locus showed some evidence of enrichment in regions annotated with the H3K27ac epigenetic mark in adipose nuclei as compared to other tissues (Online Methods, Fig. 1).

- The relevant cis-eQTL data support a potential role of FAM13A in adipose tissue, and additionally display a distinctive genetic effect on FAM13A expression in SAT versus VAT. Specifically, we intended to convey that rs9991328, which is also associated with multiple phenotypes including HDL, TG and WHRadjBMI, is a cis-eQTL only for FAM13A and not for any other gene in the region. The cis-eQTL analysis was performed using previously established parameters⁵. Briefly, variants within 1Mb of the transcription start site of each gene, with a minor

allele frequency > 0.01 and minor allele observed in at least 10 samples in the tissue of interest were tested for association with the expression of each gene. In this analysis, rs991328 was a cis-eQTL solely for FAM13A in SAT but not in VAT as depicted in **Figure 1**. Colocalization results further corroborated the evidence.

- Also, we did not observe rs991328 to be a significant trans-eQTL for any genes in either SAT or VAT. Significant trans-eQTL association data is publicly available on the GTEx portal datasets page (<https://gtexportal.org/home/datasets>) under the GTEx Analysis V6p section. Taken together with cis-eQTL data and the colocalization results, this suggested that *FAM13A* was most likely the causal gene in the region. We highlight these details in the Results section using the text below:

*p. 4, line 26: In SAT, rs991328 showed a robust association with FAM13A expression ($P=1e-08$) and was not a cis-eQTL (expression quantitative trait locus) for any other genes with transcription start sites within 1Mb of the variant. In contrast, rs991328 showed a weaker association with FAM13A expression ($P=1.5e-05$) in VAT. The variant was also not a significant trans-eQTL for any genes in adipose tissue (data from GTEx v6p). To further evaluate whether FAM13A expression mediates the genotype-phenotype associations at this locus, we performed colocalization analysis of FAM13A gene expression in SAT, VAT, liver, and skeletal muscle from GTEx (v7) with seven IR-related GWAS traits using eCAVIAR³⁰. We observed colocalizations (posterior probability > 0.8; lead variant rs991328) between SAT FAM13A expression and fasting insulin adjusted for BMI, WHRadjBMI, triglycerides and HDL-cholesterol. We did not observe similar colocalizations or strong associations between variants in the locus with expression of any other gene in any other investigated tissues, suggesting that FAM13A plays a specific role in SAT (**Fig. 1 and Fig. S1**), and that this is the causal gene underlying the GWAS signal.*

- Although we did not explicitly begin with an exploration of topologically associated domains, the cis-eQTL analysis, which considers variants with 1Mb of a gene's transcription start site, did not identify any other nearby genes that have expression associated with these variants.
- In the human cohort METSIM, increased FAM13A expression was correlated with deleterious metabolic phenotypes. Further, in F2ob/ob mice, increased Fam13a expression was negatively correlated with adipose cellular proliferation. Taken together, these lines of evidence from several different datasets led us to focus our further *in vitro* and *in vivo* experiments on the function of *FAM13A*.

[2] The analyses presented in the first paragraph of the results, in my opinion, need a bit more fleshing out to feel truly robust. First, it's not clear how many SNPs you are actually looking up. Is it only the two SNPs mentioned in the paragraph before this? Or SNPs that span the full locus?

- We thank the Reviewer for identifying the lack of clarity in how we presented SNPs associated with relevant GWAS traits and FAM13A expression across tissues. While several SNPs at the FAM13A locus are reported as associated with a wide range of metabolic phenotypes, most of these are very strongly linked, suggesting that they tag the same casual variant(s) that in turn drive the association. In order to better understand the contribution of individual SNPs to the association signal, we now performed Bayesian fine-mapping to identify 99% credible sets of variants that are putatively responsible for driving the association signal. Throughout fine-mapping analyses both on SAT eQTL and GWAS traits, rs991328 had either the highest posterior probability of being causal or was among the top 5 most important SNPs. This led us to focus our efforts on rs991328 and use it as the lead SNP for colocalization results. We highlight this analysis in the Results section using the text below:

p. 4, line 19: Bayesian finemapping analysis of the adipose subcutaneous expression association signal using the FINEMAP algorithm revealed that rs991328, which is associated with increased gene expression ($b = 0.22 \pm 0.03$, $p=1e-08$), had the highest posterior probability of driving the signal (finemapping posterior probability = 0.66) among all the variants in the region; we retained rs991328 as the lead variant for further analyses. This variant is located within an intronic region ~3.2Mb away from the transcription start site of FAM13A. The most commonly reported GWAS lead variant, rs3822072, was among the top 5 variants in the finemapping analysis (posterior probability = 0.06).

- The other SNP we mention is rs1377290, with which we performed a PheWAS in the UK Biobank. The variant rs1377290 was selected for the following reasons: (1) It has an LD R² of 1.0 with rs991328, which has the highest Bayesian fine-mapping posterior probability of being causal for the association signals in SAT eQTL analysis and several GWAS traits we considered, (2) It is directly genotyped in the UK Biobank. To better clarify these points, we edited the text in the Results section to read as follows:

p. 5, line 7: *We also performed a phenome-wide association study (PheWAS) in 337,536 individuals from the UK Biobank^{11,12} using the directly genotyped variant rs1377290 as a proxy for the imputed variant rs9991328 (LD R2 = 1.0). The rs1377290 T allele was associated with increased WHR, and decreased trunk/body fat percentage, mean platelet volume, and BMI (Fig. 2A). There were no other significant associations across the phenotypic spectrum.*

Next, to say that a SNP is an eQTL is, on its own, rather unsurprising given the ubiquity of eQTLs in the genome. Would there be a way to assess whether this particular eQTL finding is non-random? How likely is it to pull a SNP from the genome and fall upon an eQTL in adipose tissue?

- It is true that eQTLs are ubiquitous across the genome. False discoveries in the cis- and trans-eQTL analyses are accounted for by the multiple testing correction methods employed in each analysis. Specific parameters used in the analysis, such as restricting testing for cis-eQTLs to within 1Mb of the transcription start site of each gene, are employed to further increase power and are based on specific goals, such as to find local genetic variation influencing gene expression. Within the parameters chosen for each analysis, genome-wide significant eQTLs are associated with genetic variation more than random chance, and we therefore consider them as valid discoveries. Further details are provided in previously published work⁵.

Lastly, when you say that the SNP and some LD partners fall into an H3K27ac epigenetic mark tagging enhancers in adipose nuclei, could you assess whether this enrichment is non-random? For example, this paper (<https://www.nature.com/articles/ng.3437>) gives a nice method for deciding if SNPs in a locus are enriched in a particular annotation. It seems helpful to show this, particularly if the footprint of the annotation is large and it is 'easy' for a SNP to fall into the annotation? Do you show lung and pancreas in Figure 1 to demonstrate a negative comparator? The reason that data also appears isn't clear.

- We thank the Reviewer for suggesting that we should use a principled statistical method to assess enrichment of variants driving the association signal within epigenomic annotations. We now performed an enrichment analysis for H3K27ac, which tags active promoters, in which we first defined a 99% credible set of putatively causal variants using FINEMAP posterior probabilities. We then assessed the number of variants that overlapped the annotation in adipose nuclei (test statistic). To establish an empirical null distribution for this test statistic, we randomly shifted the annotations to the left or right by a maximum of 500kb and estimated the proportion of random samples in which the number of credible set variants that overlap with the annotations was greater than or equal to the test statistic. While insignificant by a slim margin at a nominal threshold, adipose nuclei showed a tendency towards enrichment ($p = 0.07$) compared with no evidence in other tested tissues, which are highlighted as negative comparators (p values in these other tissues). Further, visual inspection of H3K27ac annotation as shown in the bottom panel of Figure 1 suggests stronger concentration of this epigenomic annotation in the locus compared with the negative comparators. Due to the enrichment analysis potentially being influenced by the choice of fine-mapping approach and the number of variants included in the credible set, we chose to report this overlap due to its suggestive low p -value. We have included this approach, which is similar to the one in the paper shared by this Reviewer (see Online Methods p.14, line 23):

We performed a statistical enrichment analysis for epigenomic marks in which we defined a 99% credible set consisting of 16 linked variants at the FAM13A locus that are most likely to be responsible for driving the association at this locus. We used FINEMAP posterior probabilities to define this set – variants were included as long as their cumulative fine-mapping posterior probability for the SAT eQTL signal was ≤ 0.99 . To perform the enrichment analysis, shifted the annotations randomly by a maximum of 500kb and estimated the proportion of randomizations in which the number of credible set variants that overlap with the annotations was greater than or equal to the test statistic.

[3] The colocalisation analysis presented in the next paragraph is a far more compelling piece of data than the eQTL lookup. This analysis made me wonder if the authors had thought of running Bayesian fine-mapping in this region, before embarking on the rest of their analysis, so as to better understand which SNPs might be the likely causal ones?

- We thank the Reviewer for pointing out that fine-mapping is an important step to identify putatively causal SNPs. and for suggesting Bayesian fine-mapping. which we have performed prior to colocalization using the

FINEMAP algorithm. We used the variant with the highest posterior probability in SAT eQTL and a number of GWAS traits (rs9991328) for further colocalization analysis. We also used fine-mapping posterior probabilities to establish a 99% credible set of variants to assess enrichment with epigenomic annotations, as described above. To better describe to the viewer our approach, we have added the following added text quoted above explaining the colocalization results and choice of lead variant in the Results section:

p. 4, line 19: Bayesian finemapping analysis of the adipose subcutaneous expression association signal using the FINEMAP algorithm revealed that rs9991328, which is associated with increased gene expression ($b = 0.22 \pm 0.03$, $p = 1e-08$), had the highest posterior probability of driving the signal (finemapping posterior probability = 0.66) among all the variants in the region; we retained rs9991328 as the lead variant for further analyses. This variant is located within an intronic region ~3.2Mb away from the transcription start site of FAM13A. The most commonly reported GWAS lead variant, rs3822072, was among the top 5 variants in the finemapping analysis (posterior probability = 0.06).

[4] I am not a mouse expert, so I will leave comment on the mouse work to other reviewers. However, I did realise that most of the mice studied were male. Fat distribution -- as shown from GWAS -- has a strong sex dimorphic signature; some biology appears to be truly sex specific. Did the authors check to see if FAM13A has a dimorphic signal? Did they consider this when selecting mice for experimentation? Could the sex dimorphism in fat distribution influence how generalisable these results are to either men or women?

- We thank the reviewer for pointing out the sex- and species-specific effects of *FAM13A* on fat distribution. In our study of mice, equal levels of experimental tests were performed in female as in male. We did not observe significant difference between female WT and *Fam13a* KO mice in any of the metabolic phenotypes tested. We included all the data on female mice in Fig. S3 and S4. In humans, however, the association of *FAM13A* variants with fat distribution was more prominent in women than in men – a pattern observed for almost all loci associated with WHRadjBMI.^{3,18} The directionality of the association, though, was the same between humans and rodents, with low expression of *FAM13A* associated with favorable adiposity. The differences in the anatomical location and function of various fat depots between species may limit the translational value of rodent models to humans. We cannot generally translate the results seen in male mice to either men or women. For this reason, we added in extended discussions regarding sexual dimorphism of fat distribution in both human and mice, and urged careful experimental plan and interpretation when drawing parallel conclusions from rodents to humans.

Sexual dimorphism in fat distribution in humans (p. 12, line 12): *In human GWAS studies, the WHRadjBMI locus harboring FAM13A variants showed a stronger effect in women than in men. Sex dimorphism in the genetic regulation of fat distribution may be related to the difference in body fat shape between the sexes. Women tend to deposit fat most frequently in subcutaneous fat around hips and legs, whereas men tend to accumulate more visceral fat around the waist. During menopause, body shape of women can shift toward a more android type, including increased fat deposition around the abdomen. The sexual dimorphism in fat distribution could be, in part, due to the influence of sex hormones.*

Sexual dimorphism in fat distribution in mice (p.12, line 18): *In our study of mouse models, equal levels of metabolic phenotyping experiments were performed on male and female mice. While in male mice there was a difference in VAT/SAT between WT and *Fam13a* KO mice upon HFD, no significant phenotypic effects of *Fam13a* deficiency were observed in female mice. Fat accumulation in rodents is also influenced by sex hormones. Unlike their male counterparts, C57BL/6 female mice are relatively resistant to HFD-induced obesity, but this protection has been shown to be removed by ovariectomizing the female mice. In our study, we did not perform experiments in ovariectomized female mice upon HFD.*

Translational relevance/limitation of mouse models for the study of human fat distribution (p. 12, line 26): *Although current translational investigation on fat distribution and its metabolic consequences relies on rodent models, given differences in fat location and function between species they may not faithfully mimic all aspects of human fat biology. The notable difference between human and mice regarding the sex dimorphism observed in *FAM13A* association to fat distribution (e.g. stronger negative association in women and male mice) may be related to the species-specific anatomical location of fat depots¹⁹. For SAT, humans deposit peripheral fat into upper body (abdominal region) and lower body (around hips and thighs), while rodents accumulate subcutaneous fat anteriorly*

and posteriorly. This human/mouse difference in distribution of SAT affects local proximity to reproductive hormones, which govern fat pad development and function²⁰. For VAT, humans harbor visceral fat in the intra-abdominal region, while the largest visceral fat in rodents is the perigonadal pad, which does not have a human visceral analog. In our study, we did not rigorously evaluate mesenteric fat in mice – the most analogous pad to human visceral fat because of its direct access to the portal vein – due to difficulties in its surgical removal and separation from surrounding vessels¹⁹.

Minor comments

[1] On the whole, I wanted more data to be directly reported in the text. For example, when associations or correlations are mentioned, stating the association effect and p-value or the value of the correlation would be tremendously helpful. The writing feels a bit too qualitative at the moment (with much of the numerical bits show in the figures). Textual bits that focus on the quantitative details would be very helpful.

- We have included several datapoints in the text, including PheWAS effect sizes and p-values, colocalization posterior probabilities, FINEMAP posterior probabilities, and correlation p-values.

[2] Similar to my point above, including details such as sample sizes in the mouse analyses would be helpful.

- Sample sizes for each of the mouse analyses are now included in the corresponding figure legends.

[3] I have, as a reviewer, become a real stickler for authors placing data and code online in an accessible place. Will the data and code for this project be released? This is a crucial detail to make scientific work more open, transparent, and reproducible.

- We thank the Reviewer for pointing out that availability of data and methods is critical to reproducible research, and we agree with this wholeheartedly. All data will be available for download from the GEO database. Methods used to perform fine-mapping, colocalization and differential expression analysis as well as relevant documentation have been made publicly available by the authors of each method. Specific code used in this study will be deposited in GitHub.

References:

- 1. Chu, A.Y. *et al.* Multiethnic genome-wide meta-analysis of ectopic fat depots identifies loci associated with adipocyte development and differentiation. *Nat Genet* **49**, 125-130 (2017).
- 2. Sun, G. *et al.* Comparison of multifrequency bioelectrical impedance analysis with dual-energy X-ray absorptiometry for assessment of percentage body fat in a large, healthy population. *Am J Clin Nutr* **81**, 74-8 (2005).
- 3. Ji, Y. *et al.* Genome-Wide and Abdominal MRI Data Provide Evidence That a Genetically Determined Favorable Adiposity Phenotype Is Characterized by Lower Ectopic Liver Fat and Lower Risk of Type 2 Diabetes, Heart Disease, and Hypertension. *Diabetes* **68**, 207-219 (2019).
- 4. Tang, J. *et al.* Obesity-associated family with sequence similarity 13, member A (FAM13A) is dispensable for adipose development and insulin sensitivity. *Int J Obes (Lond)* (2018).
- 5. Consortium, G.T. *et al.* Genetic effects on gene expression across human tissues. *Nature* **550**, 204-213 (2017).
- 6. Laakso, M. *et al.* The metabolic syndrome in men study: a resource for studies of metabolic and cardiovascular diseases. *Journal of lipid research* **58**, 481-493 (2017).
- 7. Hägg, S. *et al.* Multi-Organ Expression Profiling Uncovers a Gene Module in Coronary Artery Disease Involving Transendothelial Migration of Leukocytes and LIM Domain Binding 2: The Stockholm Atherosclerosis Gene Expression (STAGE) Study. *PLoS Genetics* **5**, e1000754 (2009).
- 8. Yaghoobkar, H. *et al.* Genetic evidence for a normal-weight “metabolically obese” phenotype linking insulin resistance, hypertension, coronary artery disease and type 2 diabetes. *Diabetes*, DB_140318 (2014).
- 9. Lotta, L.A. *et al.* Integrative genomic analysis implicates limited peripheral adipose storage capacity in the pathogenesis of human insulin resistance. *Nat Genet* **49**, 17-26 (2017).
- 10. Pulit, S.L. *et al.* Meta-analysis of genome-wide association studies for body fat distribution in 694 649 individuals of European ancestry. *Human molecular genetics* **28**, 166-174 (2018).
- 11. Sudlow, C. *et al.* UK biobank: an open access resource for identifying the causes of a wide range of complex diseases of middle and old age. *PLoS medicine* **12**, e1001779 (2015).
- 12. Rao, A.S. *et al.* Large-Scale Phenome-Wide Association Study of PCSK9 Variants Demonstrates Protection Against Ischemic Stroke. *Circ Genom Precis Med* **11**, e002162 (2018).
- 13. Galarraga, M. *et al.* Adiposoft: automated software for the analysis of white adipose tissue cellularity in histological sections. *J Lipid Res* **53**, 2791-6 (2012).
- 14. Bourgeois, F., Alexiu, A. & Lemonnier, D. Dietary-induced obesity: effect of dietary fats on adipose tissue cellularity in mice. *Br J Nutr* **49**, 17-26 (1983).
- 15. Fischer-Posovszky, P., Newell, F.S., Wabitsch, M. & Tornqvist, H.E. Human SGBS cells - a unique tool for studies of human fat cell biology. *Obes Facts* **1**, 184-9 (2008).
- 16. Rosen, E.D. & Spiegelman, B.M. What we talk about when we talk about fat. *Cell* **156**, 20-44 (2014).
- 17. Clee, S.M., Nadler, S.T. & Attie, A.D. Genetic and genomic studies of the BTBR ob/ob mouse model of type 2 diabetes. *American journal of therapeutics* **12**, 491-498 (2005).
- 18. Shungin, D. *et al.* New genetic loci link adipose and insulin biology to body fat distribution. *Nature* **518**, 187-96 (2015).
- 19. Chusyd, D.E., Wang, D., Huffman, D.M. & Nagy, T.R. Relationships between Rodent White Adipose Fat Pads and Human White Adipose Fat Depots. *Front Nutr* **3**, 10 (2016).

- 20. Fried, S.K., Lee, M.J. & Karastergiou, K. Shaping fat distribution: New insights into the molecular determinants of depot- and sex-dependent adipose biology. *Obesity (Silver Spring)* **23**, 1345-52 (2015).
- 21. Small, K.S. *et al.* Regulatory variants at KLF14 influence type 2 diabetes risk via a female-specific effect on adipocyte size and body composition. *Nat Genet* **50**, 572-580 (2018).
- 22. Civelek, M. *et al.* Genetic Regulation of Adipose Gene Expression and Cardio-Metabolic Traits. *Am J Hum Genet* **100**, 428-443 (2017).
- 23. Agarwal, A.K. *et al.* AGPAT2 is mutated in congenital generalized lipodystrophy linked to chromosome 9q34. *Nature genetics* **31**, 21 (2002).
- 24. Cautivo, K.M. *et al.* AGPAT2 is essential for postnatal development and maintenance of white and brown adipose tissue. *Mol Metab* **5**, 491-505 (2016).
- 25. Ussar, S. *et al.* ASC-1, PAT2, and P2RX5 are cell surface markers for white, beige, and brown adipocytes. *Sci Transl Med* **6**, 247ra103 (2014).
- 26. Cao, Y. Angiogenesis modulates adipogenesis and obesity. *J Clin Invest* **117**, 2362-8 (2007).
- 27. Warren, C.R. *et al.* Induced Pluripotent Stem Cell Differentiation Enables Functional Validation of GWAS Variants in Metabolic Disease. *Cell Stem Cell* **20**, 547-557 e7 (2017).
- 28. Fischer, C. *et al.* A miR-327-FGF10-FGFR2-mediated autocrine signaling mechanism controls white fat browning. *Nat Commun* **8**, 2079 (2017).
- 29. Mardinoglu, A. *et al.* Extensive weight loss reveals distinct gene expression changes in human subcutaneous and visceral adipose tissue. *Sci Rep* **5**, 14841 (2015).
- 30. Hormozdiari, F. *et al.* Colocalization of GWAS and eQTL signals detects target genes. *The American Journal of Human Genetics* **99**, 1245-1260 (2016).

Reviewers' Comments:

Reviewer #1:

Remarks to the Author:

Fathzadeh et al. have provided a revised manuscript of their paper exploring the role of FAM13A in fat distribution.

General comments:

Generally speaking the authors have made substantial efforts to respond to the requests and suggestions of the reviewers. The majority of the additional/refined results support their conclusion that FAM13A is involved in differential fat deposition in both humans and rodents. Additionally there is a recent publication by Tang et al. that provides similar data and both sets of results together make a convincing story that FAM13A is indeed involved in adipose tissue function.

Specific comments:

Human cohort studies:

The authors have provided additional analyses using data from the study of Chu et al. and provide data that suggests indeed a differential association of FAM13A with SAT and VAT. The Chu et al. study is a big study and well powered. I'm not sure to completely understand the plots though. Are all the GWAS SNPs are used in this analysis or only FAM13A SNPs. If the former it would be helpful to highlight the SNPs within the FAM13A locus. If the latter the authors should test a random set of markers from other gene loci to make sure that this is not a random observation. The associations with the UK biobank data is convincing as insofar as markers in FAM13A are associated with body fat traits. However, they do not provide further evidence for the fat distribution, which is not possible to assess with Bioimpedance data.

Cellular studies:

The authors now provide the requested data on adipogenic markers and the results are in the expected direction. I'm fully satisfied with the additional data shown. Concerning the overexpression of FAM13A. As the authors state the recent paper of Tang et al. provides overexpression data in 3T3 preadipocytes and their data agree with the findings of this paper.

As suggested the authors have investigated other molecular endpoints in the SGBS ko model, namely the gene expression of adiponectin and leptin as well as basal and isoproterenol stimulated lipolysis. They don't see changes in these parameters. I have two remarks on these experiments: 1- the authors state that they didn't measure adiponectin and leptin secretion as they didn't observe differences in gene expression differences. I don't agree with their argument. The molecular mechanisms involved in the regulation of gene expression of these adipokines and their secretion are quite different. There is a good example of this in the recent paper by Carayol et al. Nat Commun. 2017 Dec 12;8(1):2084. 2- The data provided is from fully differentiated SGBS adipocytes. Ideally the authors should perform a timecourse. If FAM13A ko induces/accelerates adipogenesis one might expect earlier increase of the adipokines and/or adipogenic function like lipolysis during adipocyte maturation.

eQTL study- the authors have answered my question.

Mouse experiments

The authors have responded to my questions and requests. As far as I can see they have also responded to the questions raised by the other reviewers, especially the quantification of adipocyte sizes in SAT and VAT. As far as I'm concerned they have responded to all the concerns I had.

Reviewer #2:

Remarks to the Author:

I am impressed by the careful response to reviewers, which tries to address multiple comments by the four different reviewers. It is clear that the authors took the reviews seriously. Although the authors have made a multiple additions to text to address comments of this (and other) reviewers, there is not so much additional data. In particular, I still remain concerned about some of the data interpretation, since the numbers of mice in various groups is very small, and many of the physiological changes are also small. This may be the true biology, but it is also possible that it is not.

Secondly, it appears that the Fam13A gene is expressed primarily in mesenchymal stem cells (MSCs, presumably preadipocytes) in the fat pad. Does this mean that all of the correlations with Fam13A expression related to differences in MSC number, or is there a difference in expression per MSC in the fat pad. While they probably cannot answer this directly in the human data, they could potentially address this in the rodent data. It would also be nice to show how Fam13a expression changes with fat cell differentiation.

Reviewer #3:

Remarks to the Author:

The authors have made substantial efforts to improve the manuscript as recommended by reviewers.

The mouse phenotype is modest, even with increasing the n somewhat for some studies. The trends in adipogenesis seem to point to impaired adipogenesis but are not convincing. The recent publication by Tang et al (IJO) had similar negative findings. Thus although the genetic data are clearly interesting, the KO mouse data here does not advance the field very much. The presentation of the genetic data is improved, and SBGS (human adipocyte model) was more convincing but also a modest effect.

A few specific points:

Introduction:

Line 10. Lean with poorer cardiometabolic profile... 'resembling a normal weight but metabolically obese phenotype' - this is a confusing phrase.. as it puts emphasis on 'obese phenotype' rather than metabolically 'unhealthy' obese phenotype..

Results p6, Fig 3F..states 'significant increase in the number of small adipocytes in SAT'. However, the average size looks about the same (not stated.. please state the calculated average diameter / volume) Perhaps fewer WT cells were sized? So expression of the cell sizes as a histogram with % cells on will addresses the question being asked about the distribution into small and large cells (is there a cutoff as there is not a clear bimodal distribution to define 'small'. This percentage could be calculated for each mouse and compared.

The SBGS data are perhaps more convincing -- and the mouse does not seem like a good model. Thus, it is arguably overemphasized in the results and discussion

Much of the mouse data on adipogenesis remain unclear 'trends' . Large SEMs are hard to interpret - so given the small n of 3 in many, presentation of individual point would at least allow the reader to assess if the large SEM is due to an outlier and how much overlap there is between the groups. The paper would read much better with the mouse data in the Supplement.

Similarly, there is not convincing data for a shift in fat distribution in the mice SAT of KO might have slightly smaller fat pads in SAT with perhaps a slightly bigger difference in VAT. Neither are significant and nor are the adipocyte size and numbers with n=7. Perhaps is more animals were studied it would be 'statistically' significant, but this and other phenotypes, even on the HF diet, are of modest size. And HFD fed male mice do not have less SAT.

Discussion:

The discussion presents too much detailed speculation about sex and species differences. It is sufficient to acknowledge the issue more succinctly. Overall the reader does not walk away with a clear picture of the importance or mechanism of FAM13A effects in adipose tissue.

Reviewer #5:

Remarks to the Author:

Please find below my judgement/comments on the author's response to previous reviewer #4 comments.

[Rev 4 - Main comment #1]

Reviewer #4 was concerned about the choice of FAM13A for the functional follow-up analyses. Indeed, most GWAS signals harbor multiple genes and choosing the putative causal gene is a challenging task. To address this, the authors conducted state-of-the-art colocalization analyses to investigate the overlap of phenotype and expression association signals. They considered multiple phenotypes and genes and showed that association with relevant phenotypes is specifically colocalized by expression of FAM13A in SAT, but not by expression of any other gene. I have no concerns about the choice of FAM13A. The analyses provide compelling evidence that FAM13A is the most plausible causal gene at this locus.

Additional minor comment:

On p.4 line 28, the author mention an association of the variant with expression of $P = 10^{-5}$ in VAT. However in the regional association plot (Figure 1), there is 10^{-5} no association in the VAT panel. Could the authors please double-check and explain the inconsistency?

[Rev 4 - Main comment #2]

Reviewer #4 was unclear about the choice of SNPs at the FAM13A locus. The author's response clarifies the confusion. They chose the variant with the highest posterior probability of driving the signal (rs9991328) and for the UKB lookup chose a perfect (genotyped) proxy (rs1377290). This is now clearly described. The additional enrichment analysis are robust and a compelling addition to the manuscript. I support the author's response.

Additional minor comments:

+ Why did they chose a proxy for the UKB lookup in the geneatlas? The initial variant rs9991328 is available in the GeneAtlas already (and perfectly imputed, imp quality >0.99).

+ Interestingly, the variant shows an association with hip in UKB ($P=1e-16$), but nothing with waist ($P=0.82$). This would explains the opposite effect between WHR and BMI and may be worth mentioning.

[Rev 4 - Main comment #3]

This is a perfect response. The colocalization analyses are state-of-the-art.

[Rev 4 - Main comment #4]

I find it sufficient to add the sexually dimorphic aspect to the discussion. While I agree with the content, I am missing the references. There is not a single citation in the chapter on p12, line 11-24. Please include relevant citations.

[Rev 4 - Minor comments]

All reviewer concerns have been addressed.

Point-by-point response to the referees' comments

Response to the editor's comments:

Your manuscript entitled "FAM13A affects body fat distribution and adipocyte function" has now been seen again by 4 referees (note that we had to replace the original reviewer #4 with a new reviewer #5). You will see from their comments below that while they find your work improved, some important points are still raised. We are interested in the possibility of publishing your study in Nature Communications, but would like to consider your response to these concerns in the form of a revised manuscript before we make a final decision on publication.

- We thank the Editor for the continuous interest in our study and another opportunity to revise our manuscript.

We therefore invite you to revise and resubmit your manuscript, taking into account the points raised. Please highlight all changes in the manuscript text file. We note that both reviewers #2 and 3 remain concerned about the mouse data and we would ask that you de-emphasize them and interpret them with caution (even though we do not ask you to move them to the supplement or to add further samples at this point).

- We appreciate the Reviewers' comments and have addressed all points raised (see below the point-by-point responses to each of the reviewers' comments). We have particularly responded to the critiques raised by Reviewers #2 and #3, by de-emphasizing and cautiously interpreting the mouse data through text modifications and/or figure modification. We think these modifications have improved the clarity of the manuscript.
- We have re-organized the order of the figures by moving the mouse data down to the later part of "Results". We have shortened the main figures on the mouse data by moving certain data to the supplement.
- We also performed new experiments on mice to address additional questions brought up by the reviewers.
- Overall, we believe we have conducted a logical and consistent study about the potential effects of *FAM13A* on fat cells which is supported by considerable evidence from human genetics and *in vitro/in vivo* experiments.

We are committed to providing a fair and constructive peer-review process. Do not hesitate to contact us if you wish to discuss the revision in more detail or if there are specific requests from the reviewers that you believe are technically impossible or unlikely to yield a meaningful outcome.

- We have attempted to address all requests from the reviewers by additional experiments, modified data presentation and edited text.

Response to the Reviewers' comments:

Reviewer #1 (Remarks to the Author):

Fathzadeh et al. have provided a revised manuscript of their paper exploring the role of FAM13A in fat distribution.

General comments:

Generally speaking the authors have made substantial efforts to respond to the requests and suggestions of the reviewers. The majority of the additional/refined results support their conclusion that FAM13A is involved in differential fat deposition in both humans and rodents. Additionally there is a recent publication by Tang et al. that provides similar data and both sets of results together make a convincing story that FAM13A is indeed involved in adipose tissue function.

- We thank the reviewer for recognizing our efforts during the revision process.

Specific comments:

Human cohort studies:

The authors have provided additional analyses using data from the study of Chu et al. and provide data that suggests indeed a differential association of FAM13A with SAT and VAT. The Chu et al. study is a big study and well powered. I'm not sure to completely understand the plots though. Are all the GWAS SNPs are used in this analysis or only FAM13A SNPs. If the former it would be helpful to highlight the SNPs within the FAM13A locus. If the latter the authors should test a random set of markers from other gene loci to make sure that this is not a random observation.

The associations with the UK biobank data is convincing as insofar as markers in FAM13A are associated with body fat traits. However, they do not provide further evidence for the fat distribution, which is not possible to assess with Bioimpedance data.

- We thank the reviewer for the opportunity to clarify these issues. We performed this analysis in response to a previous Reviewer comment to explore evidence of association between *FAM13A* expression and body fat distribution in a human cohort. Due to lack of availability of appropriate gene expression data, we analyzed GWAS data from Chu *et al*¹. In response to the Reviewer's question about which SNPs are included, we note that we included SNPs within an 800kb window around rs9991328 at the *FAM13A* locus that were tested for association by Chu *et al*¹. We observed that there was a greater inflation of VAT/SAT volume ratio GWAS p-values (one-sample KS test with the uniform distribution as comparison $p = 2.8e-05$) as compared with p-values from the VAT volume ($p = 0.11$) and SAT volume ($p = 0.03$) GWASs, suggesting that there was weak sub-threshold evidence for association between *FAM13A* SNPs are body fat distribution. The inflation results are not directly comparable across loci due to the different LD patterns

at each locus, but the association patterns can be compared across traits within a locus.

- We expect there to be other loci in the genome that demonstrate this pattern as there are loci shown to have a similar profile of associations across IR-related traits using clustering². For instance, SNPs in the *LYPLAL1* locus, which was shown to cluster most closely with *FAM13A* association signals across IR-traits in independent data², show a strong association with VAT/SAT ratio (min GWAS p = 2.3e-08) and less robust associations with VAT volume (min GWAS p = 0.00012) and SAT volume (min GWAS p = 0.00024). We also selected a small random set of markers across the genome to check for similar patterns, and the results are shown below. While this is not an exhaustive search, this pattern of associations taken in context with the rest of the evidence in the manuscript highlights *FAM13A* activity in adipose biology.

Gene/locus	Chromosome	min SAT volume GWAS p-value	min VAT volume GWAS p-value	min VAT/SAT volume GWAS p-value
FABP3 *	1	0.00055	0.00052	0.00082
CDC25A	3	0.0049	0.00027	0.0025
LINC00581	6	0.00015	8.5e-05	0.00052
HGF	7	0.00069	0.00036	0.00016
GNG5P5	11	0.0015	0.00088	0.0011
OR10G3	14	4.6e-06	0.00026	0.00053
CES2	16	0.00037	0.00044	0.00073

*Genes that have shown some relevance to adipose biology

Cellular studies:

The authors now provide the requested data on adipogenic markers and the results are in the expected direction. I'm fully satisfied with the additional data shown. Concerning the overexpression of *FAM13A*, as the authors state the recent paper of Tang et al. provides overexpression data in 3T3 preadipocytes and their data agree with the findings of this paper.

- We appreciate the Reviewer's comments.

As suggested the authors have investigated other molecular endpoints in the SGBS ko model, namely the gene expression of adiponectin and leptin as well as basal and isoproterenol stimulated lipolysis. They don't see changes in these parameters. I have two remarks on these experiments: 1- the authors state that they didn't measure adiponectin and leptin secretion as they didn't observe differences in gene expression differences. I don't agree with their argument. The molecular mechanisms involved in the regulation of gene expression of these adipokines and their secretion are quite different. There is a good example of this in the recent paper by Carayol et al. Nat Commun. 2017 Dec 12;8(1):2084. 2- The data provided is from fully differentiated SGBS adipocytes. Ideally the authors should perform a time-course. If *FAM13A* ko induces/accelerates adipogenesis one might expect earlier increase of the adipokines and/or adipogenic function like lipolysis during adipocyte maturation.

- We thank the reviewer for the insightful comments and suggestions on our metabolic and functional studies in SGBS models. We completely agree with the reviewer that gene expression of adipokines does not necessarily reflect their secretory activities, and our negative findings on adipokine production and lipolysis might be due to the specific

experimental approach or timing used. During our previous SGBS cell perturbation and differentiation experiments, unfortunately, we did not save enough replicates of conditional media for ELISA measurement of adipokines or perform lipolysis on cells during early differentiation state (i.e. Day 5) or in the maturation stage (i.e. Day 12). As the Reviewer suggested, in the future we will perform a comprehensive time-course study by introducing the gene perturbation and measuring metabolic activities of the cells at various stages of adipocyte development. In the revised manuscript, we have modified the description and discussion of the metabolic data on SGBS with caution to reflect the Reviewer's comments.

p.6, line 13-18: To investigate the impact of FAM13A on mature adipocyte function, we introduced siRNAs in the late stage of adipocyte differentiation (Day 8) to separate the confounding effect of FAM13A on adipogenesis. Under this experimental timing and condition, although FAM13A knockdown (Fig. 3F) did not significantly affect gene expression of adipokines (e.g. ADIPOQ and LEP) (Fig. S3B) and lipolysis (Fig. S3C), there was an increased trend of glucose uptake into FAM13A-deficient SGBS adipocytes under both basal and insulin-stimulated conditions (Fig. 3G).

p.11, line 11-13: In-depth metabolic and thermogenic functional studies at various stages of adipocyte development are warranted to fully reveal the involvement of FAM13A on white/beige adipocyte fate determination and metabolism.

eQTL study- the authors have answered my question.

- We thank the Reviewer for those questions.

Mouse experiments:

The authors have responded to my questions and requests. As far as I can see they have also responded to the questions raised by the other reviewers, especially the quantification of adipocyte sizes in SAT and VAT. As far as I'm concerned, they have responded to all the concerns I had.

- We appreciate the comments and suggestions raised up by all Reviewers. We have modified the interpretation of our mouse data further in the revised manuscript.

Reviewer #2 (Remarks to the Author):

I am impressed by the careful response to reviewers, which tries to address multiple comments by the four different reviewers. It is clear that the authors took the reviews seriously. Although the authors have made multiple additions to text to address comments of this (and other) reviewers, there is not so much additional data. In particular, I still remain concerned about some of the data interpretation, since the numbers of mice in various groups is very small, and many of the physiological changes are also small. This may be the true biology, but it is also possible that it is not.

- We thank the Reviewer for recognizing our efforts in improving our paper by taking into account of advices from all Reviewers. We agree that many of the physiological changes in our KO mouse models are small. The similar insignificant findings were replicated by a different group who bred the animals and perform experiments totally independently from us³. Considering the Reviewer's critique, we modified the interpretation of our mouse data more cautiously by using descriptive rather than conclusive words, and pointed out the subtle phenotypic changes to the readers. We believe the results of this study will only be a beginning of a series of future studies including how *FAM13A* acts on fat cells and under what conditions these effects are most profound.

p. 12, line 12-15: While our findings and other studies give support to the notion that adipose tissue mediates FAM13A action, the metabolic phenotypes of whole-body Fam13a KO mice are subtle. Adipose-specific knockouts of Fam13a would be useful to study the role of Fam13a specifically in adipose tissue and its causal impact on systemic metabolism.

Secondly, it appears that the *Fam13A* gene is expressed primarily in mesenchymal stem cells (MSCs, presumably preadipocytes) in the fat pad. Does this mean that all of the correlations with *Fam13A* expression related to differences in MSC number, or is there a difference in expression per MSC in the fat pad? While they probably cannot answer this directly in the human data, they could potentially address this in the rodent data. It would also be nice to show how *Fam13a* expression changes with fat cell differentiation.

- We thank the Reviewer for this important open question. We have actively pursued the answers by performing additional mouse experiments during this round of revision. Responses to these points are as follows:

a) **How does *Fam13a* expression change during fat cell differentiation?**

- During adipogenic differentiation of mouse stromal vascular fraction of cells (SVFs) isolated from subcutaneous fat of wild-type mice, *Fam13a* gene expression is induced during adipogenesis (Fig.5A).

p.7, line 27-29: After growing SVFs into confluence (i.e. Day 0) and inducing them to adipogenic differentiation, Fam13a expression was found to be elevated during in vitro adipogenesis (Fig. 5A).

b) Are the observed correlations with adipose tissue level of FAM13A expression related to differences in adipose-derived MSC number?

- **Differences in MSC number between WT and *Fam13a* KO mice:** We isolated SVFs - which contain MSCs and many other cell types - from subcutaneous fat of WT and KO mice, stained SVFs with multiple fluorochrome-conjugated antibodies and analyzed the number of cell subpopulations according to the previously published protocol⁴ and based on the FACS selection strategy illustrated in **Fig. S9C**. Our results showed that SVFs of KO mice tend to contain a larger percentage of CD45- cells including adipose progenitor cells (APC, CD45-SCA1+CD31-) as well as endothelial cells (EC, CD45-SCA1+CD31+) while maintaining a lower population of chronic inflammation-related immune cells (CD45+) (**Fig. S9D-E**). After plating into the same cell density following FACS sorting, APC of KO mice tends to differentiate better into adipocytes, as compared to APC from WT (**Fig. S9F-G**). These results suggest that *Fam13a* expression in subcutaneous fat tissue of mouse model is negatively related to the number of adipose

Fig. S9C: FACS gating strategy for isolating adipose progenitor cell (APC) from SVFs of

Fig. S9D and S9E: Percentage of cell subsets in WT vs. KO fat tissue

Fig. S9F and S9G: Adipogenic differentiation of APCs isolated from WT vs. KO fat tissue

precursors as well as the adipogenic ability of these cells, which is consistent with our previous finding that the crude SVFs of *Fam13a*-deficient mice exhibit higher adipogenic potential. Of note, other cell types, such as endothelial cells residing in adipose tissue, may also contribute to improved adipogenesis *in vivo*, as endothelial cells are known to secrete lipid signals such as PPAR γ ligands that induce preadipocyte differentiation and lipid storage in response to metabolic state changes⁵. It would be of interest to follow up on the impact of *Fam13a* on the cell-to-cell communication within the adipose microenvironment rather than focusing only on the biology of a pure population of (pre)adipocytes in the future. We, therefore, thank the Reviewer for this stimulating question as it may lead us to discover more physiological changes due to disruption of *Fam13a* in adipose tissue.

- c) **Are the observed correlations with adipose tissue level of FAM13A expression related to differences in its expression per MSC in the fat pad?**
- We do not have sufficient evidence to answer it at this point. To fully address this question, in the future we would need to isolate MSCs from wild-type mice before and after HFD, and then evaluate *Fam13a* expression changes per MSC.
 - The main scope of our study is to recognize the causal effect of *FAM13A* in fat tissue development in the context of obesity-related diseases. To this end, we have applied a classical approach in which *FAM13A* expression was first knocked down either *in vitro* or *in vivo*, followed by the observations of altered outcomes in reference to the controlled samples. Although we could not fully answer the Reviewer's question regarding the correlation at this point, we performed and presented additional experiments upon request to gain more insights into the role *FAM13A* plays in fat tissue.

Reviewer #3 (Remarks to the Author):

The authors have made substantial efforts to improve the manuscript as recommended by reviewers.

- We thank the Reviewer for recognizing our efforts.

The mouse phenotype is modest, even with increasing the n somewhat for some studies. The trends in adipogenesis seem to point to impaired adipogenesis but are not convincing. The recent publication by Tang et al (IJO) had similar negative findings. Thus although the genetic data are clearly interesting, the KO mouse data here does not advance the field very much. The presentation of the genetic data is improved, and SBGS (human adipocyte model) was more convincing but also a modest effect.

- We agree with the Reviewer that a majority of the phenotypes evaluated on *Fam13a* KO mice were not significantly different than the WT mice, and that the effects of *FAM13A* knockdown on adipogenesis *in vitro* is modest. We would like to address this critique from three points:
 - a) In our *in vivo* studies, we have designed the animal experiments carefully, analyzed the results cautiously, and repeated the results in separate cohorts of mice. The modest changes in the whole-body metabolism of *Fam13a* KO mice were found by another group³ performing the animal breeding and experiments totally independently from us. Thus, we believe our mouse phenotype, while “modest”, is valid.
 - b) In our *in vitro* studies, our data in the knockdown study of a human preadipocyte cell suggested that the presence of *FAM13A* may inhibit adipogenesis, while Tang *et al.*³ reached the same conclusion by overexpressing *Fam13a* in a mouse preadipocyte cell. Results from two different cellular models and two different gene manipulation strategies, conducted by two independent groups, have provided considerable evidence that *FAM13A* plays a role in adipogenesis.
 - c) Concerning the value of our KO mouse data in advancing the field, we would like to point out that this mouse model is a total-body knockout of *Fam13a*, which may have a limitation in addressing the adipose-specific role of *Fam13a*. Our human genetic data suggested that insulin resistance-associated *FAM13A* variants correlate with its eQTL level specifically in adipose tissue. However, there is evidence that *FAM13A* also plays a role in other tissues (such as lung), which may drive total-body phenotypic changes in scattered directions. Thus, as the Reviewer implied, an adipose-specific knockout model of *Fam13a* would be a better model to disconnect the effects masked by other tissues and to highlight *Fam13a* regulation of whole-body metabolism via its action on fat. However, we think our data on total KO mice advances the field by driving more attention to the adipose-specific role of *FAM13A* for the future studies. We have discussed the limitations and implications of our mouse data in the text.

p.12, line 12-15: While our findings and other studies give support to the notion that adipose tissue mediates FAM13A action, the metabolic phenotypes of whole-body Fam13a KO mice are subtle. Adipose-specific knockouts of Fam13a would be useful to study the role of Fam13a specifically in adipose tissue and its causal impact on systemic metabolism.

A few specific points:

1. Introduction: Line 10. Lean with poorer cardiometabolic profile.... ‘resembling a normal weight but metabolically obese phenotype’ - this is a confusing phrase... as it puts emphasis on ‘obese phenotype’ rather than metabolically ‘unhealthy’ obese phenotype...

- We thank the reviewer for the comment and have changed “metabolically obese phenotype” into “metabolically unhealthy phenotype”.

p. 3, line 7-10: Recently, we and others^{1,3,4} identified a cluster of common risk variants from genome-wide association studies (GWAS) that are associated with normal or lower adiposity and, at the same time, with a poorer cardiometabolic profile (e.g. increased fasting insulin (FI), increased triglycerides, and decreased HDL-cholesterol levels), resembling a normal weight but metabolically unhealthy phenotype.

2. Results p6, Fig 3F...states ‘significant increase in the number of small adipocytes in SAT’. However, the average size looks about the same (not stated... please state the calculated average diameter / volume) Perhaps fewer WT cells were sized? So expression of the cell sizes as a histogram with % cells on will addresses the question being asked about the distribution into small and large cells (is there a cutoff as there is not a clear bimodal distribution to define ‘small’). This percentage could be calculated for each mouse and compared.

- We thank the Reviewer for these great suggestions. We made the following changes in the revised manuscript, accordingly.
- a) We edited the strong statement about the adipocyte size distribution, presented and stated the average diameter of adipocytes (**Fig. 4C**). We agree with the Reviewer that the average size of adipocytes was not different

Fig. 4C: Average size of adipocytes in WT vs. KO mice

- b) As recommended by the reviewer, to avoid the bias caused by the unequal number of adipocytes between WT and KO cells being counted, we presented the distribution of adipocyte size in a histogram with the % of cells, rather than the absolute number of cells, as the Y-axis (**Fig. 4D-4E**). In SAT, there is a shift towards smaller adipocytes in KO mice (**Fig. 4D**), while in VAT, KO mice has a shift towards larger adipocytes (**Fig. 4E**).

Fig. 4D and 4E: Comparison of adipocyte size distribution in WT vs. KO mice

Fig. S4G: Percentile analysis of adipocyte size

	SAT		VAT	
	WT (µm)	KO (µm)	WT (µm)	KO (µm)
Minimum	20.01	20	20.05	20.03
25% Percentile	27.99	27.53	33.6	37.845
Median	36.585	34.47	46.29	54.225
75% Percentile	47.4175	43.4775	63.3575	72.4325
Maximum	151.42	107.3	286.54	170.69

Fig. S4H-I: Percentage of cells below or above the cutoff

c) As mentioned by the reviewer, since there is not a clear bimodal distribution pattern to define “small” or “big” adipocytes, we used the median size of WT cells (36.6 µm in SAT, 36.4 µm in VAT, **Fig. S4G**) as the cutoff and calculated the percentage of cells that are below or above this median size for each mouse. KO mice seem to have a slight increase in the percentage of cells that fall into the “smaller” adipocytes in SAT (**Fig. S4H**) and “larger” adipocytes in VAT (**Fig. S4I**).

▪ Together, we modified the text related to adipocyte size distribution as follows:

*p. 7, line 4-9: There was no significant difference in total number of adipocytes per depot (**Fig. 4B**) or in the average size of adipocytes between WT (SAT diameter = $40.4 \pm 17.6 \mu\text{m}$, VAT diameter = $51.7 \pm 24.7 \mu\text{m}$) and KO (SAT diameter = $36.1 \pm 12.1 \mu\text{m}$, VAT diameter = $56.5 \pm 22.8 \mu\text{m}$) (**Fig. 4C**). However, when comparing the size distribution curve of adipocytes in SAT and VAT, there was a slight increase in the number of small adipocytes in SAT (**Fig. 3A, 3D, S4G and S4H**) with a corresponding modest increase in the number of relatively large adipocytes in VAT (**Fig. 3A, 3E, S4G and S4I**) of *Fam13a* KO mice.*

3. The SBGS data are perhaps more convincing -- and the mouse does not seem like a good model. Thus, it is arguably overemphasized in the results and discussion.

- To address the Reviewer’s concerns, we de-emphasized the mouse data in both Results and Discussion by using more cautious wording, moving some insignificant data to the supplement and discussing the potential limitation of the mouse model. We moved SBGS data up in the results to highlight the more significant human genetics and *in vitro* data.

4. Much of the mouse data on adipogenesis remain unclear ‘trends’. Large SEMs are hard to interpret – so given the small n of 3 in many, presentation of individual point would at least allow the reader to assess if the large SEM is due to an outlier and how much overlap these is between the groups. The paper would read much better with the mouse data in the Supplement.

- We thank the Reviewer for the suggestion of including all data points in the figures. We modified all graphs accordingly. Although there was large variation within the groups themselves, the “trends” of induced adipogenesis in KO mice were due to the overall difference between groups rather than a specific outlier in either group (**Fig. 5C and 5F**).
- We shortened the main figures for the mouse data by combining the data on chow and HFD into one figure and moving certain data to the supplement. At the suggestion of the Editor we did not move all mouse data to the supplement.

Fig. 5C: Adipogenic markers in SVF-differentiated adipocytes

Fig. 5F: Triglyceride accumulation in SVF-differentiated adipocytes

5. Similarly, there is not convincing data for a shift in fat distribution in the mice SAT of KO might have slightly smaller fat pads in SAT with perhaps a slightly bigger difference in VAT. Neither are significant and nor are the adipocyte size and numbers with n=7. Perhaps is more animals were studied it would be ‘statistically’ significant, but this and other phenotypes, even on the HF diet, are of modest size. And HFD fed male mice do not have less SAT.

- Regarding the fat distribution data, we agree with the Reviewer that under normal chow, the absolute weight of individual fat pads was not different between WT and KO. After a 14-week HFD challenge, we observed a reduced weight of VAT – a more disease-driven fat depot - and a reduced ratio of VAT/SAT in KO mice. Indeed, we did not necessarily expect to see “less SAT” in HFD-fed KO mice, as SAT might serve as a safer deposit site for excess lipids and the shift of lipid deposition from VAT to SAT might be

protective under HFD condition. There might be two reasons that we did not observe a statistically significant increase in SAT of HFD-fed KO mice. First, we only isolated and weighed inguinal adipose tissue as a representative depot for SAT. However, since SAT sits under the skin throughout the body, the weight shift to SAT might be underrepresented by the measurement of a single depot. Secondly, since we only noticed the fat distribution changes after HFD, we believe the fat distribution parameter before HFD in each animal is an important baseline value to quantify the fat shift during the HFD challenge more accurately. To solve these two problems in the future, we are planning to use CT or MRI for the quantification of SAT and VAT in the whole-body level before and after HFD challenge for each mouse but this is beyond the scope of the current manuscript.

Discussion:

The discussion presents too much detailed speculation about sex and species differences. It is sufficient to acknowledge the issue more succinctly. Overall the reader does not walk away with a clear picture of the importance or mechanism of FAM13A effects in adipose tissue.

- We significantly shortened the discussion about the differences of fat distribution between sexes and species, as below.

p. 12, line 17-25: Sex dimorphism in the association of FAM13A with body fat distribution is observed in both humans and mice. In human GWAS, the WHRadjBMI locus harboring FAM13A variants showed a stronger effect in women than in men. Sex dimorphism in the genetic regulation of fat distribution may be related to the difference in body fat shape between the sexes^{39,40}. In mouse models, while in male mice there was a difference in VAT/SAT between WT and Fam13a KO mice upon HFD, no significant phenotypic effects of Fam13a deficiency were observed in female mice. Although current translational investigation on fat distribution and its metabolic consequences relies on rodent models, they may not faithfully mimic all aspects of human fat biology due to differences in anatomical location and function of fat depots between species^{39,40}.

- To present the reader with a clearer picture of our most important findings, we re-organized the order of the figures and moved certain mouse data to the Supplement. In addition, we constructed a graphical summary to highlight our main findings (as shown on the right) and we are happy to include it in the manuscript if the Reviewer thinks it will be useful.

Reviewer #5 (Remarks to the Author):

Please find below my judgement/comments on the author's response to previous reviewer #4 comments.

[Rev 4 - Main comment #1]

Reviewer #4 was concerned about the choice of FAM13A for the functional follow-up analyses. Indeed, most GWAS signals harbor multiple genes and choosing the putative causal gene is a challenging task. To address this, the authors conducted state-of-the-art colocalization analyses to investigate the overlap of phenotype and expression association signals. They considered multiple phenotypes and genes and showed that association with relevant phenotypes is specifically colocalized by expression of FAM13A in SAT, but not by expression of any other gene. I have no concerns about the choice of FAM13A. The analyses provide compelling evidence that FAM13A is the most plausible causal gene at this locus.

- We appreciate the Reviewer's comments. We also thank the previous Reviewer for the in-depth comments and suggestions which have improved the manuscript.

Additional minor comment:

On p.4 line 28, the author mention an association of the variant with expression of $P = 10^{-5}$ in VAT. However in the regional association plot (Figure 1), there is 10^{-5} no association in the VAT panel. Could the authors please double-check and explain the inconsistency?

- We thank the Reviewer for noticing this inconsistency. We intended to convey that the association has p-value $> 1e-05$. We have edited the text to reflect this.

[Rev 4 - Main comment #2]

Reviewer #4 was unclear about the choice of SNPs at the FAM13A locus. The author's response clarifies the confusion. They chose the variant with the highest posterior probability of driving the signal (rs9991328) and for the UKB lookup chose a perfect (genotyped) proxy (rs1377290). This is now clearly described. The additional enrichment analysis are robust and a compelling addition to the manuscript. I support the author's response.

- We appreciate the Reviewer's support.

Additional minor comments:

+ Why did they chose a proxy for the UKB lookup in the geneatlas? The initial variant rs9991328 is available in the GeneAtlas already (and perfectly imputed, imp quality >0.99).

- We performed our own PheWAS analysis and did not look up results in GeneAtlas. We agree with the Reviewer that the imputation quality for rs9991328 appears to be very good. In principle, we decided to use rs1377290 because it is very strongly correlated with rs9991328 (LD $R^2 = 1.0$) and is directly imputed in the UK Biobank. We expect the results to be very similar for a PheWAS performed with rs9991328.

+ Interestingly, the variant shows an association with hip in UKB (P=1e-16), but nothing with waist (P=0.82). This would explain the opposite effect between WHR and BMI and may be worth mentioning.

- We agree with the Reviewer that this pattern in associations is interesting and supports our conclusions.

[Rev 4 - Main comment #3]

This is a perfect response. The colocalization analyses are state-of-the-art.

- We thank the Reviewer for the compliment.

[Rev 4 - Main comment #4]

I find it sufficient to add the sexually dimorphic aspect to the discussion. While I agree with the content, I am missing the references. There is not a single citation in the chapter on p12, line 11-24. Please include relevant citations.

- We apologize for missing the references. We have included relevant citations in the discussion of sexual dimorphism (quoted above in response to the last question of Reviewer#3).

[Rev 4 – Minor comments]

All reviewer concerns have been addressed.

- We thank the Reviewer for recognizing our efforts.

References included in the response to reviewers:

1. Chu, A.Y. *et al.* Multiethnic genome-wide meta-analysis of ectopic fat depots identifies loci associated with adipocyte development and differentiation. *Nat Genet* **49**, 125-130 (2017).
2. Yaghothkar, H. *et al.* Genetic evidence for a normal-weight "metabolically obese" phenotype linking insulin resistance, hypertension, coronary artery disease, and type 2 diabetes. *Diabetes* **63**, 4369-77 (2014).
3. Tang, J. *et al.* Obesity-associated family with sequence similarity 13, member A (FAM13A) is dispensable for adipose development and insulin sensitivity. *Int J Obes (Lond)* **43**, 1269-1280 (2019).
4. Krueger, KC. *et al.* Characterization of Cre recombinase activity for in vivo targeting of adipocyte precursor cells. *Stem Cell Reports* 3(6), 1147-58 (2014).
5. Gogg, S. *et al.* Human adipose tissue microvascular endothelial cells secrete PPAR γ ligands and regulate adipose tissue lipid uptake. *JCI insight* **4**(5): e125914 (2019).
6. McLaughlin, T. *et al.* Enhanced proportion of small adipose cells in insulin-resistant vs insulin-sensitive obese individuals implicates impaired adipogenesis. *Diabetologia* **50**, 1707-15 (2007).

Response to the editorial requests:

To improve the quality of methods and statistics reporting in our papers, we are now asking all authors to complete an editorial policy checklist that verifies compliance with all required editorial policies. Please ensure that the checklist is completed and uploaded with your revised article. All points on the policy checklist must be addressed; if needed, please revise your manuscript in response to these points. Please note that this form is a dynamic 'smart pdf' and must therefore be downloaded and completed in Adobe Reader, instead of opening it in a web browser. Editorial policy checklist: <https://www.nature.com/documents/nr-editorial-policy-checklist.pdf>

- We have completed the editorial policy checklist and uploaded it with our revised article.

At the same time, we ask that you ensure your manuscript complies with our editorial policies. Please ensure that the following requirements are met, and any relevant checklist is completed or updated and uploaded as a Related Manuscript file type with the revised article. Reporting requirements for life sciences research: <https://www.nature.com/documents/nr-reporting-summary.pdf>

- We have completed relevant requirements in the reporting checklist and uploaded it.

Furthermore, your manuscript should comply with our format requirements, which are summarized on the following checklist: <https://www.nature.com/documents/ncomms-manuscript-checklist.pdf>

- The format requirements are compiled in our manuscript.

In an effort to ensure reproducibility of research data, we now also require that you provide a separate source data file. The source data file should, as a minimum, contain the raw data underlying all reported averages in graphs and charts, and uncropped versions of any gels or blots presented in the figures. To learn more about our motivation behind this policy, please see <https://www.nature.com/articles/s41467-018-06012-8>.

Within the source data file, each figure or table (in the main manuscript and in the Supplementary Information) containing relevant data should be represented by a single sheet in an Excel document, or a single .txt file or other file type in a zipped folder. Blot and gel images should be pasted in and labelled with the relevant panel and identifying information such as the antibody used. We also encourage you to include any other types of raw data that may be appropriate. An example source data file is available demonstrating the correct format:

<https://www.nature.com/documents/ncomms-example-source-data.xlsx>

The file should be labelled 'Source Data', with the title and a brief description included in your cover letter, and should be mentioned in all relevant figure legends using the template text below: "Source data are provided as a Source Data file."

Data availability statements and data citations policy: All Nature Communications manuscripts must include a section titled "Data Availability" as a separate section after the Methods section but before the References. For more information on this policy, and a list of examples, please see <https://www.nature.com/documents/nr-data-availability-statements-data-citations.pdf>

- Accession codes for deposited data
- Other unique identifiers (such as DOIs and hyperlinks for any other datasets)
- At a minimum, a statement confirming that all relevant data are available from the authors
- If applicable, a statement regarding data available with restrictions
- If a dataset has a Digital Object Identifier (DOI) as its unique identifier, we strongly encourage including this in the Reference list and citing the dataset in the Data Availability Statement.
- If a source data file is provided, please add a reference to this in the data availability statement. For example:
 - "The source data underlying Figs 1a, 2a–d, 6d, h and 7c and Supplementary Figs 1a and 5d are provided as a Source Data file."

DATA SOURCES: We strongly encourage authors to deposit all new data associated with the paper in a persistent repository where they can be freely and enduringly accessed. We recommend submitting the data to discipline-specific, community-recognized repositories, where possible and a list of recommended repositories is provided here: <http://www.nature.com/sdata/policies/repositories>

If a community resource is unavailable, data can be submitted to generalist repositories such as figshare (<https://figshare.com/>) or Dryad Digital Repository (<http://datadryad.org/>). Please provide a unique identifier for the data (for example a DOI or a permanent URL) in the data availability statement, if possible. If the repository does not provide identifiers, we encourage authors to supply the search terms that will return the data. For data that have been obtained from publicly available sources, please provide a URL and the specific data product name in the data availability statement. Data with a DOI should be included in the reference list and cited where relevant.

Please refer to our data policies

here: <http://www.nature.com/authors/policies/availability.html>

- We have provided a Source Data File.

If you opted into the journal hosting details of a preprint version of your manuscript via a link on our dedicated website (<https://nature-research-under-consideration.nature.com>), it will remain on this site while you are revising your manuscript, as we consider the file to remain active. Should you wish to remove these details, please email naturecommunications@nature.com indicating your manuscript number and the link on our website that was previously sent to you. Please see our pre-publicity policy

at <http://www.nature.com/authors/policies/confidentiality.html> For more information, please refer to our FAQ page at <https://nature-research-under-consideration.nature.com/posts/19641-frequently-asked-questions>

Springer Nature encourages all authors and reviewers to adopt an Open Researcher and Contributor Identifier (ORCID). ORCID is a community-based initiative that provides an open, non-proprietary and transparent registry of unique identifiers to help disambiguate research contributions. All authors who link their ORCID to their account in our submission system will have their ORCID published on their articles, if the article is accepted for publication. Please note that this is only possible if ORCIDs are linked prior to acceptance, that is, it is not possible to add ORCIDs at proof.

Please ensure that all co-authors are aware that they can add their ORCIDs to their accounts and that they must do so prior to acceptance.

To add an ORCID please follow these instructions:

1. From the home page of the MTS click on 'Modify my Springer Nature account' under 'General tasks'.
2. In the 'Personal profile' tab, click on 'ORCID Create/link an Open Researcher Contributor ID (ORCID)'. This will re-direct you to the ORCID website.
- 3a. If you already have an ORCID account, enter your ORCID email and password and click on 'Authorize' to link your ORCID with your account on the MTS.
- 3b. If you don't yet have an ORCID account, you can easily create one by providing the required information and then clicking on 'Authorize'. This will link your newly created ORCID with your account on the MTS.

- All authors were aware of their right of adding ORCIDs prior to acceptance.

Reviewers' Comments:

Reviewer #1:

Remarks to the Author:

The authors have further improved the manuscript, which now is easy to follow and clear. They provide additional animal and cellular data that supports their conclusions. Following the other reviewers' suggestions they have re-phrased the conclusions from the mouse studies more carefully.

Specific comments:

1- As to my further questions the authors have made the appropriate changes in the text. I would still have liked to see the time-course in the cell model but I agree that this may be out of the scope for this article and should be addressed in a follow-up study.

2- concerning the question of one reviewer about the average size of the adipocytes I tend to agree with the authors that the size distribution is more informative than the overall average.

I'm fully satisfied with the authors' revisions and have no further comments.

Reviewer #2:

Remarks to the Author:

Summary of the paper:

In this manuscript, Fathzadeh et al. demonstrate that the GWAS-associated variants within the FAM13A locus, especially rs9991328, are associated with FAM13A expression in SAT, suggesting that FAM13A might have a specific role in SAT. The study links human genetic data with in vitro and mice experiments to understand the biological role of FAM13A in regulating body fat distribution. The study shows that the presence of FAM13A risk alleles perturb the body fat distribution through limiting adipogenesis in SAT, thus FAM13A negatively regulates adipocyte development.

General comments:

In general, this is an interesting and well-done study. However, several more general and specific points need to be considered.

1. The study would be strengthened by more rigorous analyses of the cellular data involving the role of FAM13A in SGBS differentiation and mature adipocyte function. Some specific suggestions are outlined in the specific comments section below.

2. The molecular mechanisms linking FAM13A and adipogenesis need further investigation. For example, the Figure 6B shows a large number of differentially expressed genes between WT and KO samples, including various genes associated with fat cell biology. The authors point out that genes in the NAD⁺ salvage pathway are also overrepresented. Given that the optimal level of intracellular NAD⁺ is important, could alternation in NAD⁺ levels serve as an upstream driver of the differentially expressed genes upon FAM13A deletion? The authors should investigate this point further to better understand the molecular mechanism of how FAM13A regulates intracellular milieu to impact fat cell biology.

Specific comments:

1. The authors state that rs9991328 is considered a lead variant (line 99, page 4); however, a different variant (rs1377290) was used for the PheWAS analyses in figure 2A instead. The reasoning is not clear. Would the associations of rs1377290 allele with increased WHR, decreased trunk/body fat percentage, mean platelet volume, and BMI be also found using the lead variant rs9991328?

2. Figure 3A shows that FAM13A expression is increased on day 6 of SGBS differentiation but

seems to be lower by day 12. Is there any statistically significant difference between the FAM13A mRNA level at D6 as compared to D12? What does this transient increase in FAM13A during adipogenesis mean?

3. Figure 3C does not show any difference in PPARG mRNA expression, although the authors state that FAM13A knockdown results in the upregulation of adipogenesis markers. This statement needs to be further supported with results including mRNA levels of PPARG downstream targets, such as Fabp4 and adiponectin. Considering the protein levels of adipogenic markers would also further support the role of FAM13A in this process. In addition, oil red o staining should be performed in scRNA and siFAM13A cells at D5 coinciding with mRNA and protein levels of adipogenic markers.

4. Figure S3 shows the upregulation of beige adipocyte markers in siFAM13A D5 cells as compared to scRNA. Why is it important to look at the beige adipocyte markers in this context? The authors do not clarify the significance of these findings

5. The CRISPRi should be validated through western blotting for FAM13A protein to support the findings in figure 3D.

6. Figure 3E does not show any statistically significant difference in mRNA expression of CEBPA and PPARG. Additional analyses of mRNA levels of PPARG downstream targets, such as Fabp4 and adiponectin, as well as western blotting of adipogenesis markers should be included.

7. To study the role of FAM13A in mature adipocyte function, the knockdown experiments followed by glucose uptake and lipolysis experiments should be carried out at a stage earlier than D12 when the FAM13A expression is still highly induced (figure 3A, comment#2). The increasing trend of glucose uptake in siFAM13A adipocytes is not convincing. This no change in adipocyte function upon FAM13A knockdown could be attributed to decreased FAM13A levels by D12 of SGBS differentiation.

8. Figure 4F shows an increase in BW in FAM13A KO males on HFD, but figure 4G shows no change in SAT weight and a decrease in VAT. What is contributing to the increase in BW? DEXA scan should be performed to find out if the lean mass is increased.

9. FAM13A knockdown in SGBS cells led to a modest increase in adipogenesis (figure 3). Figure S4E and 4G do not show any difference in SAT weight in FAM13A KO mice, suggesting that the cell-autonomous role of FAM13A in regulating adipocyte differentiation is masked by the use of full-body FAM13A KO mice. The authors should mention this point in the manuscript.

10. Induction of beige adipocyte markers in KO D10 adipocytes (figure S8A and S8B) suggests that FAM13A deletion might promote browning of SAT correlating with an increase in smaller adipocytes. Although figure S8C suggests no change in BAT markers between WT and KO, the mRNA levels of Pgc1a, Ucp1, and Cidea should be measured in the SAT of WT and KO mice to determine if FAM13A deletion is associated with the browning of white adipose tissue.

11. The glucose uptake experiments in D10 adipocytes from WT and KO shown in figure 5G and 5H are not strong findings. Therefore, it is not reasonable to suggest that FAM13A KO adipocytes are more functionally active to deposit glucose, as the authors have stated on lines 213-215 on page 8 of the manuscript.

12. Given that FAM13A is mainly expressed in SAT, why does FAM13A expression goes down 30-fold in HFD VAT, but not as much in SAT and liver (figure 6A). The authors should further comment on this. Also, the expression of FAM13A in chow of VAT seems to be higher than the expression in chow of SAT, which contradicts the author's initial findings regarding FAM13A specific expression in the SAT. Western blotting for FAM13A protein using SAT and VAT samples from WT

mice on a chow diet should be performed to clearly show the protein level and specificity of FAM13A expression in SAT as compared to VAT.

13. The gating strategy in figure S9C shows that live cells are represented as PI positive. This must be a mistake in the gating label. The PI staining is only taken up by the dead cells, so the cells selected with the first gate must be PI negative cells (live cells).

14. The authors refer to figure 6C in the discussion, which is not found in the manuscript.

15. There is a mistake in the figure labeling within the discussion. The statement on line 288, 289, page 11 of the manuscript should refer to figure S8 instead of S6.

Reviewer #3:

Remarks to the Author:

The data are presented in a much clearer and succinct manner, addressing prior concerns.

This gene is consistently associated with fat distribution so the experiments interrogate its effects in adipocyte, now that they are clarified and additional experiments added, so these results are important for the field

Point-by-point response to reviewers

REVIEWERS' COMMENTS:

Reviewer #1 (Remarks to the Author):

The authors have further improved the manuscript, which now is easy to follow and clear. They provide additional animal and cellular data that supports their conclusions. Following the other reviewers' suggestions, they have re-phrased the conclusions from the mouse studies more carefully.

Specific comments:

1- As to my further questions, the authors have made the appropriate changes in the text. I would still have liked to see the time-course in the cell model, but I agree that this may be out of the scope for this article and should be addressed in a follow-up study.

- We thank the Reviewer for this important suggestion. We will perform a comprehensive time-course study by introducing the gene perturbation and measuring metabolic activities of the cells at various stages of adipocyte development in the follow-up study.

2- Concerning the question of one reviewer about the average size of the adipocytes I tend to agree with the authors that the size distribution is more informative than the overall average.

I'm fully satisfied with the authors' revisions and have no further comments.

- We thank the Reviewer for all the comments and suggestions throughout this review process.

Reviewer #2 (Remarks to the Author):

Summary of the paper:

In this manuscript, Fathzadeh et al. demonstrate that the GWAS-associated variants within the FAM13A locus, especially rs9991328, are associated with FAM13A expression in SAT, suggesting that FAM13A might have a specific role in SAT. The study links human genetic data with in vitro and mice experiments to understand the biological role of FAM13A in regulating body fat distribution. The study shows that the presence of FAM13A risk alleles perturb the body fat distribution through limiting adipogenesis in SAT, thus FAM13A negatively regulates adipocyte development.

General comments:

In general, this is an interesting and well-done study. However, several more general and specific points need to be considered.

1. The study would be strengthened by more rigorous analyses of the cellular data involving the role of FAM13A in SGBS differentiation and mature adipocyte function. Some specific suggestions are outlined in the specific comments section below.

- We appreciate the Reviewer's suggestions regarding the cellular data. We have addressed each of the questions specifically, as shown below.

2. The molecular mechanisms linking FAM13A and adipogenesis need further investigation. For example, the Figure 6B shows a large number of differentially expressed genes between WT and KO samples,

including various genes associated with fat cell biology. The authors point out that genes in the NAD⁺ salvage pathway are also overrepresented. Given that the optimal level of intracellular NAD⁺ is important, could alternation in NAD⁺ levels serve as an upstream driver of the differentially expressed genes upon FAM13A deletion? The authors should investigate this point further to better understand the molecular mechanism of how FAM13A regulates intracellular milieu to impact fat cell biology.

- We thank the Reviewer for the insightful suggestions. We agree with the Reviewer that intracellular NAD⁺ levels in adipocytes are known to have a significant impact on whole-body insulin sensitivity by regulating adiponectin and FFA production. Confirming the contrasting levels of adipose NAD⁺ by HPLC between WT and *Fam13a* KO mice would be the first step to determine whether NAD⁺ is an intermediate player of FAM13A's action on adipocytes. Although out of scope for this paper, we are currently pursuing the molecular mechanism underlying FAM13A's role in adipose biology in the follow-up studies.

Specific comments:

1. The authors state that rs9991328 is considered a lead variant (line 99, page 4); however, a different variant (rs1377290) was used for the PheWAS analyses in figure 2A instead. The reasoning is not clear. Would the associations of rs1377290 allele with increased WHR, decreased trunk/body fat percentage, mean platelet volume, and BMI be also found using the lead variant rs9991328?

- We thank the Reviewer for the question. As explained in the text shown below, we used the directly genotyped variant rs1377290 in UK Biobank as a proxy for the imputed variant rs9991328 for the PheWAS analysis.

p5, line 8-10: We also performed a phenome-wide association study (PheWAS) in 337,536 individuals from the UK Biobank^{11, 12} using the directly genotyped variant rs1377290 as a proxy for the imputed variant rs9991328 (LD R² = 1.0).

2. Figure 3A shows that FAM13A expression is increased on day 6 of SGBS differentiation but seems to be lower by day 12. Is there any statistically significant difference between the FAM13A mRNA level at D6 as compared to D12? What does this transient increase in FAM13A during adipogenesis mean?

- We thank the Reviewer for this interesting question. The mRNA levels of *FAM13A* between D6 and D12 differentiation of SGBS are not significantly different. However, the transient increase of *FAM13A* suggested an important role of *FAM13A* in the early differentiation stage, and we provided the evidence that knockdown of *FAM13A* in preadipocytes stimulated an early onset of adipogenesis (Fig. 3B-E).

3. Figure 3C does not show any difference in PPARG mRNA expression, although the authors state that FAM13A knockdown results in the upregulation of adipogenesis markers. This statement needs to be further supported with results including mRNA levels of PPARG downstream targets, such as *Fabp4* and adiponectin. Considering the protein levels of adipogenic markers would also further support the role of FAM13A in this process. In addition, oil red o staining should be performed in scRNA and siFAM13A cells at D5 coinciding with mRNA and protein levels of adipogenic markers.

- We thank the Reviewer for the suggestion. The mRNA level of *FABP4*, the downstream target of *PPARG*, showed a trended but not statistically

significant increase in *FAM13A*-deficient cells (shown on the right), similar to *PPARG*. We agree that in the follow-up studies, complementary experiments such as ORO staining and protein measurement of adipogenic markers will provide further support to the conclusion made based on mRNA data. In addition, as Reviewer#1 suggested, we may need to carry out a time-course study to capture the most effective moment of *FAM13A* action on adipogenesis.

4. Figure S3 shows the upregulation of beige adipocyte markers in siFAM13A D5 cells as compared to scRNA. Why is it important to look at the beige adipocyte markers in this context? The authors do not clarify the significance of these findings

- We have studied the beige adipocyte markers upon a Reviewer's request in the previous revision round. We have discussed the significance of the browning effects upon *FAM13A* knockdown in the Discussion, as shown below. SGBS preadipocytes were isolated from human subcutaneous adipose tissue, which possesses the ability to differentiate into either white or beige adipocytes. Knockdown of *FAM13A* in SGBS not only increases overall adipogenesis, but also regulates cell differentiation towards the beige lineage. We modified the text in the Results to clarify the significance of the findings.

*p.11, line 4-10: SAT is known to be able to generate beige adipocytes, a mitochondria-rich and fat-burning adipocyte, while VAT is a classic depot for white adipocytes with a low number of mitochondria and limited ability for adipogenesis³². SGBS isolated from human SAT, as well as SVFs isolated from mouse SAT, showed an increased tendency to differentiate into beige adipocytes upon *FAM13A* knockdown (Supplementary Figure 3A and Supplementary Figure 8). *Fam13a* depletion in SAT may trigger a more dramatic change than in VAT by generating more beige adipocytes with improved mitochondrial efficiency.*

5. The CRISPRi should be validated through western blotting for *FAM13A* protein to support the findings in figure 3D.

- We thank the Reviewer for suggesting this important control experiment. Unfortunately, we have not yet identified an appropriate antibody to detect the long variant of *FAM13A* (117kDa), which contains the functional RhoGAP domain, in human adipocytes (see the Western Blot on the right, human *FAM13A* overexpression lysates were used as a positive control).

- Our CRISPRi sgRNAs were designed to specifically target the long isoform of *FAM13A* (as shown below). Although we have confirmed the CRISPRi knockdown of *FAM13A* in the mRNA level by using a Taqman primer (HS.PT.58.3237208, see the binding site below) specifically detecting the long variant of *FAM13A*, we agree with the Reviewer that it is important to find a working antibody that is sensitive enough to recognize the endogenous level of 117kDa *FAM13A* in SGBS

cells and validate the 117kDa protein knockdown in our CRISPRi samples in the follow-up study.

6. Figure 3E does not show any statistically significant difference in mRNA expression of *CEBPA* and *PPARG*. Additional analyses of mRNA levels of *PPARG* downstream targets, such as *Fabp4* and adiponectin, as well as western blotting of adipogenesis markers should be included.

- Similar to Question#3, we measured the mRNA level of *FABP4*, the downstream target of *PPARG*, following *FAM13A* knockdown. *FABP4* showed a trended but not statistically significant increase in *FAM13A*-deficient cells (shown on the right), similar to *PPARG*. Although beyond the scope of this paper, we agree that complementary evidence, such as protein measurement of adipogenic markers, will provide further support to the conclusion made on mRNA data.

7. To study the role of *FAM13A* in mature adipocyte function, the knockdown experiments followed by glucose uptake and lipolysis experiments should be carried out at a stage earlier than D12 when the *FAM13A* expression is still highly induced (figure 3A, comment#2). The increasing trend of glucose uptake in si*FAM13A* adipocytes is not convincing. This no change in adipocyte function upon *FAM13A* knockdown could be attributed to decreased *FAM13A* levels by D12 of SGBS differentiation.

- We agree with the Reviewer that the metabolic state of the SGBS adipocytes might be higher at an earlier stage than D12 post-differentiation. As suggested by Reviewer#1, a time-course metabolic functional study in the future will help us find the best timing to catch the maximum effects of *FAM13A* knockdown on adipocyte function.

8. Figure 4F shows an increase in BW in *FAM13A* KO males on HFD, but figure 4G shows no change in SAT weight and a decrease in VAT. What is contributing to the increase in BW? DEXA scan should be performed to find out if the lean mass is increased.

- We thank the Reviewer for asking this interesting question. There might be two potential factors contributing to the increased body weight in KO mice. 1) Increased lean mass in KO mice: as suggested by the Reviewer, EchoMRI or DEXA scan is the ideal experimental method to quantify the lean versus fat mass in the whole-body level in the follow-up study. 2) Increased SAT mass in the whole-body level: we chose to measure the weight of inguinal adipose tissue as a

representative depot for SAT. However, SAT lies all over the body and we might be underestimating the overall contribution of SAT changes to whole-body weight. As suggested by the Reviewer, DEXA scan is a more accurate method to image and quantify the body fat distribution and total amount in the future study.

9. FAM13A knockdown in SGBS cells led to a modest increase in adipogenesis (figure 3). Figure S4E and 4G do not show any difference in SAT weight in FAM13A KO mice, suggesting that the cell-autonomous role of FAM13A in regulating adipocyte differentiation is masked by the use of full-body FAM13A KO mice. The authors should mention this point in the manuscript.

- We fully agree with the Reviewer's assessment. In fact, we have already addressed this point in the manuscript:

p. 12, line 13-16: While our findings and other studies give support to the notion that adipose tissue mediates FAM13A action, the metabolic phenotypes of whole-body Fam13a KO mice are subtle. Adipose-specific knockouts of Fam13a would be useful to study the role of Fam13a specifically in adipose tissue and its causal impact on systemic metabolism.

10. Induction of beige adipocyte markers in KO D10 adipocytes (figure S8A and S8B) suggests that FAM13A deletion might promote browning of SAT correlating with an increase in smaller adipocytes. Although figure S8C suggests no change in BAT markers between WT and KO, the mRNA levels of Pgc1a, Ucp1, and Cidea should be measured in the SAT of WT and KO mice to determine if FAM13A deletion is associated with the browning of white adipose tissue.

- We agree with the Reviewer that we should have also measured the brown/beige adipocyte markers in SAT depot, in addition to the classical BAT depot. Browning/beiging effects of FAM13A is one of our main follow-up points in the future mechanistic study.

11. The glucose uptake experiments in D10 adipocytes from WT and KO shown in figure 5G and 5H are not strong findings. Therefore, it is not reasonable to suggest that FAM13A KO adipocytes are more functionally active to deposit glucose, as the authors have stated on lines 213-215 on page 8 of the manuscript.

- Since we have deleted Figure 5H and modified our statement regarding the glucose uptake results in the most recent review round (submitted in Nov. 2019), we think that the Reviewer might have made this comment based on a previous version of our manuscript. As much as we agree with the Reviewer's comments, we have already modified our manuscript as follows:

p. 8, line 16-24: To explore the functional consequences of the modestly improved subcutaneous adipogenesis, we quantified the basal and insulin-stimulated glucose uptake in newly differentiated cells and observed a non-significant trend towards induction of both basal and insulin-mediated glucose uptake in Fam13a-deficient adipocytes (Fig. 5F). To elucidate whether this slight increase in glucose uptake is due to the presence of more differentiated adipocytes in the culture of Fam13a KO observed above (Fig. 5B-E), we also performed ex vivo glucose uptake directly on SAT explants isolated from WT and Fam13a KO mice, and observed a similar directional trend (Fig. 5G). These data suggest that adipocytes from Fam13a KO mice, either differentiated in vitro or natively present in adipose tissue, are functionally active in depositing glucose and responding to insulin.

12. Given that FAM13A is mainly expressed in SAT, why does FAM13A expression goes down 30-fold in HFD VAT, but not as much in SAT and liver (figure 6A). The authors should further comment on this. Also, the expression of FAM13A in chow of VAT seems to be higher than the expression in chow of SAT, which contradicts the author's initial findings regarding FAM13A specific expression in the SAT. Western blotting for FAM13A protein using SAT and VAT samples from WT mice on a chow diet should be performed to clearly show the protein level and specificity of FAM13A expression in SAT as compared to VAT.

- We thank the Reviewer for the question. However, we would like to point out to the Reviewer that *FAM13A* expression is not specific to SAT. The disease association signal in the *FAM13A* locus obtained from GWAS colocalizes with an eQTL signal that is specific to SAT, suggesting that disease risk conferred by the variant is mediated by a change in *FAM13A* expression in SAT. Further, Supplementary Fig. 1 shows that although *FAM13A* expression in VAT is not zero, the relationship between gene expression and genotype in the *FAM13A* locus is only seen in SAT. The mouse data shown in Fig. 6A demonstrates that in response to high fat diet, *Fam13a* expression reduces in both tissues, whereas genetically conferred risk in humans via a change in *FAM13A* expression shows a SAT-specific pattern.

13. The gating strategy in figure S9C shows that live cells are represented as PI positive. This must be a mistake in the gating label. The PI staining is only taken up by the dead cells, so the cells selected with the first gate must be PI negative cells (live cells).

- We thank the Reviewer for asking this question. However, the label in Supplementary Figure 9C is "P1" (as in Portion 1), rather than "PI" (as in short for Propidium Iodide, referred to the Reviewer). We did not do PI staining. We used the forward and side scatter to gate live cells (e.g.: P1=72.1%) and then the single cells (e.g.: P2=92.1%). We labelled the percentage of each fraction in the gating strategy, according to the Journal's requirements.

14. The authors refer to figure 6C in the discussion, which is not found in the manuscript.

- We apologize for this mistake. We previously deleted Figure 6C in the modified figures but failed to delete it in the text. We thank the reviewer for noticing it.

15. There is a mistake in the figure labeling within the discussion. The statement on line 288, 289, page 11 of the manuscript should refer to figure S8 instead of S6.

- We apologize for this mistake and have corrected it in the text, as pointed out by the reviewer.

p. 11, line 6-9: SVFs isolated from SAT of Fam13a KO mice showed an increased tendency to differentiate into beige adipocytes (Supplementary Figure 3A and Supplementary Figure 8).

Reviewer #3 (Remarks to the Author):

The data are presented in a much clearer and succinct manner, addressing prior concerns.

This gene is consistently associated with fat distribution so the experiments interrogated its effects in

adipocyte, now that they are clarified and additional experiments added, so these results are important for the field

- We thank the Reviewer for all the comments and suggestion during the review process.